# Spatial Deconfounder: Interference-Aware Deconfounding for Spatial Causal Inference

Ayush Khot [* 1]   Miruna Oprescu [* 2]   Maresa Schröder [3]   Ai Kagawa [4]   Xihaier Luo [4]

## Abstract

Causal inference in spatial domains faces two intertwined challenges: (1) unmeasured spatial factors, such as weather, air pollution, or mobility, that confound treatment and outcome, and (2) interference from nearby treatments that violate standard no-interference assumptions. While existing methods typically address one by assuming away the other, we show they are deeply connected: *interference reveals structure* in the latent confounder. Leveraging this insight, we propose the **Spatial Deconfounder**, a two-stage method that reconstructs a substitute confounder from local treatment vectors using a conditional variational autoencoder (C-VAE) with a spatial prior, then estimates causal effects with a flexible outcome model. We show that this enables nonparametric identification of direct and spillover effects under weak assumptions—without multiple treatment types or a known latent-field model. Empirically, we extend `SpaCE`, a benchmark suite for spatial confounding, to include treatment interference, and show that the Spatial Deconfounder consistently improves effect estimation across real-world environmental health and social science datasets. By turning local interference into a multi-cause proxy for latent spatial confounding, our framework advances robust causal inference for spatial data.

## 1. Introduction

Causal inference in spatial settings is critical for science and policy, from estimating the health effects of pollution to

---
[1]University of Illinois at Urbana-Champaign [2] Cornell University, Cornell Tech [3]LMU Munich, Munich Center for Machine Learning (MCML) [4]Computing and Data Sciences, Brookhaven National Laboratory. Correspondence to: Ayush Khot <akhot2@illinois.edu>, Miruna Oprescu <amo78@cornell.edu>.

*Proceedings of the 43rd International Conference on Machine Learning*, Seoul, South Korea. PMLR 306, 2026. Copyright 2026 by the author(s).

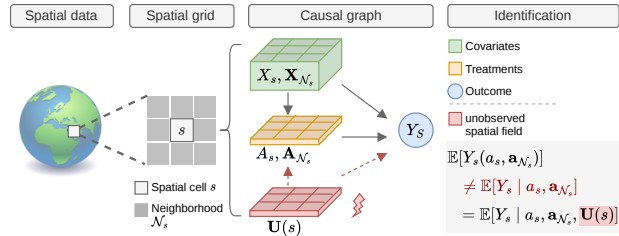

*Figure 1.* **Schematic of spatial interference/confounding**. Spatial data is represented in geographical cells indexed by site $s$ with neighborhood $\mathcal{N}_s$. The outcome at $s$ (e.g., mortality rate) is affected by the treatments (e.g., air quality) and observed confounders (e.g., demographic information) at both $s$ and $\mathcal{N}_s$. However, unobserved latent factors (e.g., humidity) can confound the relationship, rendering causal effects unidentifiable.

evaluating land use, climate interventions, and the spread of infectious disease. Most data in these domains are observational, since large-scale interventions are typically infeasible or unethical, so robust methodology is needed to draw valid conclusions. Yet observational studies in these settings face two fundamental challenges that standard methods rarely address together: (1) *spillover (interference)*, where the treatment at one site affects outcomes at nearby sites, violating the Stable Unit Treatment Value Assumption (SUTVA), and (2) *spatially structured unobserved confounding*, where latent fields such as weather or socioeconomic context jointly drive treatment exposures and outcomes. Both are pervasive, and ignoring either leads to biased conclusions.

Consider air quality and health: respiratory mortality rates depend on local pollution and on neighboring regions' pollution due to transport and mobility, while latent meteorological factors such as temperature and humidity confound both. Any method that neglects interference or hidden confounders risks misleading the actionable decisions policymakers rely on for regulation and public health.

Existing approaches for spatial causal inference typically address either interference *or* unobserved spatial confounding, but rarely both. Methods for interference model spillovers through exposure mappings or spatial/auto-regressive dependencies, enabling estimation of direct and spillover effects when all relevant confounders are observed (Hudgens & Halloran, 2008; Forastiere et al., 2021). However, these

estimators can be biased when important confounders are unmeasured. Conversely, methods for spatial treatment effect estimation under unobserved confounding use adjustment strategies such as splines, matching, or instrumental variables (Dupont et al., 2022; Papadogeorgou et al., 2019; Papadogeorgou & Samanta, 2023; Woodward et al., 2024). These approaches typically rely on explicit structure—such as smooth latent-field priors, parametric outcome models, or exclusion restrictions—and often assume away interference or absorb it only implicitly through spatial trends. As a result, they may struggle to identify and interpret spillover effects when interference is present, and can be sensitive to model misspecification.

A related but largely separate literature is the *deconfounder* framework (Wang & Blei, 2019), which shows that when each unit receives multiple treatments, their joint distribution can reveal shared latent confounders. However, existing deconfounder methods are designed for i.i.d. data with simultaneous treatments, not for spatial domains where the relevant "causes" arise through localized interactions among neighboring units. To our knowledge, no method can nonparametrically estimate treatment effects under both interference and unobserved confounding.

We close this gap with the **Spatial Deconfounder**. Our key insight is that interference *creates* the very multi-cause structure that deconfounders require: each unit receives its own treatment together with those of its neighbors, all shaped by the same latent spatial field. Rather than a nuisance, *interference becomes a source of signal for recovering hidden confounders*. Building on this, we develop a nonparametric and model-agnostic two-stage framework that first reconstructs a smooth substitute confounder using a conditional variational autoencoder (C-VAE) with a spatial prior[1], then estimates direct and spillover effects via any flexible outcome model (e.g., U-Net, GNN). This enables causal identification without requiring multiple treatment types, explicit latent-field models, or parametric outcome model specification. Our **contributions** are as follows:

1. We introduce the **Spatial Deconfounder**, a novel *nonparametric and model-agnostic* framework to *jointly* address spatial interference and unmeasured confounding by treating neighborhood treatment exposures as multi-cause signals.

2. We prove *identification* of direct and spillover effects under localized interference and a weak latent-field sufficiency assumption, without requiring a parametric model for the hidden process.

3. We extend the `SpaCE` benchmark to include structured interference and show, across climate-, health-, and

social-science datasets, that our method consistently reduces bias relative to spatial autoregressive, matching, and spline-based baselines.

By leveraging interference as a lens into the hidden structure, the Spatial Deconfounder bridges spatial causal inference and multi-cause deconfounding, opening a path to robust causal estimation in complex geographic systems.

## 2. Related Work

We give a brief overview of the related literature (see Appendix A for a comprehensive survey and discussion). Our work sits at the intersection of three main literatures: (i) spatial causal inference under interference and spatially structured confounding, (ii) deconfounding in general average treatment effect (ATE) estimation, and (iii) deep learning for spatial and latent structure modeling.

**Classical spatial causal inference.** Design- and model-based approaches for interference assume exchangeability after conditioning on *observed* covariates, often given a specified exposure mapping (E-MAP, e.g., Hudgens & Halloran, 2008; Forastiere et al., 2021; Tchetgen Tchetgen et al., 2021). Spatial econometric models capture dependence through spatial lags or autoregressive structure (e.g., Anselin, 1988) (S2SLS-LAG1, S2SLS-DURBIN), while spline- and restricted-spatial-regression approaches adjust for residual spatial trends (e.g., Hanks et al., 2015) (SPATIAL). These methods capture spatial dependence under observed-confounding assumptions, but do not resolve *unobserved* spatial confounding.

**Spatial confounding and bias-adjustment methods.** Bias from *unmeasured* spatial structure is mitigated using latent spatial effects or Bayesian priors (e.g., Hodges & Reich, 2010; Papadogeorgou & Samanta, 2023), proximity-based matching (Papadogeorgou et al., 2019) (DAPSM), orthogonalization/residualization (Dupont et al., 2022) (SPATIAL+), or instrumental variables (IVs) that leverage exogenous spatial variation (e.g., Angrist et al., 1996; Woodward et al., 2024). Recent work further shows that bias depends on the relative spatial scales of covariates and latent confounders, and that smoothing or orthogonalization succeeds only when observed covariates carry sufficient non-spatial or finer-scale variation (Paciorek, 2010; Dupont et al., 2023; Pim et al., 2026). These methods rely on explicit field models, fine-scale covariate variation, or IVs. We instead nonparametrically reconstruct latent confounding from the treatment process itself, using neighborhood treatments as the identifying multi-cause signal.

**ATE estimation under unobserved confounding.** With unmeasured confounding, point identification typically fails. Sensitivity analyses yield assumption-indexed bounds, trading point identification for robustness (e.g., VanderWeele

---

[1]We use a C-VAE instantiation in this paper, but the first-stage can be implemented with any suitable factor model that captures shared latent spatial structure.

et al., 2015; Frauen et al., 2023). Another approach is to re-construct the unobserved confounder via the *deconfounder* framework, which fits a factor model to multiple causes to infer a substitute for the latent confounder, restoring point identification (Wang & Blei, 2019; Bica et al., 2020). How-ever, existing deconfounders require many simultaneous treatments and assume no interference. We invert this: inter-ference itself yields multi-cause treatment vectors, enabling latent-field recovery even with a single treatment type.

**Deep learning for spatial modeling.** Deep spatial archi-tectures model multi-scale and long-range dependence, e.g. UNET (Ronneberger et al., 2015), GCNN (Kipf, 2016), and patch-wise/windowed transformers (e.g., Liu et al., 2021). They improve spatial prediction, but without additional causal structure do not identify treatment effects under in-terference or unobserved confounding.

**Deep latent-variable models.** C-VAEs and related deep generative models can recover latent factors from data (Kingma & Welling, 2013; Sohn et al., 2015). We adapt this idea to spatial interference: interference supplies a multi-cause signal to nonparametrically reconstruct a smooth latent confounder, enabling identification of direct and spillover effects without a specified latent field.

**Positioning of our work.** Spatial–interference methods typ-ically ignore unmeasured confounders, spatial-confounding methods rely on explicit-field models, fine-scale variation, or IV assumptions, and deconfounder methods assume i.i.d. units with multiple simultaneous causes. We show that lo-calized interference itself creates the multi-cause structure needed for deconfounding: own and neighboring treatments provide spatially indexed views of a shared latent field. This yields an interference-driven deconfounding strategy that reconstructs latent spatial confounding nonparametrically and identifies direct and spillover effects without specifying a latent-field model or requiring multiple treatment types.

## 3. Background and Setup

**Notation.** We use uppercase letters (e.g., $X$) for random variables and lowercase letters (e.g., $x$) for realizations. Bold symbols denote vectors. We write the distribution of $X$ as $P_X$, and omit subscripts when the meaning is clear.

**Data structure: lattice, neighborhoods, and observed variables.** We consider a rectangular lattice $\mathcal{S} = \{(i, j) \mid i \in [N_x], j \in [N_y]\}$, where each site $s = (i, j)$ indexes a geographic cell. For a fixed radius $r > 0$, we define the neighborhood of $s$ using the $\ell_\infty$ metric,

$$\mathcal{N}_s = \{s' \in \mathcal{S} : \|s' - s\|_\infty \leq r, \ s' \neq s\}, \quad (1)$$

where $\|s' - s\|_\infty = \max\{|i' - i|, |j' - j|\}$. Thus $\mathcal{N}_s$ is the $(2r+1) \times (2r+1)$ square centered at $s$, excluding $s$ itself. We take $r$ to be in *pixels* (multiples of the cell size), though

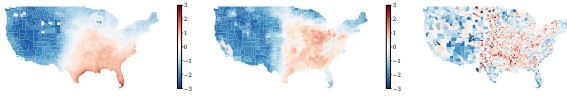

*(a)* Humidity ($U(s)$)    *(b)* PM$_{2.5}$ ($A_s$)    *(c)* Mortality ($Y_s$)

*Figure 2.* Example spatial distribution of (normalized) confounder, treatment, and outcome in real-world dataset. The confounder $U(s)$ (summer humidity) varies smoothly across space, while the treatment $A_s$ (PM$_{2.5}$) shows more local heterogeneity. The outcome $Y_s$ (respiratory and cardiovascular mortality) reflects broader spatial health patterns.

it may also be specified as a physical distance and mapped to the grid resolution. Other shapes (e.g., $\ell_2$ balls) are possible, but we use the $\ell_\infty$ ball for computational convenience.

At each site $s$ we observe covariates $\mathbf{X}_s \in \mathbb{R}^{d_x}$, a binary treatment $A_s \in \{0, 1\}$, and an outcome $Y_s \in \mathbb{R}$. For a neighborhood $\mathcal{N}_s$, we write $\mathbf{X}_{\mathcal{N}_s} = \{\mathbf{X}_{s'} : s' \in \mathcal{N}_s\}$, and analogously $A_{\mathcal{N}_s}$ and $Y_{\mathcal{N}_s}$. Realizations are denoted in low-ercase, e.g., $\boldsymbol{x}_s$, $a_s$, $y_s$, and $\boldsymbol{x}_{\mathcal{N}_s} = \{\boldsymbol{x}_{s'} : s' \in \mathcal{N}_s\}$. For clarity, we focus on binary treatments, but the framework ex-tends to continuous or multi-valued treatments via standard generalizations of the potential outcomes framework.

**Potential outcomes and interference.** We adopt Rubin's potential outcomes framework (Rubin, 2005). Standard causal inference relies on SUTVA, which rules out inter-ference, i.e., one unit's outcome cannot depend on others' treatments. In spatial settings, this assumption is often vi-olated, since treatment exposures spill over. We assume *localized interference*: the potential outcome at site $s$ de-pends only on its own treatment and those of its neighbors,

$$Y_s(\mathbf{a}) = Y_s(a_s, \mathbf{a}_{\mathcal{N}_s}), \quad (2)$$

where $\mathbf{a}$ is the full treatment vector, $a_s$ the treatment at $s$, and $\mathbf{a}_{\mathcal{N}_s} = \{a_{s'} : s' \in \mathcal{N}_s\}$. The observed data contain only the realized outcome $Y_s = Y_s(A_s, \mathbf{A}_{\mathcal{N}_s})$ under the assigned intervention.

**Causal estimands.** Let $\mathbf{a}_{\mathcal{N}_s}^{(1)}$ and $\mathbf{a}_{\mathcal{N}_s}^{(0)}$ be two realizations of the neighbor treatments. Our targets are (i) the *average direct effect*, which varies the unit's own treatment while holding neighbors fixed,

$$\tau_{\text{dir}} = \mathbb{E}\big[Y_s(1, \mathbf{a}_{\mathcal{N}_s}) - Y_s(0, \mathbf{a}_{\mathcal{N}_s})\big], \quad (3)$$

and (ii) the *average spillover effect*, which varies neighbors' treatments while holding the unit fixed,

$$\tau_{\text{spill}} = \mathbb{E}\big[Y_s(a, \mathbf{a}_{\mathcal{N}_s}^{(1)}) - Y_s(a, \mathbf{a}_{\mathcal{N}_s}^{(0)})\big], \quad a \in \{0, 1\}, \quad (4)$$

with expectations taken over the observed joint distribution of $(\mathbf{X}_s, A_{\mathcal{N}_s})$.

**Unobserved spatial confounding.** To identify the treatment effects in Equations (3) and (4), one typically assumes *ignor-ability*: potential outcomes $Y_s(a_s, \mathbf{a}_{\mathcal{N}_s})$ are independent of

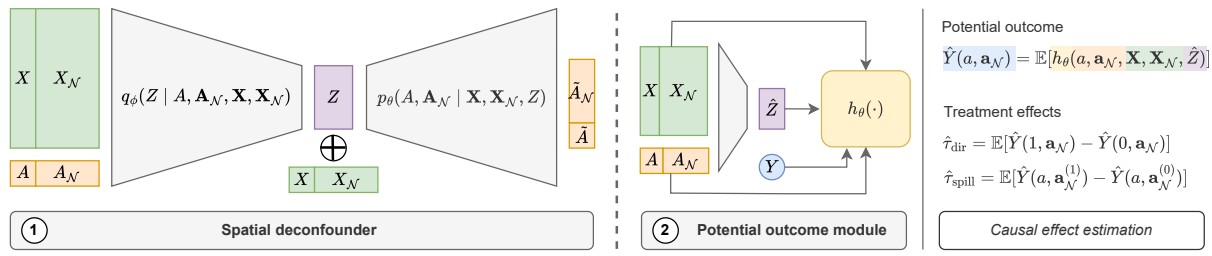

*Figure 3.* **Architecture of the spatial deconfounder & estimation framework.** Stage ①: The C-VAE takes treatments and observed confounders as input to learn the latent substitute confounder. Stage ②: We employ the reconstructed confounder together with the observed variables (now including the outcome) to train the potential outcome estimation module.

treatment assignment given observed covariates $(\mathbf{X}_s, \mathbf{X}_{\mathcal{N}_s})$. This assumption cannot be tested, and violations lead to biased causal estimates. In practice, many relevant drivers of treatment exposure and outcome remain unobserved. We posit an unobserved spatial field $U : \mathcal{S} \to \mathbb{R}^{d_U}$ that captures latent influences such as topography, wind patterns, or socioeconomic context. Because $U(s)$ may affect both treatment and outcomes, we generally have

$$\text{Cov}(A_s, U(s)) \neq 0 \quad \text{and} \quad \text{Cov}(Y_s(a, \mathbf{a}_{\mathcal{N}_s}), U(s)) \neq 0, \quad (5)$$

where the covariances are understood component-wise when $U(s)$ is vector-valued. Thus, ignorability fails when conditioning only on $\mathbf{X}_s$ and $\mathbf{X}_{\mathcal{N}_s}$. In Section 5, we show that identification can nevertheless be recovered under mild smoothness assumptions on $U$ together with our deconfounding procedure, through reconstructing a substitute latent field from observed treatment patterns.

**Motivating example.** Consider real environmental health data on a $0.25° \times 0.25°$ grid covering the continental United States. At each grid cell $s$, the treatment $A_s$ indicates whether fine particulate matter ($\text{PM}_{2.5}$) exceeds the WHO guideline of 10 $\mu$g/m³. Neighbor assignments are defined by a radius of one to two grid cells (roughly 25–50 km). The outcome $Y_s$ is the rate of respiratory and cardiovascular mortality aggregated from hospital records. Latent factors can confound this relationship; for example, a meteorological driver such as humidity varies smoothly across space and may jointly influence both pollution exposures and health outcomes. Figure 2 illustrates treatment, outcome, and such a confounder for this dataset. This example captures the type of smoothly varying, spatially shared latent structure our method targets: large-scale meteorological drivers such as humidity form a latent field $U(s)$ that jointly affects $\text{PM}_{2.5}$ exposures and mortality across neighboring counties, while any purely local one-off factors are captured in $(X_s, X_{\mathcal{N}_s})$ or assumed negligible. We formalize this as a latent-field sufficiency assumption in Section 5.

The remainder of the paper shows how the joint vector $(A_s, \mathbf{A}_{\mathcal{N}_s})$—a "multiple-cause" analogue supplied for free by interference—can be harnessed to reconstruct $U(s)$ and obtain unbiased estimates of Equations (3) and (4).

## 4. Methodology

As illustrated in Algorithm 1, our approach proceeds in two stages. First, we reconstruct a smooth substitute confounder from the joint distribution of local and neighbor treatments, using a conditional variational autoencoder (C-VAE) that leverages interference as a multi-cause signal. Second, we feed the reconstructed confounder into a flexible potential outcome module for outcome modeling and effect estimation. This separation follows standard practice in deconfounding to prevent mediators from being inadvertently learned into the substitute confounder, which would break the identifiability of the treatment effects.

---

**Algorithm 1** Spatial Deconfounder

---

**Input:** Spatial covariates $\{\mathbf{X}_s\}_{s \in \mathcal{S}}$, treatments $\{A_s\}_{s \in \mathcal{S}}$, outcomes $\{Y_s\}_{s \in \mathcal{S}}$, neighborhood radius $r$, grid Laplacian $L$

1: **Stage ①: Confounder reconstruction (C-VAE)**
2: Let encoder $q_\phi(Z_s \mid A_s, A_{\mathcal{N}_s}, \mathbf{X}_s, \mathbf{X}_{\mathcal{N}_s}) = \mathcal{N}(\mu_\phi, \text{diag } \sigma_\phi^2)$, decoder $p_\psi(A_s \mid \mathbf{X}_s, \mathbf{X}_{\mathcal{N}_s}, Z_s)$, and GMRF prior $p_\theta(Z) = \mathcal{N}(\mathbf{0}, \tau^{-1}(L + \epsilon I)^{-1})$.
3: Minimize

$$\mathcal{L}_A = \sum_s \mathbb{E}_{q_\phi}\big[-\log p_\psi(A_s \mid \mathbf{X}_s, \mathbf{X}_{\mathcal{N}_s}, Z_s)\big] + \sum_s D_{\text{KL}}\big(q_\phi \,\|\, p_\psi\big),$$

4: Set substitute confounder $\hat{Z}_s \leftarrow \mathbb{E}_{q_\phi}[Z_s]$ for all $s$.
5: **Stage ②: Potential outcome module**
6: Choose a spatial model $h$ (e.g., U-Net) to model the conditional expectation of $Y$ given all observed variables as well as the substitute confounder and fit by minimizing

$$\mathcal{L}_Y = \sum_s \Big(Y_s - h(A_s, A_{\mathcal{N}_s}, \mathbf{X}_s, \mathbf{X}_{\mathcal{N}_s}, \hat{Z}_s)\Big)^2.$$

7: Estimate effects by plug-in contrasts (Eq. 11).

---

**Stage ①: Confounder reconstruction.** We model the assignment of treatments $\{A_s\}_{s \in \mathcal{S}}$ using an interference-aware C-VAE. The encoder

$$q_\phi(Z_s \mid A_s, A_{\mathcal{N}_s}, \mathbf{X}_s, \mathbf{X}_{\mathcal{N}_s}) = \mathcal{N}\big(\mu_\phi(\cdot), \text{diag } \sigma_\phi^2(\cdot)\big) \quad (6)$$

maps the local treatment and neighborhood treatments, together with local and neighborhood covariates $(\mathbf{X}_s, \mathbf{X}_{\mathcal{N}_s})$,

into a latent embedding $Z_s$ of the unobserved spatial field $U(s)$. The decoder

$$p_\psi(A_s \mid \mathbf{X}_s, \mathbf{X}_{\mathcal{N}_s}, Z_s) = \sigma(f_\psi(\mathbf{X}_s, \mathbf{X}_{\mathcal{N}_s}, Z_s)) \quad (7)$$

predicts $A_s$ given covariates and the latent. To encode smoothness, we impose a Gaussian–Markov random-field (GMRF) prior $p_\theta(Z) = \mathcal{N}(\mathbf{0}, \tau^{-1}(L + \epsilon I)^{-1})$ with grid Laplacian $L$, or equivalently a deterministic penalty $\lambda Z^\top L Z$; anisotropic or directed precision matrices could be used when domain knowledge suggests directional dependence.

Formally, our generative model for the treatment field is

$$p_\theta(Z) = \mathcal{N}\big(\mathbf{0}, \tau^{-1}(L + \epsilon I)^{-1}\big),$$
$$p(A \mid X, Z) = \prod_{s \in \mathcal{S}} p_\psi\big(A_s \mid X_s, X_{\mathcal{N}_s}, Z_s\big),$$

with $A_s \mid X_s, X_{\mathcal{N}_s}, Z_s \sim \text{Bernoulli}\big(\sigma(f_\psi(X_s, X_{\mathcal{N}_s}, Z_s))\big)$. Thus, conditional independence of treatments holds across sites given $(Z, X)$, and spatial dependence is encoded entirely via the GMRF prior on $Z$. The "multi-cause" structure of $(A_s, A_{\mathcal{N}_s})$ enters on the inference side through the encoder $q_\phi(Z_s \mid A_s, A_{\mathcal{N}_s}, X_s, X_{\mathcal{N}_s})$, which uses local treatment patterns (plus covariates) to infer a substitute confounder for the local value of the spatial latent field.

This C-VAE is trained by minimizing

$$\mathcal{L}_A(\phi, \psi) = \sum_s \mathbb{E}_{q_\phi}\big[-\log p_\psi(A_s \mid \mathbf{X}_s, \mathbf{X}_{\mathcal{N}_s}, Z_s)\big] \quad (8)$$
$$+ \beta \sum_s D_{\text{KL}}(q_\phi \| p_\psi),$$

with KL warm-up ($\beta \uparrow 1$). After convergence, we set $\hat{Z}_s = \mathbb{E}_{q_\phi}[Z_s]$ as the reconstructed confounder.

Our C-VAE differs from standard C-VAE-type models in two ways tailored to the spatial–interference setting: (i) the encoder explicitly conditions on $(A_s, A_{\mathcal{N}_s}, X_s, X_{\mathcal{N}_s})$, using neighbor treatments as a multi-cause signal, and (ii) the latent field $Z$ is given a GMRF prior with grid Laplacian $L$, enforcing spatial dependence consistent with our latent-field sufficiency assumption (Assumption 4 below).

**Stage ②: Potential outcome module.** Given $\hat{Z}_s$, we estimate outcomes using a flexible function $h$:

$$\hat{Y}_s = \hat{\mathbb{E}}[Y \mid A_s, A_{\mathcal{N}_s}, \mathbf{X}_s, \mathbf{X}_{\mathcal{N}_s}, \hat{Z}_s] \quad (9)$$
$$= h(A_s, A_{\mathcal{N}_s}, \mathbf{X}_s, \mathbf{X}_{\mathcal{N}_s}, \hat{Z}_s)$$

by minimizing the squared error loss

$$\mathcal{L}_Y = \sum_s \Big(Y_s - h(A_s, A_{\mathcal{N}_s}, \mathbf{X}_s, \mathbf{X}_{\mathcal{N}_s}, \hat{Z}_s)\Big)^2. \quad (10)$$

This module can be instantiated with any spatial model capable of handling interference and spatial confounding. For example, a U-Net architecture (Ronneberger et al., 2015) captures multi-scale spatial dependencies through an encoder–decoder with skip connections. Notably, Oprescu et al. (2025); Ali et al. (2024) use U-Net-based architectures to account for interference and spatial confounding in spatiotemporal settings. The framework is not tied to this choice or to regular lattices: depending on the data modality, the outcome head can be replaced by patch-wise transformers, classical spatial regression models, or graph neural networks on irregular spatial units (Kipf, 2016). In the latter case, the GMRF prior can be replaced by a graph-Laplacian prior and the C-VAE can operate on arbitrary neighborhoods.

Effect estimation proceeds by plug-in contrasts: the *direct effect* is

$$\hat{\tau}_{\text{dir}} = \frac{1}{|\mathcal{S}|} \sum_{s \in \mathcal{S}} \Big[ h(1, A_{\mathcal{N}_s}, \mathbf{X}_s, \mathbf{X}_{\mathcal{N}_s}, \hat{Z}_s) \quad (11)$$
$$- h(0, A_{\mathcal{N}_s}, \mathbf{X}_s, \mathbf{X}_{\mathcal{N}_s}, \hat{Z}_s) \Big],$$

and analogously for spillover effects by varying $\mathbf{A}_{\mathcal{N}_s}$. By drawing multiple $\hat{Z}_s$ from the full posterior $q_\phi$ instead of the mean, we can obtain uncertainty bands on $\hat{Z}_s$. We can then obtain uncertainty bands (with respect to the substitute confounder) by evaluating Eq. 11 on different draws of $\hat{Z}_s$.

**Predictive checks.** Following Rubin (1984), we assess whether the substitute confounder adequately explains the treatment assignment through posterior predictive checks. On a held-out validation set, we draw $M$ replicated treatment vectors $\mathbf{a}^{(1)}, \dots, \mathbf{a}^{(M)}$ from the decoder $p_\psi$ and compare them against the observed assignment $\mathbf{a}$. Specifically, we compute the predictive $p$-value

$$p = \frac{1}{M} \sum_{m=1}^{M} \mathbf{1}\Big\{ T(\mathbf{a}^{(m)}) < T(\mathbf{a}) \Big\}, \quad (12)$$

where $T(\mathbf{a})$ is a discrepancy statistic measuring model fit. Following Wang & Blei (2019), we use

$$T(\mathbf{a}) = \mathbb{E}_{Z \sim q_\phi}[\log p_\psi(\mathbf{a} \mid \mathbf{X}, Z)], \quad (13)$$

the marginal log-likelihood of the observed assignment under the posterior distribution of $Z$. A value of $p$ close to $0.5$ indicates that the C-VAE reproduces the treatment assignment distribution well, whereas extreme values signal misspecification or poor substitute-confounder recovery. We therefore use $p$ as a practical diagnostic before trusting effect estimates: since proxy error in $\hat{Z}$ translates into residual confounding bias (see Proposition 1 in Appendix B), we only consider C-VAE models with $0.25 < p < 0.75$.

# 5. Theoretical Properties of the Spatial Deconfounder

We now provide conditions under which the Spatial Deconfounder establishes causal identifiability of the direct and spillover effects in Equations (3) and (4). Our argument separates two steps: (i) an *identification* step showing that if a substitute confounder from observed neighborhood exposures exists, then direct and spillover effects are identified; and (ii) an *estimation* step stating conditions under which our Stage ① procedure (a C-VAE instantiation of a conditional factor model) consistently recovers this target.

We begin with assumptions on consistency, positivity, and interference structure.

**Assumption 1** (Spatial consistency). *The observed outcome equals the potential outcome under the assigned individual and neighborhood treatments. That is,*

$$Y_s = Y_s(a_s, \mathbf{a}_{\mathcal{N}_s})$$

*if a site $s$ receives treatment $a_s$ and its neighborhood $\mathcal{N}_s$ receives the vector of treatments $\mathbf{a}_{\mathcal{N}_s}$.*

**Assumption 2** (Spatial positivity). *For any site $s$, covariates $(\mathbf{X}_s, \mathbf{X}_{\mathcal{N}_s})$, and treatment exposures $(a_s, \mathbf{a}_{\mathcal{N}_s})$, the probability of assignment is strictly positive: $0 < \Pr(a_s, \mathbf{a}_{\mathcal{N}_s} \mid \mathbf{X}_s, \mathbf{X}_{\mathcal{N}_s}) < 1$. Furthermore, we require* latent positivity *conditional on Z, i.e., $0 < \Pr(a_s, \mathbf{a}_{\mathcal{N}_s} \mid \mathbf{X}_s, \mathbf{X}_{\mathcal{N}_s}, \mathbf{Z}_s) < 1$ if $\Pr(a_s, \mathbf{a}_{\mathcal{N}_s}, \mathbf{X}_s, \mathbf{X}_{\mathcal{N}_s}, \mathbf{Z}_s) > 0$.*

**Assumption 3** (Localized interference). *The potential outcome at site $s$ depends only on its own treatment and those of its neighbors $\mathcal{N}_s$, not on treatments outside $\mathcal{N}_s$.*

Assumptions 1–3 are standard in the causal inference literature (e.g., Chen et al., 2024; Forastiere et al., 2021) and ensure that the potential outcomes and the direct/spillover estimands in Equations (3) and (4) are well-defined under localized interference. Identification additionally requires assumptions on the confounding structure. Classical approaches for spatial treatment effects assume ignorability of the joint neighborhood exposure given observed covariates; we relax this and allow unobserved confounding driven by a shared latent spatial field $U : \mathcal{S} \to \mathbb{R}^{d_U}$ spanning the grid, while requiring that confounders affecting purely local variation are observed in $(\mathbf{X}_s, \mathbf{X}_{\mathcal{N}_s})$.

**Assumption 4** (Latent field sufficiency). *All confounders that act only on a single site are observed in $(\mathbf{X}_s, \mathbf{X}_{\mathcal{N}_s})$. Any remaining unobserved confounding is mediated through a shared spatial latent field $U : \mathcal{S} \to \mathbb{R}^{d_U}$ that affects treatment assignments across multiple sites. In particular, there is no additional unobserved confounder $\tilde{U}$ that changes $(A_s, A_{\mathcal{N}_s}, Y_s(a, \mathbf{a}_{\mathcal{N}_s}))$ at some site $s$ without also influencing treatments at other sites $s'$.*

Assumption 4 is the spatial analogue of the "no single-cause confounders" assumption in the deconfounder literature

(e.g., Wang & Blei, 2019; Bica et al., 2020): all purely local confounders are observed, and any remaining unobserved confounding arises from a shared latent field $U$ that induces dependence across sites. This is precisely the regime in which the neighborhood exposure $(A_s, A_{\mathcal{N}_s})$ can act as a multi-cause signal: multiple components of exposure are jointly shaped by the same latent spatial structure. Under a factor-model representation of the joint exposure, Proposition 5 of Wang & Blei (2019) implies that there exists a *population* substitute confounder $Z_s^\star$ (measurable with respect to $(A_s, A_{\mathcal{N}_s}, \mathbf{X}_s, \mathbf{X}_{\mathcal{N}_s})$) such that the joint assignment $(A_s, A_{\mathcal{N}_s})$ is ignorable given $(\mathbf{X}_s, \mathbf{X}_{\mathcal{N}_s}, Z_s^\star)$.

Finally, we connect this population target to what our Stage ① model learns.

**Assumption 5** (Recoverable substitute confounder and Stage ① consistency). *There exists a population substitute confounder $Z_s^\star$ that is a deterministic function of the observed neighborhood exposure and covariates,*

$$Z_s^\star = f_\phi(A_s, A_{\mathcal{N}_s}, \mathbf{X}_s, \mathbf{X}_{\mathcal{N}_s}),$$

*such that conditioning on $(\mathbf{X}_s, \mathbf{X}_{\mathcal{N}_s}, Z_s^\star)$ renders the joint exposure $(A_s, A_{\mathcal{N}_s})$ ignorable as in Definition 1. Moreover, the fitted Stage ① model yields an estimator*

$$\hat{Z}_s = f_{\hat{\phi}}(A_s, A_{\mathcal{N}_s}, \mathbf{X}_s, \mathbf{X}_{\mathcal{N}_s})$$

*that converges to $Z_s^\star$ (e.g., $q_{\hat{\phi}}(Z_s \mid A_s, A_{\mathcal{N}_s}, \mathbf{X}_s, \mathbf{X}_{\mathcal{N}_s})$ concentrates at $Z_s^\star$) as the sample size grows.*

Assumption 5 states two requirements: (i) an *identification* requirement—that a population substitute confounder $Z_s^\star$ measurable with respect to $(A_s, A_{\mathcal{N}_s}, \mathbf{X}_s, \mathbf{X}_{\mathcal{N}_s})$ exists and restores ignorability—and (ii) an *estimation* requirement—that the chosen Stage ① factor model consistently recovers $Z_s^\star$. We do not claim that C-VAEs are identifiable in full generality; rather, the assumption should be read as a well-specification/consistency condition on the selected conditional factor model class. In practice, we encourage stable recovery by incorporating spatial priors and can further regularize toward identifiability using objectives such as the IMA-regularized loss of Reizinger et al. (2022).

**Remark 1** (Relation to known deconfounder limitations). *Critiques of the "deconfounder" methods note that unconstrained factor models may admit multiple substitute confounders consistent with the observed treatment distribution, leading to non-identifiability of causal effects (D'Amour, 2019). We do not claim to avoid this issue in full generality. Our setting mitigates it by adding structure relative to the generic multi-cause case: the GMRF prior restricts Z to smooth spatial fields, and the multi-cause signal comes from neighboring treatments shaped by the same latent spatial driver, rather than unrelated causes such as items in a recommender system. These restrictions make substitute-confounder recovery more plausible in the regimes we target, though not guaranteed.*

**Intuition.** Under interference, each site's treatment is observed together with those of its neighbors. Because both $A_s$ and $A_{\mathcal{N}_s}$ are influenced by the same latent field $U$, they provide multiple noisy "views" of the underlying spatial structure. By fitting a factor model to the joint distribution of own and neighbor treatments, we target the population substitute confounder $Z_s^\star$ and estimate it with $\hat{Z}_s$. Conditioning on this substitute confounder (together with observed covariates) restores ignorability, enabling estimation of direct and spillover effects.

**Sensitivity to proxy error.** In Appendix B, we show that if the outcome regression is Lipschitz in $Z$, then using $\hat{Z}$ instead of $Z^\star$ induces $O(\mathbb{E}\|\hat{Z} - Z^\star\|)$ error in the direct and spillover treatment effects.

For notational simplicity, we write $Z_s$ for the population target $Z_s^\star$ in what follows.

**Theorem 1** (Causal identifiability). *Suppose Assumptions 1–5 hold. Let $Z_s$ be a piecewise constant function of the assigned neighborhood exposure and covariates $(a, \mathbf{a}_{\mathcal{N}}, \boldsymbol{x}, \boldsymbol{x}_{\mathcal{N}})$ and let the outcome be a separable function of the observed and unobserved variables:*

$$\mathbb{E}_Y\big[Y_s(a, \mathbf{a}_{\mathcal{N}}) \,|\, \mathbf{X}_s = \boldsymbol{x}, \mathbf{X}_{\mathcal{N}_s} = \boldsymbol{x}_{\mathcal{N}}, Z_s = z\big]$$
$$= f_1(a, \mathbf{a}_{\mathcal{N}}, \boldsymbol{x}, \boldsymbol{x}_{\mathcal{N}}) + f_2(z), \tag{14}$$

$$\mathbb{E}_Y\big[Y_s \,|\, A_s = a, \mathbf{A}_{\mathcal{N}_s} = \mathbf{a}_{\mathcal{N}}, \mathbf{X}_s = \boldsymbol{x}, \mathbf{X}_{\mathcal{N}_s} = \boldsymbol{x}_{\mathcal{N}}, Z_s = z\big]$$
$$= f_3(a, \mathbf{a}_{\mathcal{N}}, \boldsymbol{x}, \boldsymbol{x}_{\mathcal{N}}) + f_4(z), \tag{15}$$

*for continuously differentiable functions $f_1, f_2, f_3, f_4$. Consequently, the direct and spillover effects are identifiable as*

$$\tau_{\mathrm{dir}} = \mathbb{E}_{\mathbf{X}_s, \mathbf{x}_{\mathcal{N}_s}, Z}\Big[\mathbb{E}_Y\big[Y_s \,|\, A_s = 1, \mathbf{A}_{\mathcal{N}_s}, \mathbf{X}_s, \mathbf{X}_{\mathcal{N}_s}, Z_s\big]$$
$$- \mathbb{E}_Y\big[Y_s \,|\, A_s = 0, \mathbf{A}_{\mathcal{N}_s}, \mathbf{X}_s, \mathbf{X}_{\mathcal{N}_s}, Z_s\big]\Big], \tag{16}$$

$$\tau_{\mathrm{spill}} = \mathbb{E}_{\mathbf{X}_s, \mathbf{x}_{\mathcal{N}_s}, Z}\Big[\mathbb{E}_Y\big[Y_s \,|\, a, \mathbf{A}_{\mathcal{N}_s} = \mathbf{a}_{\mathcal{N}_s}^{(1)}, \mathbf{X}_s, \mathbf{X}_{\mathcal{N}_s}, Z_s\big]$$
$$- \mathbb{E}_Y\big[Y_s \,|\, a, \mathbf{A}_{\mathcal{N}_s} = \mathbf{a}_{\mathcal{N}_s}^{(0)}, \mathbf{X}_s, \mathbf{X}_{\mathcal{N}_s}, Z_s\big]\Big]. \tag{17}$$

*Proof.* The proof is provided in Appendix B. $\square$

**Remark 2** (Separability). *Our identifiability result applies to settings with separable structural equations, a standard assumption in related work (e.g., Wang & Blei, 2019; Papadogeorgou & Samanta, 2023). In spatial applications, this can capture latent factors that shift outcomes but are not fully observed, such as baseline respiratory risk from long-run pollution exposure, chronic disease burden, or regional variation in care-seeking. Systematic outcome measurement error can be viewed similarly.*

## 6. Experiments

We evaluate the Spatial Deconfounder on semi-synthetic datasets from the SpaCE benchmark (Tec et al., 2024), modified to incorporate both local interference and spatial

confounding on real-world environmental data. To simulate unobserved confounding, we mask key covariates after data generation, i.e., we completely remove them from the dataset. We then compare different instantiations of our method against a range of spatial baselines under both local and spatial confounding scenarios. The section proceeds as follows: we describe the SpaCE environment and our data generation process, introduce the baselines and evaluation metrics, and finally interpret the results.

Additional details—including data generation, residual sampling, packages, hyperparameter tuning, and validation procedures—can be found in Appendix C. Replication code is available at https://github.com/moprescu/Spatial-Deconfounder.

**Datasets and the SpaCE benchmark.** We build on the SpaCE benchmark (Tec et al., 2024), which constructs realistic semi-synthetic spatial causal datasets from real treatments, covariates, outcomes, and spatial graphs. SpaCE fits flexible machine-learning models to real outcomes and then generates synthetic potential outcomes with decoupled residual errors, providing known counterfactuals for evaluation. Its original data-generating process targets spatial confounding, but does not explicitly model localized treatment interference or define direct and spillover effects. As a result, it cannot be used to directly evaluate methods, such as ours, that target both unobserved spatial confounding and localized spillover effects.

To address this, we extend the SpaCE data generation process in two ways. First, we project the raw environmental data onto a uniform $0.25° \times 0.25°$ latitude–longitude grid, allowing convolutional architectures to exploit spatial locality while preserving large-scale patterns. Second, we incorporate *interference* into the potential outcome model by allowing outcomes to depend not only on local treatment $A_s$ but also on neighbor treatments $A_{\mathcal{N}_s}$ within radius $r_d$. We generate outcomes under two confounding regimes:

$$\text{(Local confounding)} \quad \hat{Y}_s = f(A_s, A_{\mathcal{N}_s}, X_s) + R_s, \tag{18}$$

$$\text{(Spatial confounding)} \quad \hat{Y}_s = f(A_s, A_{\mathcal{N}_s}, X_s, X_{\mathcal{N}_s}) + R_s, \tag{19}$$

where $f$ is a predictive function learned from the observed data, $X_s$ are observed covariates, and $R_s$ are exogenous residuals. The local setting restricts confounding to site-level variables, while the spatial setting also allows neighborhood covariates to act as confounders.

**Semi-synthetic data generation.** To construct $\hat{Y}_s$, we proceed in four steps: (1) fit $f$ using ensembles of machine learning models to predict observed outcomes $Y_s$, (2) compute residuals $\hat{R}_s = Y_s - f(\cdot)$ and estimate their spatial distribution $P_R$, (3) replace endogenous residuals with exogenous noise $R_s \sim P_R$, and (4) generate counterfactuals by varying local and neighbor treatments while holding

*Table 1.* Performance under *local confounding*. Results averaged over 10 runs with 95% confidence intervals. $r_d$: neighborhood radius in data generation; $R$: neighborhood radius used by the deconfounder. Lower values for DIR and SPILL indicate less bias. $p$ indicates the predictive-check $p$-value, with values near 0.5 indicating good model fit. Best and second-best values are bolded and underlined, respectively.

| Env | Conf | Method | DIR | SPILL | $p$ |
|---|---|---|---|---|---|
| $PM_{2.5}$ $\downarrow$ $m$ ($r_d=1$) | $\rho_{pop}$ | C-VAE-SPATIAL+ ($R=1$) | **0.02 ± 0.01** | **0.02 ± 0.00** | 0.51 ± 0.07 |
| | | C-VAE-SPATIAL+ ($R=2$) | 0.04 ± 0.01 | 0.04 ± 0.01 | 0.50 ± 0.05 |
| | | DAPSM | 0.25 ± 0.01 | n/a | n/a |
| | | GCNN | 0.36 ± 0.03 | n/a | n/a |
| | | GMERROR | 0.03 ± 0.00 | n/a | n/a |
| | | DURBIN | 0.03 ± 0.00 | 0.07 ± 0.00 | n/a |
| | | S2SLS-LAG1 | 0.03 ± 0.00 | 0.07 ± 0.00 | n/a |
| | | SPATIAL+ | 0.05 ± 0.02 | n/a | n/a |
| | | SPATIAL | **0.02 ± 0.00** | n/a | n/a |
| | | E-MAP-SPATIAL+ ($R=1$) | 0.04 ± 0.02 | 0.07 ± 0.07 | n/a |
| | | E-MAP-SPATIAL ($R=1$) | **0.02 ± 0.00** | 0.07 ± 0.07 | n/a |
| | $q_{summer}$ | C-VAE-SPATIAL+ ($R=1$) | **0.02 ± 0.01** | **0.02 ± 0.00** | 0.53 ± 0.05 |
| | | C-VAE-SPATIAL+ ($R=2$) | 0.04 ± 0.01 | 0.04 ± 0.01 | 0.51 ± 0.05 |
| | | DAPSM | 0.30 ± 0.03 | n/a | n/a |
| | | GCNN | 0.41 ± 0.03 | n/a | n/a |
| | | GMERROR | 0.20 ± 0.00 | n/a | n/a |
| | | DURBIN | 0.20 ± 0.00 | 0.06 ± 0.00 | n/a |
| | | S2SLS-LAG1 | 0.20 ± 0.00 | 0.06 ± 0.00 | n/a |
| | | SPATIAL+ | 0.05 ± 0.02 | n/a | n/a |
| | | SPATIAL | **0.02 ± 0.00** | n/a | n/a |
| | | E-MAP-SPATIAL+ ($R=1$) | 0.05 ± 0.02 | 0.07 ± 0.07 | n/a |
| | | E-MAP-SPATIAL ($R=1$) | **0.02 ± 0.00** | 0.07 ± 0.07 | n/a |
| $SO_4$ $\downarrow$ $PM_{2.5}$ ($r_d=2$) | $NH_4$ | C-VAE-SPATIAL+ ($R=1$) | **0.02 ± 0.00** | 0.32 ± 0.00 | 0.51 ± 0.02 |
| | | C-VAE-SPATIAL+ ($R=2$) | **0.02 ± 0.00** | 0.32 ± 0.00 | 0.49 ± 0.03 |
| | | DAPSM | 1.23 ± 0.00 | n/a | n/a |
| | | GCNN | 0.26 ± 0.09 | n/a | n/a |
| | | GMERROR | 0.10 ± 0.00 | n/a | n/a |
| | | DURBIN | 0.10 ± 0.00 | 0.55 ± 0.00 | n/a |
| | | S2SLS-LAG1 | 0.10 ± 0.00 | 0.49 ± 0.00 | n/a |
| | | SPATIAL+ | 0.04 ± 0.00 | n/a | n/a |
| | | SPATIAL | 0.40 ± 0.00 | n/a | n/a |
| | | E-MAP-SPATIAL+ ($R=2$) | 0.04 ± 0.00 | 0.36 ± 0.00 | n/a |
| | | E-MAP-SPATIAL ($R=2$) | 0.40 ± 0.00 | 0.36 ± 0.01 | n/a |
| | $OC$ | C-VAE-SPATIAL+ ($R=1$) | **0.03 ± 0.00** | 0.32 ± 0.00 | 0.50 ± 0.03 |
| | | C-VAE-SPATIAL+ ($R=2$) | **0.03 ± 0.00** | 0.32 ± 0.00 | 0.50 ± 0.04 |
| | | DAPSM | 1.24 ± 0.01 | n/a | n/a |
| | | GCNN | 0.30 ± 0.10 | n/a | n/a |
| | | GMERROR | 0.21 ± 0.00 | n/a | n/a |
| | | DURBIN | 0.22 ± 0.00 | 0.58 ± 0.00 | n/a |
| | | S2SLS-LAG1 | 0.21 ± 0.00 | 0.49 ± 0.00 | n/a |
| | | SPATIAL+ | **0.03 ± 0.00** | n/a | n/a |
| | | SPATIAL | 0.40 ± 0.00 | n/a | n/a |
| | | E-MAP-SPATIAL+ ($R=2$) | **0.03 ± 0.00** | 0.37 ± 0.00 | n/a |
| | | E-MAP-SPATIAL ($R=2$) | 0.40 ± 0.00 | 0.36 ± 0.01 | n/a |

*Table 2.* Performance under *spatial confounding*. Results averaged over 10 runs with 95% confidence intervals. $r_d$: neighborhood radius in data generation; $R$: neighborhood radius used by the deconfounder. Lower values for DIR and SPILL indicate less bias. $p$ indicates the predictive-check $p$-value, with values near 0.5 indicating good model fit. Best and second-best values are bolded and underlined, respectively.

| Env | Conf | Method | DIR | SPILL | $p$ |
|---|---|---|---|---|---|
| $PM_{2.5}$ $\downarrow$ $m$ ($r_d=1$) | $\rho_{pop}$ | C-VAE-UNET ($R=1$) | 0.06 ± 0.02 | 0.16 ± 0.08 | 0.53 ± 0.03 |
| | | C-VAE-UNET ($R=2$) | 0.04 ± 0.01 | 0.07 ± 0.02 | 0.51 ± 0.04 |
| | | DAPSM | 0.20 ± 0.01 | n/a | n/a |
| | | GCNN | 0.17 ± 0.06 | n/a | n/a |
| | | GMERROR | 0.05 ± 0.00 | n/a | n/a |
| | | DURBIN | 0.05 ± 0.00 | 0.05 ± 0.00 | n/a |
| | | S2SLS-LAG1 | 0.05 ± 0.00 | **0.04 ± 0.00** | n/a |
| | | SPATIAL+ | 0.12 ± 0.09 | n/a | n/a |
| | | SPATIAL | **0.03 ± 0.00** | n/a | n/a |
| | | UNET | 0.06 ± 0.01 | 0.17 ± 0.04 | n/a |
| | | E-MAP-SPATIAL+ ($R=1$) | 0.11 ± 0.09 | 0.06 ± 0.06 | n/a |
| | | E-MAP-SPATIAL ($R=1$) | **0.03 ± 0.00** | 0.07 ± 0.06 | n/a |
| $SO_4$ $\downarrow$ $PM_{2.5}$ ($r_d=1$) | $OC$ | C-VAE-UNET ($R=1$) | 0.09 ± 0.00 | 0.12 ± 0.02 | 0.48 ± 0.05 |
| | | C-VAE-UNET ($R=2$) | **0.06 ± 0.01** | 0.07 ± 0.03 | 0.54 ± 0.03 |
| | | DAPSM | 1.57 ± 0.00 | n/a | n/a |
| | | GCNN | 0.42 ± 0.15 | n/a | n/a |
| | | GMERROR | 0.13 ± 0.00 | n/a | n/a |
| | | DURBIN | 0.13 ± 0.00 | **0.01 ± 0.00** | n/a |
| | | S2SLS-LAG1 | 0.13 ± 0.00 | 0.08 ± 0.00 | n/a |
| | | SPATIAL+ | 0.12 ± 0.08 | n/a | n/a |
| | | SPATIAL | 0.07 ± 0.00 | n/a | n/a |
| | | UNET | 0.07 ± 0.02 | 0.05 ± 0.02 | n/a |
| | | E-MAP-SPATIAL+ ($R=1$) | 0.10 ± 0.07 | 0.05 ± 0.01 | n/a |
| | | E-MAP-SPATIAL ($R=1$) | 0.07 ± 0.00 | 0.05 ± 0.01 | n/a |

confounders and residuals fixed. To simulate hidden confounding, we identify influential covariates by measuring the change in predictive performance when each is removed, then mask the most important ones at training and testing.

**Raw datasets.** From the full SpaCE suite, we focus in the main text on two collections:

*Air Pollution and Mortality:* County-level data for the mainland US in 2010, including elderly mortality (CDC), fine particulate matter ($PM_{2.5}$) exposure (Di et al., 2019), behavioral risk factors (BRFSS) (Centers for Disease Control and Prevention, 2010), and Census demographics (U.S. Census Bureau, 2010). We study the effect of $PM_{2.5}$ exposure (treatment) on mortality ($PM_{2.5} \rightarrow m$), with different masked confounders.

*$PM_{2.5}$ Components:* High-resolution ($1 \times 1$ km) gridded data on total $PM_{2.5}$ (Di et al., 2019) and its chemical composition (Amini et al., 2022), using annual averages for 2000. We focus on the effect of sulfate on overall $PM_{2.5}$ ($SO_4 \rightarrow PM_{2.5}$), with key latent drivers such as *ammonium* ($NH_4$) and *organic carbon* (OC) masked.

The datasets are complementary: the first captures socioeconomic and demographic confounding, while the second reflects atmospheric chemistry. Additional datasets and hidden-confounder variants are described in Appendix D.

**Baselines and model variants.** We benchmark against classical and modern spatial methods: S2SLS-LAG1 (Anselin, 1988), a spatial-lag model with a spatially lagged outcome; DURBIN (Anselin, 1988), which additionally includes spatially lagged covariates; GMERROR (Anselin, 1988), a generalized moments estimator for spatial error models; spline-based SPATIAL and residualized SPATIAL+ (Dupont et al., 2022); exposure-mapped variants E-MAP-SPATIAL and E-MAP-SPATIAL+, which augment the corresponding models with the mean neighborhood treatment exposure

$|\mathcal{N}_s|^{-1} \sum_{j \in \mathcal{N}_s} A_j$; GCNN (Kipf, 2016) for non-linear neighbor aggregation; DAPSM (Papadogeorgou et al., 2019) for proximity-based matching; and UNET (Ronneberger et al., 2015), which can capture spillovers via neighbor treatments but does not adjust for hidden confounding.

For the *Spatial Deconfounder*, we instantiate the potential outcome module differently by setting the head to SPATIAL+ under local confounding (to ensure fairness) and to UNET under spatial confounding (to flexibly capture multi-scale structure). We also vary the neighborhood radius $R \in \{1, 2\}$ considered by the model and the latent confounder dimension in the C-VAE ($d_Z \in \{1, 2, 4, 8, 16, 32\}$).

**Evaluation metrics.** We assess performance on the direct (DIR) and spillover (SPILL) effects. As standard in causal inference (Hill, 2011; Shi et al., 2019; Cheng et al., 2022), we report standardized absolute bias, $\sigma_y^{-1} |\hat{\tau} - \tau|$, with true effect $\tau$, estimate $\hat{\tau}$, and outcome standard deviation $\sigma_y$.

**Results.** Tables 1 and 2 report performance under local and spatial confounding across multiple environments, masked confounders, and interference radii. Overall, the Spatial Deconfounder variants achieve strong performance on both direct and spillover effects, often matching or improving over the strongest non-oracle baselines. In local-confounding settings, C-VAE-SPATIAL+ substantially improves over matching, graph, and spatial autoregressive baselines on direct effects while also estimating spillovers; this remains true even for less smooth hidden confounders such as population density ($\rho_{\text{pop}}$). In spatial-confounding settings, C-VAE-UNET generally improves over UNET, indicating that substitute-confounder reconstruction helps even when the outcome head captures spatial dependence. Spillover-aware baselines such as E-MAP, S2SLS-LAG1, and DURBIN can be competitive, especially when the spillover structure is simple; however, they do not reconstruct latent confounding and can degrade when hidden spatial structure is strong.

**Remark 3** (Generality of the semi-synthetic benchmark). *Although our theory relies on idealized assumptions, the semi-synthetic benchmark does not enforce them by construction. Outcomes are generated by fitting a flexible function f to real observational data and masking influential covariates that also drive treatment. In particular, we do not enforce fixed spillover strength, smooth masked confounders (e.g., $\rho_{pop}$), or separability of the outcome model. This yields smooth and non-smooth confounders, stronger and weaker confounding regimes, and heterogeneous interference strength. This explains why no single method dominates everywhere: our goal is not to claim uniform dominance, but to show that interference-driven deconfounding remains robust and often reduces bias across diverse spatial data-generating processes.*

**Additional experiments.** Additional semi-synthetic results in Appendix D show similar trends across broader datasets, confounders, and radii. When our method underperforms, the settings typically involve weak confounding, very smooth confounding, or simple spillover structure, where stronger parametric or exposure-mapping assumptions can be advantageous. We also include a real-world Arctic sea-ice case study in Appendix E: using Pan-Arctic data from Ali et al. (2024), we estimate the effect of downward longwave radiation (LWDN) in the Laptev and East Siberian seas on annual and summer sea ice concentration. The Spatial Deconfounder recovers the physically expected direction and seasonality of the response, consistent with controlled climate-model evidence (Kapsch et al., 2016). Overall, these results support the premise that interference provides useful signal, not merely nuisance variation, for causal inference under unobserved spatial confounding.

**Stress tests.** We run targeted stress tests to characterize failure modes; see Appendix E for details. Sparsity sweeps from 10% to 70% show that performance degrades as overlap and the neighborhood treatment signal weaken, with predictive $p$-values moving away from 0.5. Single-cause confounder tests violate Assumption 4 by injecting localized unobserved confounders; as expected, bias increases and non-deconfounding baselines become competitive. Radius and latent-dimension sweeps show stability to moderate misspecification. We also test asymmetric interference topology and confounder-modulated spillovers; the former shows robustness to topology misspecification, while the latter is adversarial and degrades all methods. Overall, the method is strongest when hidden confounding is spatially shared, and weakens under highly localized confounding, poor support, or severe outcome-structure violations.

## 7. Conclusion

We introduce the **Spatial Deconfounder**, a framework that jointly addresses interference and unobserved spatial confounding by treating neighborhood treatments as a multi-cause signal. A C-VAE with a spatial prior reconstructs a substitute confounder, enabling estimation of direct and spillover effects with flexible outcome models. We prove identification under assumptions on the latent spatial field and outcome structure.

More broadly, our results suggest a shift in perspective: rather than treating interference solely as a nuisance, it can provide signal about hidden structure. While our guarantees rely on idealized assumptions, our semi-synthetic experiments on minimally modified environmental-health data show consistent bias reductions relative to strong spatial and deep-learning baselines, supporting the practical value of interference-driven multi-cause representations. Future work includes richer uncertainty quantification, comparison to Bayesian models, and extensions to spatiotemporal settings and continuous treatments.

## Acknowledgments

Ayush Khot was supported by the U.S. National Science Foundation Graduate Research Fellowship Program (NSF GRFP). Miruna Oprescu, Ai Kagawa, and Xihaier Luo were supported by the U.S. Department of Energy, Office of Science, Office of Advanced Scientific Computing Research (ASCR), under awards DE-SC0023112, KJ0402020/CC124, and KJ0401010/CC147, respectively. Any opinions, findings, and conclusions or recommendations expressed in this material are those of the author(s) and do not necessarily reflect the views of the National Science Foundation or the U.S. Department of Energy, Office of Science.

We are grateful to the ICML reviewers and area chairs for their thoughtful, constructive feedback. Their comments pushed us to strengthen the theory, expand the empirical evaluation, clarify the positioning, and better articulate the limitations of the framework. The final version is substantially better because of their input.

## Impact Statement

This work contributes to machine learning and causal inference by introducing a framework for more reliable effect estimation in spatial domains. Applications include environmental health, climate science, and social sciences, where accurate causal estimates can inform policy decisions. At the same time, we caution against uncritical use in high-stakes settings: violations of assumptions or biases in observational data may yield misleading conclusions. We encourage responsible deployment—especially in contexts affecting vulnerable populations—and recommend pairing our method with domain expertise, sensitivity analyses, and uncertainty quantification.

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

# A. Extended Literature Review

The **Spatial Deconfounder** draws on three strands of prior work: (i) spatial causal inference under interference and spatially structured confounding, (ii) deconfounding methods for ATE estimation with unobserved confounders, and (iii) deep learning for spatial and latent structure modeling. We detail each in the sections that follow.

## A.1. Spatial Causal Inference Under Interference and Spatially Structured Confounding

**Classical spatial causal inference.** Most estimators of direct and spillover effects assume that bias can be removed by conditioning on *observed* covariates, often together with a specified exposure mapping or interference structure. Design-based work—grounded in exposure mappings, partial-interference designs, and randomization inference—derives estimators or hypothesis tests under known neighborhood or network structure (E-MAP, e.g., Hudgens & Halloran, 2008; Sobel, 2006; Aronow & Samii, 2017; Forastiere et al., 2021; Tchetgen Tchetgen et al., 2021). Model-based strategies then adjust for that structure while still relying on measured covariates or correct functional form: spatial econometric models capture dependence through spatial lags, autoregressive structure, or spatially lagged covariates (e.g., Anselin, 1988) (S2SLS-LAG1, S2SLS-DURBIN), while spline/GAM and restricted-spatial-regression approaches adjust for residual spatial trends (e.g., Hanks et al., 2015) (SPATIAL). Deep graph/convolutional architectures can pool information across nearby units to improve prediction or imputation, but by themselves do not furnish identification without additional causal assumptions (Kipf, 2016). Domain-specific simulators (e.g., wildfire spread or atmospheric transport) encode spatial dependence through process-based physics and are often used as inputs to causal analyses, yet they typically still condition on observed drivers or require design-identifying assumptions (e.g. Larsen et al., 2022; Zigler et al., 2025). All of the above *presume exchangeability given observed covariates (or a valid design)*; if important spatial determinants of treatment and outcome are unmeasured, residual confounding bias can remain.

**Spatial confounding and bias-adjustment methods.** A growing literature tackles *unmeasured* spatial confounding directly. One family augments outcome models with latent spatial random effects (e.g., BYM/ICAR or GMRF priors) to absorb smooth hidden structure; this can reduce bias when the confounder is well captured by the basis, but may leave bias or distort fixed effects under misspecification (Rue & Held, 2005; Hodges & Reich, 2010). Restricted spatial regression and related orthogonalization schemes constrain the latent field away from covariates to mitigate bias (Hanks et al., 2015). Building on this idea, Dupont et al. (2022) (SPATIAL+) explicitly orthogonalizes spatial structure in the covariates from the outcome trend to purge bias from unmeasured *spatial* confounding. Propensity-score strategies that incorporate spatial proximity—such as distance-adjusted propensity score matching (Papadogeorgou et al., 2019) (DAPSM)—aim to proxy smooth unmeasured confounders via geography. Instrumental-variable designs exploit exogenous spatial variation, such as wind direction, policy boundaries, or thermal inversions, to identify causal effects despite hidden confounding, but require strong relevance and exclusion conditions that are difficult to validate under interference (e.g., Angrist et al., 1996; Imbens & Rubin, 2015; Deryugina et al., 2019; Woodward et al., 2024). Finally, Bayesian frameworks that jointly model interference and latent spatial fields (e.g., Papadogeorgou & Samanta, 2023) achieve identification under specified priors and structural assumptions. Recent work further shows that bias depends on the relative spatial scales of covariates and latent confounders, and that smoothing or orthogonalization succeeds only when observed covariates carry sufficient non-spatial or finer-scale variation (Paciorek, 2010; Dupont et al., 2023; Pim et al., 2026). In short, existing approaches exploit smoothness, fine-scale covariate variation, IV-style exogenous variation, or strong priors. None exploit interference patterns as a *signal* for nonparametrically reconstructing latent confounding from the treatment process—a gap our Spatial Deconfounder addresses.

## A.2. Deconfounding Methods for ATE Estimation with Unobserved Confounders

When confounders are unmeasured, point identification of causal effects generally fails. One approach is to derive bounds through sensitivity analysis (e.g., VanderWeele et al., 2015; Dorn et al., 2025; Oprescu et al., 2023; Frauen et al., 2023), trading identifiability for robustness. Another is the *deconfounder* framework, which fits a factor model to multiple causes in order to infer a substitute for the latent confounder, thereby restoring point identification (Wang & Blei, 2019; Bica et al., 2020; Hatt & Feuerriegel, 2024). This stream is closest in spirit to our work: like us, it leverages multiplicity of treatments as a proxy for hidden structure. However, existing deconfounder methods require datasets with many simultaneous treatments (e.g., recommender systems, panel data) and assume no interference. Our approach resolves both limitations: interference itself naturally generates multiple-cause treatment vectors, enabling latent field recovery even with a single treatment type.

### A.3. Deep Learning for Spatial and Latent Structure Modeling

**Deep learning for spatial modeling.** Modern deep architectures capture rich spatial structure but, on their own, remain predictive rather than identifying. U-Nets and encoder–decoder variants model multi-scale patterns on grids (Ronneberger et al., 2015; Oktay et al., 2018) (UNET); graph neural networks extend to irregular domains (Kipf, 2016; Hamilton et al., 2017; Veličković et al., 2017) (GCNN); and patch-wise transformers model long-range dependencies on images and geospatial rasters (Dosovitskiy et al., 2020; Liu et al., 2021). Spatiotemporal extensions (e.g., ConvLSTM and graph/vision transformers) further capture dynamics (Shi et al., 2015). These tools provide flexible representations but require additional causal structure for identification.

**Deep latent-variable models.** Finally, conditional variational autoencoders (C-VAEs) and related deep generative models are widely used for representation learning with latent factors (Kingma & Welling, 2013; Sohn et al., 2015). Beyond C-VAEs, the broader family of latent-variable models includes variational autoencoders with structured priors (Rezende et al., 2014; Maaløe et al., 2016), disentangled representation learning (Higgins et al., 2017), normalizing flows (Rezende & Mohamed, 2015), and diffusion-based generative models (Ho et al., 2020; Kingma et al., 2021), all of which offer flexible ways to recover hidden structure from high-dimensional data. While these methods are not causal in themselves, they provide natural tools for reconstructing latent processes from observed multi-cause data. In our framework, a C-VAE combined with a spatial prior enables smooth, nonparametric recovery of a substitute confounder from local treatment vectors, which is then used for causal identification. Other architectures (e.g., diffusion models or flow-based methods) could, in principle, be substituted, but the key contribution lies in adapting deep latent-factor reconstruction to the spatial interference setting, where treatments on neighboring units jointly reveal the latent field.

### A.4. Causal Generative Models

Recent work has proposed using expressive generative models as parameterizations of structural causal models. One stream of work uses autoregressive flows to obtain identifiable SCMs given a causal ordering (e.g., Javaloy et al., 2023; Khemakhem et al., 2021). Others combine diffusion- or GAN-based models with structural equations to model complex, high-dimensional counterfactuals (Sanchez & Tsaftaris, 2022; Kocaoglu et al., 2017). However, all of the methods assume unconfoundedness and are thus orthogonal to our Spatial Deconfounder. A different stream of literature combines causal inference and generative modeling under hidden confounding (e.g., Xia et al., 2021; Almodóvar et al., 2025). Similar to our work, the recently proposed DeCaFlow (Almodóvar et al., 2025) extends this line by learning confounded SCMs with causal normalizing flows and variational inference based on the deconfounder framework. However, these works are restricted to specific variables types, e.g., continuous treatments, and do not apply to the spatial setting. Building upon proxy variables, follow-up work on the deconfounder clarifies identifiability conditions in multi-cause settings (Wang & Blei, 2021). Similarly, this work assumes multiple treatments in an independent setting and does not apply to spatial causal inference tasks.

### A.5. Deep Identifiable Models and Network Deconfounding

A complementary line of work focuses on identifiability in deep latent variable models. Sparse deep generative models establish identifiability of VAEs under sparsity constraints (Moran et al., 2022), while Intact-VAE (Wu & Fukumizu, 2021) and $\beta$-Intact-VAE (Wu & Fukumizu, 2022) provide identifiable generative models for causal inference under unobserved confounding, IVs, proxies, and networked confounding. Applications to medical data show how identifiable VAEs can recover meaningful latent prognostic factors (Ma et al., 2023). These methods are typically designed for i.i.d. or network-structured observations and often rely on known adjacency structure, e.g., using neighbor information to help identify latent confounders in network deconfounding tasks. Our Spatial Deconfounder differs by targeting a specific spatial setting with localized grid-interference. More importantly, we note that our Spatial Deconfounder is not limited to the use of a C-VAE. The framework is model-agnostic and can be combined with other generative factor models. In contrast to these identifiable deep models, our focus is on a spatial–interference design: we show that interference-generated multi-cause vectors $(A_s, A_{\mathcal{N}_s})$, together with a spatial prior on $Z$, are sufficient to identify both direct and spillover effects without specifying a parametric latent-field model.

### A.6. Our Work

Our contribution lies at the intersection of spatial causal inference, deconfounding under unobserved confounding, and modern deep latent-variable modeling. Existing spatial-interference methods typically assume that all relevant confounders

are observed. Conversely, methods for unmeasured spatial confounding mitigate bias through explicit field models, fine-scale covariate variation, IV-style exogenous variation, or strong priors. In parallel, the *deconfounder* framework shows that multiplicity of causes can reveal substitutes for unobserved confounders, but is designed for i.i.d. settings with many simultaneous treatments and does not naturally extend to spatial domains where interference and locality are intrinsic.

The *Spatial Deconfounder* closes this gap by treating localized interference itself as the source of multi-cause information: a unit's own treatment and neighboring treatments provide spatially indexed views of a shared latent field. By training a C-VAE with a spatial prior, we nonparametrically reconstruct latent spatial confounding from these local treatment vectors. The resulting substitute confounder enables an interference-driven deconfounding strategy for identifying direct and spillover effects without specifying a latent-field model or requiring multiple treatment types.

# B. Proofs and Additional Results

We first provide background by stating supporting definitions and lemmas. Then we prove our main theorem on the identifiability of the treatment effects.

## B.1. Supporting Lemmas and definitions

**Definition 1** (Ignorability). *The grid treatment $(a_s, \mathbf{a}_{\mathcal{N}_s})$ is* ignorable *given $Z_s, \mathbf{X}_s, \mathbf{X}_{\mathcal{N}_s}$, if for all $s = 1, \ldots, n$ and for all $(a, \mathbf{a}_{\mathcal{N}}) \in \mathcal{A}^{|\mathcal{S}|}$*

$$(A_s, \mathbf{A}_{\mathcal{N}_s}) \perp\!\!\!\perp Y_s(a, \mathbf{a}_{\mathcal{N}}) \mid Z_s, \mathbf{X}_s, \mathbf{X}_{\mathcal{N}_s}. \tag{20}$$

**Definition 2** (Factor models). *A factor model of the assigned spatial treatments is a latent-variable model*

$$p_\phi(z_{1:|\mathcal{S}|}, \boldsymbol{x}_{1:|\mathcal{S}|}, \boldsymbol{x}_{\mathcal{N}_{1:|\mathcal{S}|}}, a_{1:|\mathcal{S}|}, \mathbf{a}_{\mathcal{N}_{1:|\mathcal{S}|}}) \tag{21}$$

$$= p(z_{1:|\mathcal{S}|}, \boldsymbol{x}_{1:|\mathcal{S}|}, \boldsymbol{x}_{\mathcal{N}_{1:|\mathcal{S}|}}) \prod_{s=1}^{|\mathcal{S}|} p_\phi(a_s \mid z_s, \boldsymbol{x}_s, \boldsymbol{x}_{\mathcal{N}_s}) \prod_{k \in \mathcal{N}_s} p_\phi(a_k \mid z_s, \boldsymbol{x}_s, \boldsymbol{x}_{\mathcal{N}_s}) \tag{22}$$

*rendering the assigned treatments conditionally independent.*

**Lemma 1.** *For the relation between the substitute confounder and factor models, it holds under weak regularity conditions*

1. *Assume the true distributions of the treatments $p(a_{1:|\mathcal{S}|}, \mathbf{a}_{\mathcal{N}_{1:|\mathcal{S}|}})$ can be represented by a factor model employing the substitute confounder $Z$, i.e., $p_\phi(z_{1:|\mathcal{S}|}, \boldsymbol{x}_{1:|\mathcal{S}|}, \boldsymbol{x}_{\mathcal{N}_{1:|\mathcal{S}|}}, a_{1:|\mathcal{S}|}, \mathbf{a}_{\mathcal{N}_{1:|\mathcal{S}|}})$. With the assumption of latent field sufficiency (see Assumption 4), the assigned treatments $(a, \mathbf{a}_{\mathcal{N}})$ are ignorable given $Z_s, \mathbf{X}_s,$ and $\mathbf{X}_{\mathcal{N}_s}$, i.e.,*

$$(A_s, \mathbf{A}_{\mathcal{N}_s}) \perp\!\!\!\perp Y_s(a, \mathbf{a}_{\mathcal{N}}) \mid Z_s, \mathbf{X}_s, \mathbf{X}_{\mathcal{N}_s}. \tag{23}$$

2. *A factor model that represents the distribution of the assigned treatments always exists.*

*Proof.* The statement follows from Proposition 5 in (Wang & Blei, 2019). $\qquad\square$

## B.2. Proof of the main theorem

**Theorem 1** (Causal identifiability). *Suppose Assumptions 1–5 hold. Let $Z_s$ be a piecewise constant function of the assigned neighborhood exposure and covariates $(a, \mathbf{a}_{\mathcal{N}}, \boldsymbol{x}, \boldsymbol{x}_{\mathcal{N}})$ and let the outcome be a separable function of the observed and unobserved variables:*

$$\mathbb{E}_Y\big[Y_s(a, \mathbf{a}_{\mathcal{N}}) \mid \mathbf{X}_s = \boldsymbol{x}, \mathbf{X}_{\mathcal{N}_s} = \boldsymbol{x}_{\mathcal{N}}, Z_s = z\big]$$
$$= f_1(a, \mathbf{a}_{\mathcal{N}}, \boldsymbol{x}, \boldsymbol{x}_{\mathcal{N}}) + f_2(z), \tag{14}$$

$$\mathbb{E}_Y\big[Y_s \mid A_s = a, \mathbf{A}_{\mathcal{N}_s} = \mathbf{a}_{\mathcal{N}}, \mathbf{X}_s = \boldsymbol{x}, \mathbf{X}_{\mathcal{N}_s} = \boldsymbol{x}_{\mathcal{N}}, Z_s = z\big]$$
$$= f_3(a, \mathbf{a}_{\mathcal{N}}, \boldsymbol{x}, \boldsymbol{x}_{\mathcal{N}}) + f_4(z), \tag{15}$$

*for continuously differentiable functions $f_1, f_2, f_3, f_4$. Consequently, the direct and spillover effects are identifiable as*

$$\tau_{\text{dir}} = \mathbb{E}_{\mathbf{X}_s, \boldsymbol{x}_{\mathcal{N}_s}, Z}\Big[\mathbb{E}_Y\big[Y_s \mid A_s = 1, \mathbf{A}_{\mathcal{N}_s}, \mathbf{X}_s, \mathbf{X}_{\mathcal{N}_s}, Z_s\big]$$

$$- \mathbb{E}_Y\big[Y_s \mid A_s = 0, \mathbf{A}_{\mathcal{N}_s}, \mathbf{X}_s, \mathbf{X}_{\mathcal{N}_s}, Z_s\big]\Big], \tag{16}$$

$$\tau_{\text{spill}} = \mathbb{E}_{\mathbf{X}_s, \boldsymbol{x}_{\mathcal{N}_s}, Z}\Big[\mathbb{E}_Y\big[Y_s \mid a, \mathbf{A}_{\mathcal{N}_s} = \mathbf{a}_{\mathcal{N}_s}^{(1)}, \mathbf{X}_s, \mathbf{X}_{\mathcal{N}_s}, Z_s\big]$$

$$- \mathbb{E}_Y\big[Y_s \mid a, \mathbf{A}_{\mathcal{N}_s} = \mathbf{a}_{\mathcal{N}_s}^{(0)}, \mathbf{X}_s, \mathbf{X}_{\mathcal{N}_s}, Z_s\big]\Big]. \tag{17}$$

*Proof.* First, observe that by the power-property and the separability of the outcome, we have

$$\mathbb{E}_Y[Y_s(a, \mathbf{a}_{\mathcal{N}})] = \mathbb{E}_{\mathbf{X}, \mathbf{X}_{\mathcal{N}}, Z}\big[\mathbb{E}_Y[Y_s(a, \mathbf{a}_{\mathcal{N}}) \mid \mathbf{X}_s, \mathbf{X}_{\mathcal{N}_s}, Z_s]\big] \tag{24}$$

$$= \mathbb{E}_{\mathbf{X}, \mathbf{X}_{\mathcal{N}}}[f_1(a, \mathbf{a}_{\mathcal{N}}, \mathbf{X}_s, \mathbf{X}_{\mathcal{N}_s})] + \mathbb{E}_Z[f_2(Z_s)]. \tag{25}$$

For the direct and indirect effects $\tau_{dir}$ and $\tau_{ind}$ follows

$$\tau_{dir} = \mathbb{E}_{\mathbf{X},\mathbf{x}_{\mathcal{N}}}[f_1(A_s = 1, \mathbf{a}_{\mathcal{N}_s}, \mathbf{X}_s, \mathbf{X}_{\mathcal{N}_s})] - \mathbb{E}_{\mathbf{X},\mathbf{x}_{\mathcal{N}}}[f_1(A_s = 0, \mathbf{a}_{\mathcal{N}_s}, \mathbf{X}_s, \mathbf{X}_{\mathcal{N}_s})] \tag{26}$$

$$= \int_{C(1,0)} \nabla_{\nu}\mathbb{E}_{\mathbf{X},\mathbf{x}_{\mathcal{N}}}[f_1(\nu, \mathbf{a}_{\mathcal{N}}, \mathbf{X}_s, \mathbf{X}_{\mathcal{N}_s})]d\nu, \quad \nu \in \mathbb{R} \tag{27}$$

and

$$\tau_{ind} = \mathbb{E}_{\mathbf{X},\mathbf{x}_{\mathcal{N}}}[f_1(a_s, \mathbf{A}_{\mathcal{N}_s} = \mathbf{a}_{\mathcal{N}_s}^{(1)}, \mathbf{X}_s, \mathbf{X}_{\mathcal{N}_s})] - \mathbb{E}_{\mathbf{X},\mathbf{x}_{\mathcal{N}}}[f_1(a_s, \mathbf{A}_{\mathcal{N}_s} = \mathbf{a}_{\mathcal{N}_s}^{(0)}, \mathbf{X}_s, \mathbf{X}_{\mathcal{N}_s})] \tag{28}$$

$$= \int_{C(a_{\mathcal{N}_s}^{(1)}, a_{\mathcal{N}_s}^{(0)})} \nabla_{\kappa}\mathbb{E}_{\mathbf{X},\mathbf{x}_{\mathcal{N}}}[f_1(a_s, \mathbf{A}_{\mathcal{N}_s} = \kappa, \mathbf{X}_s, \mathbf{X}_{\mathcal{N}_s})]d\kappa, \quad \kappa \in \mathbb{R}^{|\mathcal{S}|-1}. \tag{29}$$

We thus need to find an expression for the gradient to rewrite the integral in terms of observable quantities.

To do so, we first consider the conditional expected outcome. By Assumption 5 there exists a function $g$ such that $Z = g(a, \mathbf{a}_{\mathcal{N}}, \mathbf{X}, \mathbf{X}_{\mathcal{N}})$. Therefore, it holds

$$\mathbb{E}_{\mathbf{X},\mathbf{X}_{\mathcal{N}},Z}\big[\mathbb{E}_Y[Y_s \mid A_s = a_s, \mathbf{A}_{\mathcal{N}_s} = \mathbf{a}_{\mathcal{N}_s}, \mathbf{X}_{\mathcal{N}_s}, Z_s]\big] \tag{30}$$

$$= \mathbb{E}_{\mathbf{X},\mathbf{X}_{\mathcal{N}}}\big[\mathbb{E}_Y[Y_s \mid A_s = a_s, \mathbf{A}_{\mathcal{N}_s} = \mathbf{a}_{\mathcal{N}_s}, \mathbf{X}_s, \mathbf{X}_{\mathcal{N}_s}, Z_s = g(a_s, \mathbf{a}_{\mathcal{N}_s}, \mathbf{X}_s, \mathbf{X}_{\mathcal{N}_s})]\big] \tag{31}$$

$$= \mathbb{E}_{\mathbf{X},\mathbf{X}_{\mathcal{N}}}\big[\mathbb{E}_Y[Y_s(a_s, \mathbf{a}_{\mathcal{N}_s}) \mid A_s = a_s, \mathbf{A}_{\mathcal{N}_s} = \mathbf{a}_{\mathcal{N}_s}, \mathbf{X}_s, \mathbf{X}_{\mathcal{N}_s}, Z_s = g(a_s, \mathbf{a}_{\mathcal{N}_s}, \mathbf{X}_s, \mathbf{X}_{\mathcal{N}_s})]\big], \tag{32}$$

where the latter equality follows from Assumption 1.

As $Y_s(a_s, \mathbf{a}_{\mathcal{N}_s}) \perp\!\!\!\perp A_s, \mathbf{A}_{\mathcal{N}_s} \mid \mathbf{X}_s, \mathbf{X}_{\mathcal{N}_s}, Z_s$ (by Lemma 1) and the outcomes are assumed to be separable, it follows

$$\mathbb{E}_{\mathbf{X},\mathbf{X}_{\mathcal{N}},Z}\big[\mathbb{E}_Y[Y_s \mid A_s = a_s, \mathbf{A}_{\mathcal{N}_s} = \mathbf{a}_{\mathcal{N}_s}, \mathbf{X}_s, \mathbf{X}_{\mathcal{N}_s}, Z_s]\big] \tag{33}$$

$$= \mathbb{E}_{\mathbf{X},\mathbf{X}_{\mathcal{N}}}\big[\mathbb{E}_Y[Y_s(a_s, \mathbf{a}_{\mathcal{N}_s}) \mid \mathbf{X}_s, \mathbf{X}_{\mathcal{N}_s}, Z_s = g(a_s, \mathbf{a}_{\mathcal{N}_s}, \mathbf{X}_s, \mathbf{X}_{\mathcal{N}_s})]\big] \tag{34}$$

$$= \mathbb{E}_{\mathbf{X},\mathbf{X}_{\mathcal{N}}}[f_1(a_s, \mathbf{a}_{\mathcal{N}_s}, \mathbf{X}_s, \mathbf{X}_{\mathcal{N}_s})] + \mathbb{E}_Z[f_2(g(a_s, \mathbf{a}_{\mathcal{N}_s}, \mathbf{X}_s, \mathbf{X}_{\mathcal{N}_s}))]. \tag{35}$$

Recall that by the definition of the conditional expected outcome, we have

$$\mathbb{E}_{\mathbf{X},\mathbf{X}_{\mathcal{N}},Z}\big[\mathbb{E}_Y[Y_s \mid A_s = a_s, \mathbf{A}_{\mathcal{N}_s} = \mathbf{a}_{\mathcal{N}_s}, \mathbf{X}_s, \mathbf{X}_{\mathcal{N}_s}, Z_s]\big] = \tag{36}$$

$$\mathbb{E}_{\mathbf{X},\mathbf{X}_{\mathcal{N}}}[f_3(a_s, \mathbf{a}_{\mathcal{N}_s}, \mathbf{X}_s, \mathbf{X}_{\mathcal{N}_s})] + \mathbb{E}_Z[f_4(g(a_s, \mathbf{a}_{\mathcal{N}_s}, \mathbf{X}_s, \mathbf{X}_{\mathcal{N}_s}))]. \tag{37}$$

Now, we are ready to consider the gradients in 28. Observe that for the gradients of the conditional outcome, it holds

$$\nabla_{a_s}\mathbb{E}_{\mathbf{X},\mathbf{X}_{\mathcal{N}},Z}\big[\mathbb{E}_Y[Y_s \mid a_s, \mathbf{A}_{\mathcal{N}_s} = \mathbf{a}_{\mathcal{N}_s}, \mathbf{X}_s, \mathbf{X}_{\mathcal{N}_s}, Z_s]\big] \tag{38}$$

$$= \nabla_{a_s}\mathbb{E}_{\mathbf{X},\mathbf{X}_{\mathcal{N}}}[f_1(a_s, \mathbf{a}_{\mathcal{N}_s}, \mathbf{X}_s, \mathbf{X}_{\mathcal{N}_s})] + \nabla_{a_s}\mathbb{E}_Z[f_2(g(a_s, \mathbf{a}_{\mathcal{N}_s}))] \tag{39}$$

$$= \nabla_{a_s}\mathbb{E}_{\mathbf{X},\mathbf{X}_{\mathcal{N}}}[f_3(a_s, \mathbf{a}_{\mathcal{N}_s}, \mathbf{X}_s, \mathbf{X}_{\mathcal{N}_s})] + \nabla_{a_s}\mathbb{E}_Z[f_4(g(a_s, \mathbf{a}_{\mathcal{N}_s}))] \tag{40}$$

with a similar expression for $\nabla_{\mathbf{a}_{\mathcal{N}_s}}$. Note that, up to a set of Lebesgue measure zero, the gradients of $f_2$ and $f_4$ disappear, i.e.,

$$\nabla_{a_s}\mathbb{E}_Z[f_2(g(a_s, \mathbf{a}_{\mathcal{N}_s}, \mathbf{X}_s, \mathbf{X}_{\mathcal{N}_s}))] = \nabla_{g(a_s, \mathbf{a}_{\mathcal{N}_s}, \mathbf{X}_s, \mathbf{X}_{\mathcal{N}_s})}f_2\nabla_{a_s}g(a_s, \mathbf{a}_{\mathcal{N}_s}, \mathbf{X}_s, \mathbf{X}_{\mathcal{N}_s}) = 0 \tag{41}$$

and

$$\nabla_{a_s}\mathbb{E}_Z[f_4(g(a_s, \mathbf{a}_{\mathcal{N}_s}, \mathbf{X}_s, \mathbf{X}_{\mathcal{N}_s}))] = \nabla_{g(a_s, \mathbf{a}_{\mathcal{N}_s}, \mathbf{X}_s, \mathbf{X}_{\mathcal{N}_s})}f_4\nabla_{a_s}g(a_s, \mathbf{a}_{\mathcal{N}_s}, \mathbf{X}_s, \mathbf{X}_{\mathcal{N}_s}) = 0 \tag{42}$$

as

$$\nabla_{a_s}g(a_s, \mathbf{a}_{\mathcal{N}_s}, \mathbf{X}_s, \mathbf{X}_{\mathcal{N}_s}) = 0.$$

Similarly,

$$\nabla_{\mathbf{a}_{\mathcal{N}_s}}\mathbb{E}_Z[f_2(g(a_s, \mathbf{a}_{\mathcal{N}_s}, \mathbf{X}_s, \mathbf{X}_{\mathcal{N}_s}))] = \nabla_{\mathbf{a}_{\mathcal{N}_s}}\mathbb{E}_Z[f_4(g(a_s, \mathbf{a}_{\mathcal{N}_s}, \mathbf{X}_s, \mathbf{X}_{\mathcal{N}_s}))] = 0.$$

Overall, we receive

$$\nabla_{a_s}\mathbb{E}_{\mathbf{X},\mathbf{X}_{\mathcal{N}}}[f_1(a_s, \mathbf{a}_{\mathcal{N}_s}, \mathbf{X}_s, \mathbf{X}_{\mathcal{N}_s})] = \nabla_{a_s}\mathbb{E}_{\mathbf{X},\mathbf{X}_{\mathcal{N}}}[f_3(a_s, \mathbf{a}_{\mathcal{N}_s}, \mathbf{X}_s, \mathbf{X}_{\mathcal{N}_s})] \tag{43}$$

and

$$\nabla_{\mathbf{a}_{\mathcal{N}_s}} \mathbb{E}_{\mathbf{X}, \mathbf{X}_{\mathcal{N}}}[f_1(a_s, \mathbf{a}_{\mathcal{N}_s}, \mathbf{X}_s, \mathbf{X}_{\mathcal{N}_s})] = \nabla_{\mathbf{a}_{\mathcal{N}_s}} \mathbb{E}_{\mathbf{X}, \mathbf{X}_{\mathcal{N}}}[f_3(a_s, \mathbf{a}_{\mathcal{N}_s}, \mathbf{X}_s, \mathbf{X}_{\mathcal{N}_s})]. \tag{44}$$

Finally, we can identify the direct treatment $\tau_{dir}$ effect as

$$\tau_{dir} = \int_{C(1,0)} \nabla_\nu \mathbb{E}_{\mathbf{X}, \mathbf{X}_{\mathcal{N}}}[f_1(\nu, \mathbf{a}_{\mathcal{N}}, \mathbf{X}_s, \mathbf{X}_{\mathcal{N}_s})]d\nu, \quad \nu \in \mathbb{R} \tag{45}$$

$$= \int_{C(1,0)} \nabla_\nu \mathbb{E}_{\mathbf{X}, \mathbf{X}_{\mathcal{N}}}[f_3(\nu, \mathbf{a}_{\mathcal{N}}, \mathbf{X}_s, \mathbf{X}_{\mathcal{N}_s})]d\nu, \quad \nu \in \mathbb{R} \tag{46}$$

$$= \mathbb{E}_{\mathbf{X}, \mathbf{X}_{\mathcal{N}}}[f_3(A_s = 1, \mathbf{a}_{\mathcal{N}}, \mathbf{X}_s, \mathbf{X}_{\mathcal{N}_s})] - \mathbb{E}_{\mathbf{X}, \mathbf{X}_{\mathcal{N}}}[f_3(A_s = 0, \mathbf{a}_{\mathcal{N}}, \mathbf{X}_s, \mathbf{X}_{\mathcal{N}_s})] \tag{47}$$

$$= \mathbb{E}_{\mathbf{X}, \mathbf{X}_{\mathcal{N}}}[f_3(A_s = 1, \mathbf{a}_{\mathcal{N}}, \mathbf{X}_s, \mathbf{X}_{\mathcal{N}_s})] + \mathbb{E}_Z[f_4(Z_s)] \tag{48}$$

$$- \mathbb{E}_{\mathbf{X}, \mathbf{X}_{\mathcal{N}}}[f_3(A_s = 0, \mathbf{a}_{\mathcal{N}}, \mathbf{X}_s, \mathbf{X}_{\mathcal{N}_s})] - \mathbb{E}_Z[f_4(Z_s)] \tag{49}$$

$$= \mathbb{E}_{Z, \mathbf{X}, \mathbf{X}_{\mathcal{N}}}\Big[\mathbb{E}_Y\big[Y_s \mid a_s{=}1, \mathbf{a}_{\mathcal{N}_s}, \mathbf{X}_s, \mathbf{X}_{\mathcal{N}_s}, Z_s\big] - \mathbb{E}_Y\big[Y_s \mid a_s{=}0, \mathbf{a}_{\mathcal{N}_s}, \mathbf{X}_s, \mathbf{X}_{\mathcal{N}_s}, Z_s\big]\Big] \tag{50}$$

and similarly the indirect treatment effect $\tau_{ind}$ as

$$\tau_{ind} = \int_{C(a_{\mathcal{N}_s}^{(1)}, a_{\mathcal{N}_s}^{(0)})} \nabla_\kappa \mathbb{E}_{\mathbf{X}, \mathbf{X}_{\mathcal{N}}}[f_1(a_s, \mathbf{A}_{\mathcal{N}_s} = \kappa, \mathbf{X}_s, \mathbf{X}_{\mathcal{N}_s})]d\kappa \tag{51}$$

$$= \int_{C(a_{\mathcal{N}_s}^{(1)}, a_{\mathcal{N}_s}^{(0)})} \nabla_\kappa \mathbb{E}_{\mathbf{X}, \mathbf{X}_{\mathcal{N}}}[f_3(a_s, \mathbf{A}_{\mathcal{N}_s} = \kappa, \mathbf{X}_s, \mathbf{X}_{\mathcal{N}_s})]d\kappa \tag{52}$$

$$= \mathbb{E}_{\mathbf{X}, \mathbf{X}_{\mathcal{N}}}[f_3(a_s, \mathbf{a}_{\mathcal{N}_s}^{(1)}, \mathbf{X}_s, \mathbf{X}_{\mathcal{N}_s})] - \mathbb{E}_{\mathbf{X}, \mathbf{X}_{\mathcal{N}}}[f_3(a_s, a_{\mathcal{N}_s}^{(0)}, \mathbf{X}_s, \mathbf{X}_{\mathcal{N}_s})] \tag{53}$$

$$= \mathbb{E}_{\mathbf{X}, \mathbf{X}_{\mathcal{N}}}[f_3(a_s, \mathbf{a}_{\mathcal{N}_s}^{(1)}, \mathbf{X}_s, \mathbf{X}_{\mathcal{N}_s})] + \mathbb{E}_Z[f_4(Z_s)] \tag{54}$$

$$- \mathbb{E}_{\mathbf{X}, \mathbf{X}_{\mathcal{N}}}[f_3(a_s, \mathbf{a}_{\mathcal{N}_s}^{(0)}, \mathbf{X}_s, \mathbf{X}_{\mathcal{N}_s})] - \mathbb{E}_Z[f_4(Z_s)] \tag{55}$$

$$= \mathbb{E}_{Z, \mathbf{X}, \mathbf{X}_{\mathcal{N}}}\Big[\mathbb{E}_Y\big[Y_s \mid a_s, \mathbf{a}_{\mathcal{N}_s}^{(1)}, \mathbf{X}_s, \mathbf{X}_{\mathcal{N}_s}, Z_s\big] - \mathbb{E}_Y\big[Y_s \mid a_s, \mathbf{a}_{\mathcal{N}_s}^{(0)}, \mathbf{X}_s, \mathbf{X}_{\mathcal{N}_s}, Z_s\big]\Big] \tag{56}$$

Overall, we proved that the substitute confounder generated by our spatial deconfounder renders the treatment effects identifiable. $\qquad\square$

### B.3. Sensitivity to proxy error

**Proposition 1** (Sensitivity of identified effects to proxy error). *Let*

$$m(a, \mathbf{a}_{\mathcal{N}}, x, x_{\mathcal{N}}, z) := \mathbb{E}[Y_s \mid A_s = a, \mathbf{A}_{\mathcal{N}_s} = \mathbf{a}_{\mathcal{N}}, \mathbf{X}_s = x, \mathbf{X}_{\mathcal{N}_s} = x_{\mathcal{N}}, Z_s^\star = z]$$

*denote the (oracle) outcome regression indexed by the population substitute confounder $Z_s^\star$. Assume that $m$ is L-Lipschitz in $z$, uniformly over $(a, \mathbf{a}_{\mathcal{N}}, x, x_{\mathcal{N}})$:*

$$\big|m(a, \mathbf{a}_{\mathcal{N}}, x, x_{\mathcal{N}}, z) - m(a, \mathbf{a}_{\mathcal{N}}, x, x_{\mathcal{N}}, z')\big| \leq L\|z - z'\| \qquad \forall z, z'.$$

*Define the identified direct and spillover effect functionals (cf. Equations (3) and (4)) evaluated at a generic proxy $W_s$ by*

$$\tau_{\mathrm{dir}}(W) := \mathbb{E}\Big[m\big(1, \mathbf{A}_{\mathcal{N}_s}, \mathbf{X}_s, \mathbf{X}_{\mathcal{N}_s}, W_s\big) - m\big(0, \mathbf{A}_{\mathcal{N}_s}, \mathbf{X}_s, \mathbf{X}_{\mathcal{N}_s}, W_s\big)\Big],$$

$$\tau_{\mathrm{spill}}(W) := \mathbb{E}\Big[m\big(a, \mathbf{a}_{\mathcal{N}_s}^{(1)}, \mathbf{X}_s, \mathbf{X}_{\mathcal{N}_s}, W_s\big) - m\big(a, \mathbf{a}_{\mathcal{N}_s}^{(0)}, \mathbf{X}_s, \mathbf{X}_{\mathcal{N}_s}, W_s\big)\Big].$$

*Then, for any proxy $\hat{Z}_s$,*

$$\big|\tau_{\mathrm{dir}}(\hat{Z}) - \tau_{\mathrm{dir}}(Z^\star)\big| \leq 2L\,\mathbb{E}\Big[\|\hat{Z}_s - Z_s^\star\|\Big], \qquad \big|\tau_{\mathrm{spill}}(\hat{Z}) - \tau_{\mathrm{spill}}(Z^\star)\big| \leq 2L\,\mathbb{E}\Big[\|\hat{Z}_s - Z_s^\star\|\Big].$$

*In particular, if $\mathbb{E}\|\hat{Z}_s - Z_s^\star\| \to 0$, then $\tau_{\mathrm{dir}}(\hat{Z}) \to \tau_{\mathrm{dir}}(Z^\star)$ and $\tau_{\mathrm{spill}}(\hat{Z}) \to \tau_{\mathrm{spill}}(Z^\star)$.*

*Proof.* For the direct effect,

$$
\begin{aligned}
&\left|\tau_{\mathrm{dir}}(\hat{Z}) - \tau_{\mathrm{dir}}(Z^\star)\right| \\
&= \Big| \mathbb{E}\Big[ m\big(1, \mathbf{A}_{\mathcal{N}_s}, \mathbf{X}_s, \mathbf{X}_{\mathcal{N}_s}, \hat{Z}_s\big) - m\big(0, \mathbf{A}_{\mathcal{N}_s}, \mathbf{X}_s, \mathbf{X}_{\mathcal{N}_s}, \hat{Z}_s\big) \\
&\qquad\quad - m\big(1, \mathbf{A}_{\mathcal{N}_s}, \mathbf{X}_s, \mathbf{X}_{\mathcal{N}_s}, Z^\star_s\big) + m\big(0, \mathbf{A}_{\mathcal{N}_s}, \mathbf{X}_s, \mathbf{X}_{\mathcal{N}_s}, Z^\star_s\big) \Big] \Big| \\
&\leq \mathbb{E}\Big[ \Big| m\big(1, \mathbf{A}_{\mathcal{N}_s}, \mathbf{X}_s, \mathbf{X}_{\mathcal{N}_s}, \hat{Z}_s\big) - m\big(1, \mathbf{A}_{\mathcal{N}_s}, \mathbf{X}_s, \mathbf{X}_{\mathcal{N}_s}, Z^\star_s\big) \Big| \Big] \\
&\quad + \mathbb{E}\Big[ \Big| m\big(0, \mathbf{A}_{\mathcal{N}_s}, \mathbf{X}_s, \mathbf{X}_{\mathcal{N}_s}, \hat{Z}_s\big) - m\big(0, \mathbf{A}_{\mathcal{N}_s}, \mathbf{X}_s, \mathbf{X}_{\mathcal{N}_s}, Z^\star_s\big) \Big| \Big] \\
&\leq L\, \mathbb{E}\Big[ \|\hat{Z}_s - Z^\star_s\| \Big] + L\, \mathbb{E}\Big[ \|\hat{Z}_s - Z^\star_s\| \Big] = 2L\, \mathbb{E}\Big[ \|\hat{Z}_s - Z^\star_s\| \Big],
\end{aligned}
$$

where the first inequality is the triangle inequality and the second uses the Lipschitz condition. The spillover bound is identical, replacing $(1, \mathbf{A}_{\mathcal{N}_s})$ and $(0, \mathbf{A}_{\mathcal{N}_s})$ by $(a, \mathbf{a}^{(1)}_{\mathcal{N}_s})$ and $(a, \mathbf{a}^{(0)}_{\mathcal{N}_s})$. $\qquad\square$

# C. Implementation Details

This section provides implementation details for our experimental setup. We cover four aspects:

1. **Semi-synthetic data generation:** construction of counterfactual outcomes under interference and spatial confounding using the `SpaCE` benchmark framework, with hidden confounders simulated by masking key covariates.

2. **Predictive model:** how the outcome model $f$ is estimated with ensembles of machine-learning models, including convolutional networks for spatial structure.

3. **Software and hyperparameters:** the AutoML framework used for training and tuning, along with default settings.

4. **Benchmarks:** implementation details for baseline methods.

**Semi-synthetic outcomes.** Recall from Section 6 that we construct counterfactual outcomes via

$$\hat{Y}_s = f(A_s, \mathbf{A}_{\mathcal{N}_s}, \mathbf{X}_s) + R_s \quad \text{or} \quad \hat{Y}_s = f(A_s, \mathbf{A}_{\mathcal{N}_s}, \mathbf{X}_s, \mathbf{X}_{\mathcal{N}_s}) + R_s,$$

where $f$ is a predictive model learned from real-world environmental data and $R_s$ are exogenous, spatially correlated residuals with the same distribution as the endogenous residuals.

**Predictive model with interference.** We estimate $f$ using ensembles of machine-learning models, with ensemble weights determined by predictive accuracy on held-out validation data. Following Tec et al. (2024) and the benchmarking guidelines of Curth et al. (2021), this avoids bias toward causal estimators tied to a single model class. To capture spatial structure, we include ResNet-18 (He et al., 2016) as one of the base learners. Training and hyperparameter tuning are automated with the `AutoGluon` Python package (Erickson et al., 2020), which performs model selection, hyperparameter search, and overfitting control with minimal human intervention. Default settings for `AutoGluon` are summarized in Table 3.

*Table 3.* Hyperparameters used in AutoML

| Parameter | Value |
| --- | --- |
| package | AutoGluon v1.4.0 |
| fit.presets | good_quality |
| fit.tuning_data | custom with Algorithm 2 |
| fit.use_bag_holdout | true |
| fit.time_limit | null |
| feature_importance.time_limit | 900 |
| hyperparameters | get_hyperparameter_config('multimodal') |
| hyperparameters.AG_AUTOMM.optim.max_epochs | 10 |
| hyperparameters.AG_AUTOMM.model.timm_image.checkpoint_name | resnet18 |

**Spatially-aware train-validation split.** We implement a *spatially-aware* train-validation data split (Roberts et al., 2017) that takes interference into account to avoid overfitting due to spatial correlations. We only consider nodes with complete neighborhoods for training and validation. This spatial splitting strategy identifies a limited number of validation nodes and applies breadth-first search to exclude their adjacent neighbors from the training dataset. For this study, we define each grid cell to have edges connecting it to its 8 surrounding cells. This algorithm is described in Algorithm 2.

**Synthetic Residual Generation.** Following the approach established in (Tec et al., 2024), we generate synthetic residuals using a Gaussian Markov Random Field (GMRF) from a spatial graph. Specifically, we sample the synthetic residuals according to: $\boldsymbol{R} \sim_{\text{iid}} \text{MultivariateNormal}(\mathbf{0}, \hat{\lambda}(\mathbb{D} - \hat{\rho}\mathbb{A}\mathbb{D})^{-1})$, where $\mathbb{A}$ represents the spatial graph's adjacency matrix, $\mathbb{D}$ denotes a diagonal matrix containing the degree (number of neighbors) for each spatial location, $\hat{\rho}$ parameterizes the spatial dependence between observations and their neighbors (estimated from the true residuals obtained from $f$), and $\hat{\lambda}$ is calibrated to preserve the exact variance of the observed residuals. We refer the reader to (Tec et al., 2024) for additional details.

**Benchmark Training and Hyperparameter Tuning.** To ensure a fair comparison, we use the RAY TUNE (Liaw et al., 2018) framework for hyperparameter tuning. For all but DAPSM, the tuning metric is implemented as mean-squared error (MSE) from a validation set obtained with the spatially-aware splitting method in Algorithm 2. We use this splitting algorithm for computing the tuning metric since random splitting would result in extreme overfitting (Roberts et al., 2017). For DAPSM we use the covariate balance criterion following Papadogeorgou et al. (2019). After selecting the best hyperparameters, the method is retrained on the full data. Table 4 summarizes our hyperparameter search space for different baseline models. For C-VAE models with radius R evaluated on a dataset of radius $r_d$, training and validation are restricted to nodes with radius $r_m = \max(r_d, \text{R})$. Each C-VAE model also specifies a latent confounder dimension $d_Z \in \{1, 2, 4, 8, 16, 32\}$. The licenses

---

**Algorithm 2** Spatially-aware validation split selection with radius and complete neighborhoods

---

**Input:** Graph as map of neighbors $s \to \mathbb{N}_s$ where $\mathbb{N}_s \subset \mathbb{S}$ is the set of neighbors of $s$.
**Params:** Fraction $\alpha$ of seed validation points (default $\alpha = 0.02$); number of BFS levels $L$ to include in the validation set (default $L = 1$); buffer size $B$ indicating the number of BFS levels to leave outside training and validation (default $B = 1$); radius $r_m$ of the model to consider when determining the split (default $r_m = 1$)
**Output:** Set of training nodes $\mathbb{T} \subset \mathbb{S}$ and validation nodes $\mathbb{V} \subset \mathbb{S}$.
 1: *# Helper function to check if node has complete r-hop neighborhood*
 2: **function** HASCOMPLETENEIGHBORHOOD($s, r$):
 3:    expected_count $= (2r + 1)^2$ *# For square grid*
 4:    actual_neighbors $=$ GetNeighborsWithinRadius($s, r$)
 5:    **return** |actual_neighbors| $=$ expected_count
 6: *# Filter to only nodes with complete neighborhoods*
 7: $\mathbb{S}_{valid} = \{s \in \mathbb{S} : \text{HASCOMPLETENEIGHBORHOOD}(s, r_m)\}$
 8: *# Initialize validation set with seed nodes from valid nodes only*
 9: $\mathbb{V} = \text{SampleWithoutReplacement}(\mathbb{S}_{valid}, \alpha)$
10: *# Expand validation set with neighbors*
11: **for** $\ell \in \{0, \dots, L - 1\}$ **do**
12:    tmp $= \mathbb{V}$
13:    **for** $s \in$ tmp **do**
14:      $\mathbb{V} = \mathbb{V} \cup \mathbb{N}_s$
15:    **end for**
16: **end for**
17: *# Compute buffer*
18: $\mathbb{B} = \mathbb{V}$
19: **for** $b \in \{0, \dots, B - 1 + r_m\}$ **do**
20:    tmp $= \mathbb{B}$
21:    **for** $s \in$ tmp **do**
22:      $\mathbb{B} = \mathbb{B} \cup \mathbb{N}_s$
23:    **end for**
24: **end for**
25: *# Exclude buffer for training set (from valid nodes only)*
26: $\mathbb{T} = \mathbb{S}_{valid} \setminus \mathbb{B}$
27: **return** $\mathbb{T}, \mathbb{V}$

---

| Model | Iterations | Tuning Metric | Value |
|---|---|---|---|
| C-VAE-SPATIAL+ | 100 | weight_decay_C-VAE | loguniform between 1e-4 and 1e-3 |
| | | beta_max ($\beta$) | loguniform between 1e-8 and 10 |
| | | lam_t $[PM_{2.5} \to m \ (r_d = 1)]$ | loguniform between 1e-3 and 1.0 |
| | | lam_t (other) | loguniform between 1e-5 and 1.0 |
| | | lam_y | loguniform between 1e-5 and 1.0 |
| C-VAE-UNET | 60 | weight_decay_C-VAE | loguniform between 1e-4 and 1e-3 |
| | | beta_max ($\beta$) | loguniform between 1e-3 and 1 |
| | | weight_decay_head | loguniform between 1e-4 and 1e-3 |
| | | unet_base_chan | 16 or 32 |
| DAPSM | N/A | propensity_score_penalty_value | choose from [0.001, 0.01, 0.1, 1.0] |
| | | propensity_score_penalty_type | l1 or l2 |
| | | spatial_weight | uniform between 0.0 and 1.0 |
| GCNN | N/A | hidden_dim | 16 or 32 |
| | | hidden_layers | 1 or 2 |
| | | weight_decay | loguniform between 1e-6 and 1e-1 |
| | | lr | 1e-3 or 3e-4 |
| | | epochs | 1000 or 2500 |
| | | dropout | loguniform between 1e-3 to 0.5 |
| SPATIAL+ | 2,500 | lam_t | loguniform between 1e-5 and 1.0 |
| | | lam_y | loguniform between 1e-5 and 1.0 |
| SPATIAL | 2,500 | lam | loguniform between 1e-5 and 1.0 |
| UNET | 50 | unet_base_chan | choose from [8, 16, 32] |

*Table 4.* Hyperparameter configurations evaluated for each model using a validation set. `Iterations` denotes the number of Ray Tune trials performed per model.

of the data sources used for training are summarized in the supplement of (Tec et al., 2024), which allow sharing and reuse for non-commercial purposes.

## D. Further Experimental Results

Our full experimental results are available for local confounding and spatial confounding at Table 5 and Table 6, respectively. There is a general pattern that C-VAE models tend to outperform benchmarks in estimating direct effects. In particular, C-VAE are the only local confounding methods that can also estimate spillover effects. In spatial confounding datasets with $r_d = 1$, deconfounders tend to have better direct effect and spillover estimation than UNET. We also plot the latent space of the C-VAE in Figure 4 to show that it recovers the large-scale spatial structure of the true confounder.

*Table 5.* Performance under *local confounding*. Results averaged over 10 runs with 95% confidence intervals. $r_d$: neighborhood radius in data generation; R: neighborhood radius used by the deconfounder. Lower values for ATE and SPILL indicate less bias. $p$ indicates the $p$-value of the predictive check, with values near 0.5 indicating good model fit to 0.5. Best and second-best values are bolded and underlined, respectively.

| Environment | Confounder | Method | DIR | SPILL | $p$ |
|---|---|---|---|---|---|
| $PM_{2.5} \rightarrow m \ (r_d = 1)$ | $\rho_{pop}$ | C-VAE-SPATIAL+ (R=1) | **0.02 ± 0.01** | **0.02 ± 0.00** | 0.51 ± 0.07 |
| | | C-VAE-SPATIAL+ (R=2) | 0.04 ± 0.01 | 0.04 ± 0.01 | 0.50 ± 0.05 |
| | | DAPSM | 0.25 ± 0.01 | n/a | n/a |
| | | GCNN | 0.36 ± 0.03 | n/a | n/a |
| | | GMERROR | 0.03 ± 0.00 | n/a | n/a |
| | | DURBIN | 0.03 ± 0.00 | 0.07 ± 0.00 | n/a |
| | | S2SLS-LAG1 | 0.03 ± 0.00 | 0.07 ± 0.00 | n/a |
| | | SPATIAL+ | 0.05 ± 0.02 | n/a | n/a |
| | | SPATIAL | **0.02 ± 0.00** | n/a | n/a |
| | | E-MAP-SPATIAL+ (R=1) | 0.04 ± 0.02 | 0.07 ± 0.07 | n/a |
| | | E-MAP-SPATIAL (R=1) | **0.02 ± 0.00** | 0.07 ± 0.07 | n/a |
| | $q_{summer}$ | C-VAE-SPATIAL+ (R=1) | **0.02 ± 0.01** | **0.02 ± 0.00** | 0.53 ± 0.05 |
| | | C-VAE-SPATIAL+ (R=2) | 0.04 ± 0.01 | 0.04 ± 0.01 | 0.51 ± 0.05 |
| | | DAPSM | 0.30 ± 0.03 | n/a | n/a |
| | | GCNN | 0.41 ± 0.03 | n/a | n/a |
| | | GMERROR | 0.20 ± 0.00 | n/a | n/a |
| | | DURBIN | 0.20 ± 0.00 | 0.06 ± 0.00 | n/a |
| | | S2SLS-LAG1 | 0.20 ± 0.00 | 0.06 ± 0.00 | n/a |
| | | SPATIAL+ | 0.05 ± 0.02 | n/a | n/a |
| | | SPATIAL | **0.02 ± 0.00** | n/a | n/a |
| | | E-MAP-SPATIAL+ (R=1) | 0.05 ± 0.02 | 0.07 ± 0.07 | n/a |
| | | E-MAP-SPATIAL (R=1) | **0.02 ± 0.00** | 0.07 ± 0.07 | n/a |
| $PM_{2.5} \rightarrow m \ (r_d = 2)$ | $\rho_{pop}$ | C-VAE-SPATIAL+ (R=1) | 0.19 ± 0.00 | **0.14 ± 0.00** | 0.52 ± 0.07 |
| | | C-VAE-SPATIAL+ (R=2) | 0.19 ± 0.00 | **0.14 ± 0.00** | 0.52 ± 0.02 |
| | | DAPSM | 0.16 ± 0.01 | n/a | n/a |
| | | GCNN | 0.18 ± 0.03 | n/a | n/a |
| | | GMERROR | **0.07 ± 0.00** | n/a | n/a |
| | | DURBIN | **0.07 ± 0.00** | 0.30 ± 0.00 | n/a |
| | | S2SLS-LAG1 | **0.07 ± 0.00** | 0.28 ± 0.00 | n/a |
| | | SPATIAL+ | 0.18 ± 0.00 | n/a | n/a |
| | | SPATIAL | 0.27 ± 0.00 | n/a | n/a |
| | | E-MAP-SPATIAL+ (R=2) | 0.19 ± 0.00 | 0.21 ± 0.00 | n/a |
| | | E-MAP-SPATIAL (R=2) | 0.27 ± 0.00 | 0.16 ± 0.05 | n/a |
| | $q_{summer}$ | C-VAE-SPATIAL+ (R=1) | 0.20 ± 0.00 | **0.14 ± 0.00** | 0.48 ± 0.06 |
| | | C-VAE-SPATIAL+ (R=2) | 0.19 ± 0.00 | **0.14 ± 0.00** | 0.49 ± 0.04 |
| | | DAPSM | 0.20 ± 0.01 | n/a | n/a |
| | | GCNN | 0.16 ± 0.05 | n/a | n/a |
| | | GMERROR | **0.09 ± 0.00** | n/a | n/a |
| | | DURBIN | **0.09 ± 0.00** | 0.25 ± 0.00 | n/a |
| | | S2SLS-LAG1 | **0.09 ± 0.00** | 0.27 ± 0.00 | n/a |
| | | SPATIAL+ | 0.18 ± 0.00 | n/a | n/a |
| | | SPATIAL | 0.27 ± 0.00 | n/a | n/a |
| | | E-MAP-SPATIAL+ (R=2) | 0.19 ± 0.00 | 0.20 ± 0.00 | n/a |
| | | E-MAP-SPATIAL (R=2) | 0.27 ± 0.00 | 0.16 ± 0.04 | n/a |
| $SO_4 \rightarrow PM_{2.5} \ (r_d = 1)$ | $NH_4$ | C-VAE-SPATIAL+ (R=1) | **0.07 ± 0.03** | 0.10 ± 0.00 | 0.51 ± 0.05 |
| | | C-VAE-SPATIAL+ (R=2) | 0.17 ± 0.03 | 0.09 ± 0.00 | 0.50 ± 0.04 |
| | | DAPSM | 1.44 ± 0.00 | n/a | n/a |
| | | GCNN | 0.52 ± 0.16 | n/a | n/a |
| | | GMERROR | 0.09 ± 0.00 | n/a | n/a |
| | | DURBIN | 0.09 ± 0.00 | 0.48 ± 0.00 | n/a |
| | | S2SLS-LAG1 | 0.09 ± 0.00 | 0.14 ± 0.00 | n/a |
| | | SPATIAL+ | 0.12 ± 0.07 | n/a | n/a |
| | | SPATIAL | 0.20 ± 0.00 | n/a | n/a |
| | | E-MAP-SPATIAL+ (R=1) | 0.10 ± 0.06 | 0.03 ± 0.01 | n/a |
| | | E-MAP-SPATIAL (R=1) | 0.20 ± 0.00 | 0.03 ± 0.01 | n/a |
| | $OC$ | C-VAE-SPATIAL+ (R=1) | 0.08 ± 0.03 | 0.10 ± 0.00 | 0.50 ± 0.06 |
| | | C-VAE-SPATIAL+ (R=2) | 0.14 ± 0.02 | 0.09 ± 0.00 | 0.50 ± 0.03 |
| | | DAPSM | 1.45 ± 0.00 | n/a | n/a |
| | | GCNN | 0.77 ± 0.22 | n/a | n/a |
| | | GMERROR | **0.00 ± 0.00** | n/a | n/a |
| | | DURBIN | **0.00 ± 0.00** | **0.02 ± 0.00** | n/a |
| | | S2SLS-LAG1 | **0.00 ± 0.00** | 0.15 ± 0.00 | n/a |

| | | | | | |
|---|---|---|---|---|---|
| | | SPATIAL+ | 0.11 ± 0.06 | n/a | n/a |
| | | SPATIAL | 0.20 ± 0.00 | n/a | n/a |
| | | E-MAP-SPATIAL+ (R=1) | 0.10 ± 0.06 | 0.03 ± 0.01 | n/a |
| | | E-MAP-SPATIAL (R=1) | 0.20 ± 0.00 | 0.03 ± 0.01 | n/a |
| $SO_4 \rightarrow PM_{2.5}$ ($r_d = 2$) | $NH_4$ | C-VAE-SPATIAL+ (R=1) | 0.02 ± 0.00 | **0.32 ± 0.00** | 0.51 ± 0.02 |
| | | C-VAE-SPATIAL+ (R=2) | 0.02 ± 0.00 | **0.32 ± 0.00** | 0.49 ± 0.03 |
| | | DAPSM | 1.23 ± 0.00 | n/a | n/a |
| | | GCNN | 0.26 ± 0.09 | n/a | n/a |
| | | GMERROR | 0.10 ± 0.00 | n/a | n/a |
| | | DURBIN | 0.10 ± 0.00 | 0.55 ± 0.00 | n/a |
| | | S2SLS-LAG1 | 0.10 ± 0.00 | 0.49 ± 0.00 | n/a |
| | | SPATIAL+ | 0.04 ± 0.00 | n/a | n/a |
| | | SPATIAL | 0.40 ± 0.00 | n/a | n/a |
| | | E-MAP-SPATIAL+ (R=2) | 0.04 ± 0.00 | 0.36 ± 0.00 | n/a |
| | | E-MAP-SPATIAL (R=2) | 0.40 ± 0.00 | 0.36 ± 0.01 | n/a |
| | $OC$ | C-VAE-SPATIAL+ (R=0) | 0.03 ± 0.00 | **0.32 ± 0.00** | 0.50 ± 0.03 |
| | | C-VAE-SPATIAL+ (R=1) | 0.03 ± 0.00 | **0.32 ± 0.00** | 0.50 ± 0.03 |
| | | C-VAE-SPATIAL+ (R=2) | 0.03 ± 0.00 | **0.32 ± 0.00** | 0.50 ± 0.04 |
| | | DAPSM | 1.24 ± 0.01 | n/a | n/a |
| | | GCNN | 0.30 ± 0.10 | n/a | n/a |
| | | GMERROR | 0.21 ± 0.00 | n/a | n/a |
| | | DURBIN | 0.22 ± 0.00 | 0.58 ± 0.00 | n/a |
| | | S2SLS-LAG1 | 0.21 ± 0.00 | 0.49 ± 0.00 | n/a |
| | | SPATIAL+ | 0.03 ± 0.00 | n/a | n/a |
| | | SPATIAL | 0.40 ± 0.00 | n/a | n/a |
| | | E-MAP-SPATIAL+ (R=1) | **0.03 ± 0.00** | 0.37 ± 0.00 | n/a |
| | | E-MAP-SPATIAL (R=1) | 0.40 ± 0.00 | 0.36 ± 0.01 | n/a |

*Table 6.* Performance under *spatial confounding*. Results averaged over 10 runs with 95% confidence intervals. $r_d$: neighborhood radius in data generation; R: neighborhood radius used by the deconfounder. Lower values for ATE and SPILL indicate less bias. $p$ indicates the $p$-value of the predictive check, with values near 0.5 indicating good model fit to 0.5. Best and second-best values are bolded and underlined, respectively.

| Environment | Confounder | Method | DIR | SPILL | $p$ |
|---|---|---|---|---|---|
| $PM_{2.5} \rightarrow m$ ($r_d = 1$) | $\rho_{pop}$ | C-VAE-UNET (R=1) | 0.06 ± 0.02 | 0.16 ± 0.08 | 0.53 ± 0.03 |
| | | C-VAE-UNET (R=2) | 0.04 ± 0.01 | 0.07 ± 0.02 | 0.51 ± 0.04 |
| | | DAPSM | 0.20 ± 0.01 | n/a | n/a |
| | | GCNN | 0.17 ± 0.06 | n/a | n/a |
| | | GMERROR | 0.05 ± 0.00 | n/a | n/a |
| | | DURBIN | 0.05 ± 0.00 | 0.05 ± 0.00 | n/a |
| | | S2SLS-LAG1 | 0.05 ± 0.00 | **0.04 ± 0.00** | n/a |
| | | SPATIAL+ | 0.12 ± 0.09 | n/a | n/a |
| | | SPATIAL | **0.03 ± 0.00** | n/a | n/a |
| | | UNET | 0.06 ± 0.01 | 0.17 ± 0.04 | n/a |
| | | E-MAP-SPATIAL+ (R=1) | 0.11 ± 0.09 | 0.06 ± 0.06 | n/a |
| | | E-MAP-SPATIAL+ (R=2) | 0.03 ± 0.02 | **0.04 ± 0.03** | n/a |
| | | E-MAP-SPATIAL (R=1) | **0.03 ± 0.00** | 0.07 ± 0.06 | n/a |
| | | E-MAP-SPATIAL (R=2) | **0.03 ± 0.00** | 0.05 ± 0.03 | n/a |
| | $q_{summer}$ | C-VAE-UNET (R=1) | 0.05 ± 0.01 | 0.10 ± 0.03 | 0.47 ± 0.04 |
| | | C-VAE-UNET (R=2) | 0.04 ± 0.01 | 0.10 ± 0.05 | 0.50 ± 0.05 |
| | | DAPSM | 0.28 ± 0.04 | n/a | n/a |
| | | GCNN | 0.23 ± 0.03 | n/a | n/a |
| | | GMERROR | 0.16 ± 0.00 | n/a | n/a |
| | | DURBIN | 0.16 ± 0.00 | 0.67 ± 0.00 | n/a |
| | | S2SLS-LAG1 | 0.16 ± 0.00 | **0.03 ± 0.00** | n/a |
| | | SPATIAL+ | 0.12 ± 0.09 | n/a | n/a |
| | | SPATIAL | **0.03 ± 0.00** | n/a | n/a |
| | | UNET | 0.04 ± 0.01 | 0.10 ± 0.05 | n/a |
| | | E-MAP-SPATIAL+ (R=1) | 0.11 ± 0.09 | 0.07 ± 0.06 | n/a |
| | | E-MAP-SPATIAL+ (R=2) | 0.03 ± 0.02 | 0.05 ± 0.04 | n/a |
| | | E-MAP-SPATIAL (R=1) | **0.03 ± 0.00** | 0.08 ± 0.06 | n/a |
| | | E-MAP-SPATIAL (R=2) | **0.03 ± 0.00** | 0.06 ± 0.04 | n/a |
| $PM_{2.5} \rightarrow m$ ($r_d = 2$) | $\rho_{pop}$ | C-VAE-UNET (R=1) | 0.16 ± 0.00 | 0.14 ± 0.04 | 0.49 ± 0.03 |
| | | C-VAE-UNET (R=2) | 0.16 ± 0.00 | 0.14 ± 0.05 | 0.48 ± 0.04 |
| | | DAPSM | 0.15 ± 0.02 | n/a | n/a |
| | | GCNN | 0.15 ± 0.04 | n/a | n/a |
| | | GMERROR | **0.06 ± 0.00** | n/a | n/a |
| | | DURBIN | 0.07 ± 0.00 | 0.19 ± 0.00 | n/a |
| | | S2SLS-LAG1 | **0.06 ± 0.00** | 0.19 ± 0.00 | n/a |
| | | SPATIAL+ | 0.08 ± 0.02 | n/a | n/a |
| | | SPATIAL | 0.15 ± 0.00 | n/a | n/a |
| | | UNET | 0.15 ± 0.01 | 0.15 ± 0.03 | n/a |
| | | E-MAP-SPATIAL+ (R=1) | 0.08 ± 0.02 | **0.08 ± 0.02** | n/a |
| | | E-MAP-SPATIAL+ (R=2) | 0.08 ± 0.02 | **0.08 ± 0.02** | n/a |
| | | E-MAP-SPATIAL (R=1) | 0.15 ± 0.00 | **0.08 ± 0.02** | n/a |

| | | | | | |
|---|---|---|---|---|---|
| | | E-MAP-SPATIAL (R=2) | $0.15 \pm 0.00$ | **$0.08 \pm 0.02$** | n/a |
| | $q_{summer}$ | C-VAE-UNET (R=1) | $0.15 \pm 0.01$ | $0.16 \pm 0.09$ | $0.55 \pm 0.05$ |
| | | C-VAE-UNET (R=2) | $0.16 \pm 0.00$ | $0.13 \pm 0.07$ | $0.51 \pm 0.05$ |
| | | DAPSM | $0.21 \pm 0.01$ | n/a | n/a |
| | | GCNN | $0.23 \pm 0.03$ | n/a | n/a |
| | | GMERROR | $0.10 \pm 0.00$ | n/a | n/a |
| | | DURBIN | $0.10 \pm 0.00$ | $0.19 \pm 0.00$ | n/a |
| | | S2SLS-LAG1 | $0.10 \pm 0.00$ | $0.19 \pm 0.00$ | n/a |
| | | SPATIAL+ | **$0.07 \pm 0.03$** | n/a | n/a |
| | | SPATIAL | $0.15 \pm 0.00$ | n/a | n/a |
| | | UNET | $0.15 \pm 0.00$ | $0.08 \pm 0.04$ | n/a |
| | | E-MAP-SPATIAL+ (R=1) | $0.08 \pm 0.02$ | $0.08 \pm 0.02$ | n/a |
| | | E-MAP-SPATIAL+ (R=2) | $0.08 \pm 0.03$ | $0.08 \pm 0.02$ | n/a |
| | | E-MAP-SPATIAL (R=1) | $0.15 \pm 0.00$ | **$0.07 \pm 0.02$** | n/a |
| | | E-MAP-SPATIAL (R=2) | $0.15 \pm 0.00$ | **$0.07 \pm 0.02$** | n/a |
| $SO_4 \rightarrow PM_{2.5}\ (r_d = 1)$ | $NH_4$ | C-VAE-UNET (R=1) | $0.07 \pm 0.01$ | $0.08 \pm 0.03$ | $0.51 \pm 0.05$ |
| | | C-VAE-UNET (R=2) | $0.08 \pm 0.01$ | **$0.04 \pm 0.03$** | $0.52 \pm 0.03$ |
| | | DAPSM | $1.56 \pm 0.00$ | n/a | n/a |
| | | GCNN | $0.55 \pm 0.09$ | n/a | n/a |
| | | GMERROR | $0.22 \pm 0.00$ | n/a | n/a |
| | | DURBIN | $0.22 \pm 0.00$ | $0.27 \pm 0.00$ | n/a |
| | | S2SLS-LAG1 | $0.22 \pm 0.00$ | $0.08 \pm 0.00$ | n/a |
| | | SPATIAL+ | $0.12 \pm 0.08$ | n/a | n/a |
| | | SPATIAL | $0.07 \pm 0.00$ | n/a | n/a |
| | | UNET | **$0.04 \pm 0.01$** | $0.19 \pm 0.04$ | n/a |
| | | E-MAP-SPATIAL+ (R=1) | $0.11 \pm 0.07$ | $0.05 \pm 0.01$ | n/a |
| | | E-MAP-SPATIAL+ (R=2) | $0.14 \pm 0.09$ | $0.06 \pm 0.02$ | n/a |
| | | E-MAP-SPATIAL (R=1) | $0.07 \pm 0.00$ | $0.05 \pm 0.01$ | n/a |
| | | E-MAP-SPATIAL (R=2) | $0.07 \pm 0.00$ | $0.06 \pm 0.02$ | n/a |
| | $OC$ | C-VAE-UNET (R=1) | $0.09 \pm 0.00$ | $0.12 \pm 0.02$ | $0.48 \pm 0.05$ |
| | | C-VAE-UNET (R=2) | **$0.06 \pm 0.01$** | $0.07 \pm 0.03$ | $0.54 \pm 0.03$ |
| | | DAPSM | $1.57 \pm 0.00$ | n/a | n/a |
| | | GCNN | $0.42 \pm 0.15$ | n/a | n/a |
| | | GMERROR | $0.13 \pm 0.00$ | n/a | n/a |
| | | DURBIN | $0.13 \pm 0.00$ | **$0.01 \pm 0.00$** | n/a |
| | | S2SLS-LAG1 | $0.13 \pm 0.00$ | $0.08 \pm 0.00$ | n/a |
| | | SPATIAL+ | $0.12 \pm 0.08$ | n/a | n/a |
| | | SPATIAL | $0.07 \pm 0.00$ | n/a | n/a |
| | | UNET | $0.07 \pm 0.02$ | $0.05 \pm 0.02$ | n/a |
| | | E-MAP-SPATIAL+ (R=1) | $0.10 \pm 0.07$ | $0.05 \pm 0.01$ | n/a |
| | | E-MAP-SPATIAL+ (R=2) | $0.14 \pm 0.08$ | $0.06 \pm 0.02$ | n/a |
| | | E-MAP-SPATIAL (R=1) | $0.07 \pm 0.00$ | $0.05 \pm 0.01$ | n/a |
| | | E-MAP-SPATIAL (R=2) | $0.07 \pm 0.00$ | $0.06 \pm 0.02$ | n/a |
| $SO_4 \rightarrow PM_{2.5}\ (r_d = 2)$ | $NH_4$ | C-VAE-UNET (R=1) | $0.16 \pm 0.01$ | $0.09 \pm 0.06$ | $0.48 \pm 0.03$ |
| | | C-VAE-UNET (R=2) | $0.15 \pm 0.00$ | **$0.07 \pm 0.03$** | $0.47 \pm 0.04$ |
| | | DAPSM | $1.47 \pm 0.00$ | n/a | n/a |
| | | GCNN | $0.66 \pm 0.21$ | n/a | n/a |
| | | GMERROR | $0.16 \pm 0.00$ | n/a | n/a |
| | | DURBIN | $0.15 \pm 0.00$ | $1.59 \pm 0.00$ | n/a |
| | | S2SLS-LAG1 | $0.16 \pm 0.00$ | $0.23 \pm 0.00$ | n/a |
| | | SPATIAL+ | $0.08 \pm 0.04$ | n/a | n/a |
| | | SPATIAL | $0.16 \pm 0.00$ | n/a | n/a |
| | | UNET | $0.15 \pm 0.01$ | $0.11 \pm 0.04$ | n/a |
| | | E-MAP-SPATIAL+ (R=1) | **$0.06 \pm 0.03$** | $0.11 \pm 0.05$ | n/a |
| | | E-MAP-SPATIAL+ (R=2) | **$0.06 \pm 0.03$** | $0.11 \pm 0.05$ | n/a |
| | | E-MAP-SPATIAL (R=1) | $0.16 \pm 0.00$ | $0.11 \pm 0.05$ | n/a |
| | | E-MAP-SPATIAL (R=2) | $0.16 \pm 0.00$ | $0.11 \pm 0.05$ | n/a |
| | $OC$ | C-VAE-UNET (R=1) | $0.15 \pm 0.01$ | $0.11 \pm 0.05$ | $0.50 \pm 0.03$ |
| | | C-VAE-UNET (R=2) | $0.15 \pm 0.01$ | **$0.08 \pm 0.05$** | $0.50 \pm 0.02$ |
| | | DAPSM | $1.49 \pm 0.01$ | n/a | n/a |
| | | GCNN | $0.67 \pm 0.12$ | n/a | n/a |
| | | GMERROR | $0.09 \pm 0.00$ | n/a | n/a |
| | | DURBIN | $0.08 \pm 0.00$ | $0.32 \pm 0.00$ | n/a |
| | | S2SLS-LAG1 | $0.09 \pm 0.00$ | $0.23 \pm 0.00$ | n/a |
| | | SPATIAL+ | $0.07 \pm 0.04$ | n/a | n/a |
| | | SPATIAL | $0.16 \pm 0.00$ | n/a | n/a |
| | | UNET | $0.15 \pm 0.01$ | **$0.08 \pm 0.04$** | n/a |
| | | E-MAP-SPATIAL+ (R=1) | **$0.06 \pm 0.03$** | $0.11 \pm 0.05$ | n/a |
| | | E-MAP-SPATIAL+ (R=2) | **$0.06 \pm 0.03$** | $0.11 \pm 0.05$ | n/a |
| | | E-MAP-SPATIAL (R=1) | $0.16 \pm 0.00$ | $0.11 \pm 0.05$ | n/a |
| | | E-MAP-SPATIAL (R=2) | $0.16 \pm 0.00$ | $0.11 \pm 0.05$ | n/a |

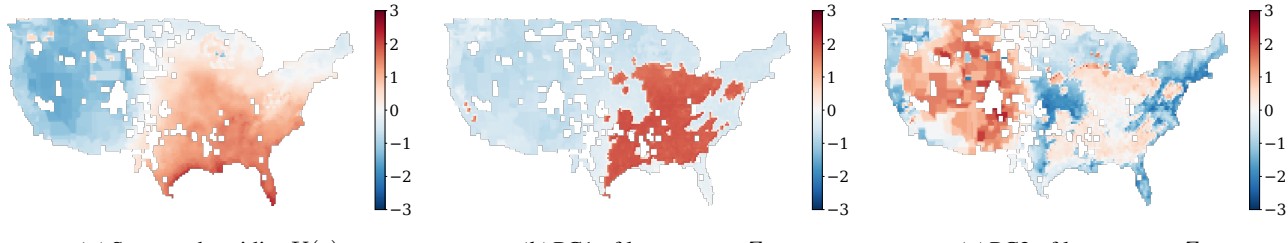

| (a) Summer humidity $U(s)$ | (b) PC1 of latent space $Z_s$ | (c) PC2 of latent space $Z_s$ |

*Figure 4.* Reconstructed latent confounder compared to the true (unobserved) spatial field. The leading principal component of the latent space $Z_s$ (PC1) captures the treatment, while the second principal component (PC2) recovers large-scale spatial structure of the true confounder.

# E. Additional Robustness Tests

## E.1. Treatment sparsity

The results in Table 7 examine our method under sparse treatment conditions with 10%, 30%, 50%, and 70% of grid cells receiving treatment. Under low sparsity, C-VAE-SPATIAL+ achieves direct effect estimation comparable with E-MAP-SPATIAL+. In addition, the predictive $p$-value is consistent across different treatment sparsity, showing good model calibration in sparse settings. However, at higher treatment coverage, our model does not match the performance of exposure mapping.

*Table 7.* Performance under **sparse** *local confounding*. Results averaged over 10 runs with 95% confidence intervals. $r_d$: neighborhood radius in data generation; R: neighborhood radius used by the deconfounder. Lower values for ATE and SPILL indicate less bias. $p$ indicates the predictive $p$-value, with values near 0.5 indicating good model fit to 0.5. Percentage in environment denotes the fraction of observations receiving treatment.

| Environment | Confounder | Method | DIR | SPILL | $p$ |
|---|---|---|---|---|---|
| $SO_4 \rightarrow PM_{2.5}$ ($r_d = 1$) (10%) | $NH_4$ | C-VAE-SPATIAL+ (R=0) | 0.08 ± 0.02 | 0.22 ± 0.00 | 0.54 ± 0.04 |
| | | C-VAE-SPATIAL+ (R=1) | 0.09 ± 0.04 | 0.22 ± 0.00 | 0.49 ± 0.02 |
| | | C-VAE-SPATIAL+ (R=2) | 0.09 ± 0.03 | 0.23 ± 0.00 | 0.51 ± 0.04 |
| | | DAPSM | **0.02 ± 0.00** | n/a | n/a |
| | | GCNN | 0.43 ± 0.10 | n/a | n/a |
| | | GMERROR | 0.04 ± 0.00 | n/a | n/a |
| | | DURBIN | 0.03 ± 0.00 | 0.46 ± 0.00 | n/a |
| | | S2SLS-LAG1 | 0.04 ± 0.00 | 0.28 ± 0.00 | n/a |
| | | SPATIAL+ | 0.09 ± 0.03 | n/a | n/a |
| | | SPATIAL | 0.20 ± 0.00 | n/a | n/a |
| | | E-MAP-SPATIAL+ (R=1) | 0.07 ± 0.03 | **0.15 ± 0.09** | n/a |
| | | E-MAP-SPATIAL+ (R=2) | 0.14 ± 0.03 | 0.16 ± 0.11 | n/a |
| | | E-MAP-SPATIAL (R=1) | 0.20 ± 0.00 | 0.16 ± 0.09 | n/a |
| | | E-MAP-SPATIAL (R=2) | 0.21 ± 0.00 | 0.16 ± 0.12 | n/a |
| | $OC$ | C-VAE-SPATIAL+ (R=0) | 0.10 ± 0.02 | 0.22 ± 0.00 | 0.59 ± 0.04 |
| | | C-VAE-SPATIAL+ (R=1) | 0.09 ± 0.03 | 0.22 ± 0.00 | 0.51 ± 0.03 |
| | | C-VAE-SPATIAL+ (R=2) | 0.11 ± 0.05 | 0.23 ± 0.00 | 0.48 ± 0.03 |
| | | DAPSM | **0.05 ± 0.02** | n/a | n/a |
| | | GCNN | 0.68 ± 0.19 | n/a | n/a |
| | | GMERROR | 0.26 ± 0.00 | n/a | n/a |
| | | DURBIN | 0.27 ± 0.00 | 0.35 ± 0.00 | n/a |
| | | S2SLS-LAG1 | 0.26 ± 0.00 | 0.28 ± 0.00 | n/a |
| | | SPATIAL+ | 0.08 ± 0.02 | n/a | n/a |
| | | SPATIAL | 0.20 ± 0.00 | n/a | n/a |
| | | E-MAP-SPATIAL+ (R=1) | 0.08 ± 0.03 | **0.15 ± 0.09** | n/a |
| | | E-MAP-SPATIAL+ (R=2) | 0.16 ± 0.04 | **0.15 ± 0.11** | n/a |
| | | E-MAP-SPATIAL (R=1) | 0.20 ± 0.00 | 0.16 ± 0.09 | n/a |
| | | E-MAP-SPATIAL (R=2) | 0.21 ± 0.00 | **0.15 ± 0.11** | n/a |
| $SO_4 \rightarrow PM_{2.5}$ ($r_d = 1$) (30%) | $NH_4$ | C-VAE-SPATIAL+ (R=0) | 0.14 ± 0.04 | 0.32 ± 0.00 | 0.56 ± 0.03 |
| | | C-VAE-SPATIAL+ (R=1) | 0.20 ± 0.06 | 0.32 ± 0.00 | 0.49 ± 0.04 |
| | | C-VAE-SPATIAL+ (R=2) | 0.18 ± 0.06 | 0.33 ± 0.00 | 0.52 ± 0.05 |
| | | DAPSM | 1.00 ± 0.00 | n/a | n/a |
| | | GCNN | 0.40 ± 0.09 | n/a | n/a |
| | | GMERROR | **0.03 ± 0.00** | n/a | n/a |
| | | DURBIN | **0.03 ± 0.00** | **0.10 ± 0.00** | n/a |
| | | S2SLS-LAG1 | **0.03 ± 0.00** | 0.38 ± 0.00 | n/a |
| | | SPATIAL+ | 0.11 ± 0.05 | n/a | n/a |
| | | SPATIAL | 0.33 ± 0.00 | n/a | n/a |
| | | E-MAP-SPATIAL+ (R=1) | 0.10 ± 0.04 | 0.20 ± 0.03 | n/a |
| | | E-MAP-SPATIAL+ (R=2) | 0.17 ± 0.04 | 0.21 ± 0.03 | n/a |

| | | | | | |
|---|---|---|---|---|---|
| | | E-MAP-SPATIAL (R=1) | 0.33 ± 0.00 | 0.20 ± 0.03 | n/a |
| | | E-MAP-SPATIAL (R=2) | 0.33 ± 0.00 | 0.20 ± 0.03 | n/a |
| | $OC$ | C-VAE-SPATIAL+ (R=0) | 0.15 ± 0.04 | 0.32 ± 0.00 | 0.53 ± 0.02 |
| | | C-VAE-SPATIAL+ (R=1) | 0.20 ± 0.07 | 0.32 ± 0.00 | 0.50 ± 0.04 |
| | | C-VAE-SPATIAL+ (R=2) | 0.20 ± 0.04 | 0.33 ± 0.00 | 0.49 ± 0.03 |
| | | DAPSM | 1.00 ± 0.00 | n/a | n/a |
| | | GCNN | 0.33 ± 0.14 | n/a | n/a |
| | | GMERROR | **0.07 ± 0.00** | n/a | n/a |
| | | DURBIN | **0.07 ± 0.00** | 0.55 ± 0.00 | n/a |
| | | S2SLS-LAG1 | **0.07 ± 0.00** | 0.38 ± 0.00 | n/a |
| | | SPATIAL+ | 0.12 ± 0.04 | n/a | n/a |
| | | SPATIAL | 0.33 ± 0.00 | n/a | n/a |
| | | E-MAP-SPATIAL+ (R=1) | 0.10 ± 0.03 | 0.20 ± 0.03 | n/a |
| | | E-MAP-SPATIAL+ (R=2) | 0.16 ± 0.03 | 0.20 ± 0.03 | n/a |
| | | E-MAP-SPATIAL (R=1) | 0.33 ± 0.00 | **0.19 ± 0.03** | n/a |
| | | E-MAP-SPATIAL (R=2) | 0.33 ± 0.00 | 0.20 ± 0.03 | n/a |
| $SO_4 \rightarrow PM_{2.5}\ (r_d = 1)\ (50\%)$ | $NH_4$ | C-VAE-SPATIAL+ (R=0) | **0.07 ± 0.02** | 0.09 ± 0.00 | 0.48 ± 0.02 |
| | | C-VAE-SPATIAL+ (R=1) | **0.07 ± 0.03** | 0.10 ± 0.00 | 0.51 ± 0.05 |
| | | C-VAE-SPATIAL+ (R=2) | 0.17 ± 0.03 | 0.09 ± 0.00 | 0.50 ± 0.04 |
| | | DAPSM | 1.44 ± 0.00 | n/a | n/a |
| | | GCNN | 0.52 ± 0.16 | n/a | n/a |
| | | GMERROR | 0.09 ± 0.00 | n/a | n/a |
| | | DURBIN | 0.09 ± 0.00 | 0.48 ± 0.00 | n/a |
| | | S2SLS-LAG1 | 0.09 ± 0.00 | 0.14 ± 0.00 | n/a |
| | | SPATIAL+ | 0.12 ± 0.07 | n/a | n/a |
| | | SPATIAL | 0.20 ± 0.00 | n/a | n/a |
| | | E-MAP-SPATIAL+ (R=1) | 0.10 ± 0.06 | 0.03 ± 0.01 | n/a |
| | | E-MAP-SPATIAL+ (R=2) | 0.13 ± 0.06 | **0.02 ± 0.01** | n/a |
| | | E-MAP-SPATIAL (R=1) | 0.20 ± 0.00 | 0.03 ± 0.01 | n/a |
| | | E-MAP-SPATIAL (R=2) | 0.19 ± 0.00 | **0.02 ± 0.01** | n/a |
| | $OC$ | C-VAE-SPATIAL+ (R=0) | 0.07 ± 0.03 | 0.09 ± 0.00 | 0.50 ± 0.04 |
| | | C-VAE-SPATIAL+ (R=1) | 0.08 ± 0.03 | 0.10 ± 0.00 | 0.50 ± 0.06 |
| | | C-VAE-SPATIAL+ (R=2) | 0.14 ± 0.02 | 0.09 ± 0.00 | 0.50 ± 0.03 |
| | | DAPSM | 1.45 ± 0.00 | n/a | n/a |
| | | GCNN | 0.77 ± 0.22 | n/a | n/a |
| | | GMERROR | **0.00 ± 0.00** | n/a | n/a |
| | | DURBIN | **0.00 ± 0.00** | **0.02 ± 0.00** | n/a |
| | | S2SLS-LAG1 | **0.00 ± 0.00** | 0.15 ± 0.00 | n/a |
| | | SPATIAL+ | 0.11 ± 0.06 | n/a | n/a |
| | | SPATIAL | 0.20 ± 0.00 | n/a | n/a |
| | | E-MAP-SPATIAL+ (R=1) | 0.10 ± 0.06 | 0.03 ± 0.01 | n/a |
| | | E-MAP-SPATIAL+ (R=2) | 0.13 ± 0.05 | **0.02 ± 0.01** | n/a |
| | | E-MAP-SPATIAL (R=1) | 0.20 ± 0.00 | 0.03 ± 0.01 | n/a |
| | | E-MAP-SPATIAL (R=2) | 0.19 ± 0.00 | **0.02 ± 0.01** | n/a |
| $SO_4 \rightarrow PM_{2.5}\ (r_d = 1)\ (70\%)$ | $NH_4$ | C-VAE-SPATIAL+ (R=0) | 0.28 ± 0.05 | 0.45 ± 0.00 | 0.55 ± 0.04 |
| | | C-VAE-SPATIAL+ (R=1) | 0.32 ± 0.04 | 0.46 ± 0.00 | 0.55 ± 0.05 |
| | | C-VAE-SPATIAL+ (R=2) | 0.37 ± 0.04 | 0.46 ± 0.00 | 0.51 ± 0.02 |
| | | DAPSM | 0.29 ± 0.07 | n/a | n/a |
| | | GCNN | 0.29 ± 0.09 | n/a | n/a |
| | | GMERROR | **0.17 ± 0.00** | n/a | n/a |
| | | DURBIN | **0.17 ± 0.00** | 2.54 ± 0.00 | n/a |
| | | S2SLS-LAG1 | **0.17 ± 0.00** | 0.51 ± 0.00 | n/a |
| | | SPATIAL+ | 0.22 ± 0.08 | n/a | n/a |
| | | SPATIAL | 0.44 ± 0.00 | n/a | n/a |
| | | E-MAP-SPATIAL+ (R=1) | 0.24 ± 0.07 | 0.36 ± 0.03 | n/a |
| | | E-MAP-SPATIAL+ (R=2) | 0.29 ± 0.04 | **0.35 ± 0.03** | n/a |
| | | E-MAP-SPATIAL (R=1) | 0.44 ± 0.00 | **0.35 ± 0.03** | n/a |
| | | E-MAP-SPATIAL (R=2) | 0.44 ± 0.00 | **0.35 ± 0.03** | n/a |
| | $OC$ | C-VAE-SPATIAL+ (R=0) | 0.28 ± 0.05 | 0.46 ± 0.00 | 0.55 ± 0.05 |
| | | C-VAE-SPATIAL+ (R=1) | 0.26 ± 0.07 | 0.46 ± 0.00 | 0.47 ± 0.04 |
| | | C-VAE-SPATIAL+ (R=2) | 0.27 ± 0.08 | 0.46 ± 0.00 | 0.50 ± 0.04 |
| | | DAPSM | **0.19 ± 0.00** | n/a | n/a |
| | | GCNN | 0.33 ± 0.07 | n/a | n/a |
| | | GMERROR | 0.24 ± 0.00 | n/a | n/a |
| | | DURBIN | 0.24 ± 0.00 | 0.58 ± 0.00 | n/a |
| | | S2SLS-LAG1 | 0.24 ± 0.00 | 0.51 ± 0.00 | n/a |
| | | SPATIAL+ | 0.21 ± 0.08 | n/a | n/a |
| | | SPATIAL | 0.44 ± 0.00 | n/a | n/a |
| | | E-MAP-SPATIAL+ (R=1) | 0.24 ± 0.07 | 0.36 ± 0.03 | n/a |
| | | E-MAP-SPATIAL+ (R=2) | 0.29 ± 0.04 | 0.36 ± 0.03 | n/a |
| | | E-MAP-SPATIAL (R=1) | 0.44 ± 0.00 | **0.35 ± 0.03** | n/a |
| | | E-MAP-SPATIAL (R=2) | 0.44 ± 0.00 | **0.35 ± 0.03** | n/a |

## E.2. Performance under single-cause confounders

We evaluate our method under violation of Assumption 4 by introducing a localized single-cause unobserved confounder named $SC$. We select $\mathcal{C} = \{c_1, \ldots, c_n\}$ as cluster centers, drawn uniformly from the set of spatial sites, where $n = \lceil s|\mathcal{S}| \rceil$

and $s$ denotes the sparsity. Each cluster center is assigned a peak intensity $\alpha_c \sim U(0.5, 1.0)$ for any site $s$, the resulting single-cause confounder is

$$SC_s = \max_{c \in \mathcal{C}} \alpha_c \exp\left(-\frac{d(s,c)}{2}\right)$$

where $d(s,c)$ is the shortest distance path between $s$ and $c$. We then inject $SC$ into both the treatment and outcome by adding $0.8 \times \mathrm{std}(X) \times SC$ to each variable where $X$ denotes the respective treatment or outcome variable. The treatments are binarized by applying a threshold.

Table 8 presents performance when Assumption 4 is violated. When the localized unobserved confounder affects only 10% or 20% of cells, C-VAE-SPATIAL+ remains competitive with SPATIAL+ and retains accurate spillover estimates. At 30% localization, however, performance degrades, and SPATIAL+ and E-MAP-SPATIAL+ become competitive. This confirms that highly localized unobserved confounding is a failure mode for multi-cause deconfounding.

*Table 8.* Performance under *local confounding* with **single-cause unobserved confounder** $SC$. Results averaged over 10 runs with 95% confidence intervals. $r_d$: neighborhood radius in data generation; R: neighborhood radius used by the deconfounder. Lower values for ATE and SPILL indicate less bias. $p$ indicates the predictive $p$-value, with values near 0.5 indicating good model fit to 0.5. Percentage in environment denotes the fraction of observations receiving treatment.

| Environment | Confounder | Method | DIR | SPILL | $p$ |
|---|---|---|---|---|---|
| $PM_{2.5} \rightarrow m \ (r_d = 1)$ (10%) | $SC$ | C-VAE-SPATIAL+ (R=0) | 0.04 ± 0.01 | **0.01 ± 0.00** | 0.53 ± 0.02 |
| | | C-VAE-SPATIAL+ (R=1) | 0.04 ± 0.01 | **0.01 ± 0.00** | 0.50 ± 0.03 |
| | | C-VAE-SPATIAL+ (R=2) | 0.04 ± 0.02 | **0.01 ± 0.00** | 0.47 ± 0.03 |
| | | DAPSM | 0.52 ± 0.01 | n/a | n/a |
| | | GCNN | 0.13 ± 0.04 | n/a | n/a |
| | | GMERROR | 0.20 ± 0.00 | n/a | n/a |
| | | DURBIN | 0.21 ± 0.00 | 0.24 ± 0.00 | n/a |
| | | S2SLS-LAG1 | 0.20 ± 0.00 | 0.04 ± 0.00 | n/a |
| | | SPATIAL+ | **0.03 ± 0.01** | n/a | n/a |
| | | SPATIAL | 0.08 ± 0.00 | n/a | n/a |
| | | E-MAP-SPATIAL+ (R=1) | 0.04 ± 0.01 | 0.07 ± 0.07 | n/a |
| | | E-MAP-SPATIAL+ (R=2) | 0.04 ± 0.01 | 0.04 ± 0.01 | n/a |
| | | E-MAP-SPATIAL (R=1) | 0.08 ± 0.00 | 0.08 ± 0.07 | n/a |
| | | E-MAP-SPATIAL (R=2) | 0.08 ± 0.00 | 0.04 ± 0.01 | n/a |
| $PM_{2.5} \rightarrow m \ (r_d = 1)$ (20%) | $SC$ | C-VAE-SPATIAL+ (R=0) | **0.03 ± 0.01** | **0.01 ± 0.00** | 0.50 ± 0.03 |
| | | C-VAE-SPATIAL+ (R=1) | 0.06 ± 0.01 | **0.01 ± 0.00** | 0.51 ± 0.03 |
| | | C-VAE-SPATIAL+ (R=2) | 0.10 ± 0.02 | 0.03 ± 0.00 | 0.53 ± 0.05 |
| | | DAPSM | 0.59 ± 0.00 | n/a | n/a |
| | | GCNN | 0.07 ± 0.05 | n/a | n/a |
| | | GMERROR | 0.18 ± 0.00 | n/a | n/a |
| | | DURBIN | 0.18 ± 0.00 | 0.08 ± 0.00 | n/a |
| | | S2SLS-LAG1 | 0.18 ± 0.00 | 0.06 ± 0.00 | n/a |
| | | SPATIAL+ | 0.04 ± 0.02 | n/a | n/a |
| | | SPATIAL | 0.08 ± 0.00 | n/a | n/a |
| | | E-MAP-SPATIAL+ (R=1) | 0.04 ± 0.01 | 0.02 ± 0.01 | n/a |
| | | E-MAP-SPATIAL+ (R=2) | 0.05 ± 0.01 | 0.02 ± 0.01 | n/a |
| | | E-MAP-SPATIAL (R=1) | 0.08 ± 0.00 | 0.02 ± 0.01 | n/a |
| | | E-MAP-SPATIAL (R=2) | 0.09 ± 0.00 | 0.02 ± 0.01 | n/a |
| $PM_{2.5} \rightarrow m \ (r_d = 1)$ (30%) | $SC$ | C-VAE-SPATIAL+ (R=0) | 0.12 ± 0.01 | 0.16 ± 0.00 | 0.51 ± 0.03 |
| | | C-VAE-SPATIAL+ (R=1) | 0.15 ± 0.04 | 0.16 ± 0.00 | 0.51 ± 0.02 |
| | | C-VAE-SPATIAL+ (R=2) | 0.14 ± 0.06 | 0.17 ± 0.00 | 0.48 ± 0.04 |
| | | DAPSM | 0.58 ± 0.00 | n/a | n/a |
| | | GCNN | 0.18 ± 0.05 | n/a | n/a |
| | | GMERROR | 0.23 ± 0.00 | n/a | n/a |
| | | DURBIN | 0.23 ± 0.00 | 0.23 ± 0.00 | n/a |
| | | S2SLS-LAG1 | 0.23 ± 0.00 | 0.21 ± 0.00 | n/a |
| | | SPATIAL+ | **0.10 ± 0.01** | n/a | n/a |
| | | SPATIAL | 0.15 ± 0.00 | n/a | n/a |
| | | E-MAP-SPATIAL+ (R=1) | **0.10 ± 0.01** | 0.14 ± 0.02 | n/a |
| | | E-MAP-SPATIAL+ (R=2) | **0.10 ± 0.02** | **0.13 ± 0.02** | n/a |
| | | E-MAP-SPATIAL (R=1) | 0.15 ± 0.00 | **0.13 ± 0.02** | n/a |
| | | E-MAP-SPATIAL (R=2) | 0.15 ± 0.00 | **0.13 ± 0.02** | n/a |

## E.3. Performance under interference topology misspecification

We evaluate our method under asymmetric interference topology. We modified the data-generating process so that interference flows only eastward ($j' \geq j$), simulating wind-driven pollution transport. Our model still uses the default symmetric $\ell_\infty$ neighborhoods at prediction time. In Table 9, the C-VAE-SPATIAL+ models have significantly lower

direct effect bias and similar spillover bias compared to E-MAP-SPATIAL+, which makes our method robust to topology misspecification.

*Table 9.* Performance under *local confounding* with **asymmetric interference topology**. Results averaged over 10 runs with 95% confidence intervals. $r_d$: neighborhood radius in data generation; R: neighborhood radius used by the deconfounder. Lower values for ATE and SPILL indicate less bias. $p$ indicates the predictive $p$-value, with values near 0.5 indicating good model fit to 0.5. Percentage in environment denotes the fraction of observations receiving treatment.

| Environment | Confounder | Method | DIR | SPILL | $p$ |
|---|---|---|---|---|---|
| $PM\_2.5 \rightarrow m\ (r=1)$ | $q_{summer}$ | C-VAE-SPATIAL+ (R=0) | **0.01 ± 0.01** | **0.03 ± 0.00** | 0.49 ± 0.02 |
| | | C-VAE-SPATIAL+ (R=1) | 0.02 ± 0.01 | **0.03 ± 0.00** | 0.52 ± 0.03 |
| | | C-VAE-SPATIAL+ (R=2) | 0.03 ± 0.02 | 0.04 ± 0.00 | 0.52 ± 0.03 |
| | | DAPSM | 0.35 ± 0.03 | n/a | n/a |
| | | GCNN | 0.27 ± 0.05 | n/a | n/a |
| | | GMERROR | 0.23 ± 0.00 | n/a | n/a |
| | | DURBIN | 0.25 ± 0.00 | 0.35 ± 0.01 | n/a |
| | | S2SLS-LAG1 | 0.23 ± 0.00 | 0.07 ± 0.00 | n/a |
| | | SPATIAL+ | 0.13 ± 0.05 | n/a | n/a |
| | | SPATIAL | **0.01 ± 0.00** | n/a | n/a |
| | | E-MAP-SPATIAL+ (R=1) | 0.12 ± 0.05 | **0.03 ± 0.02** | n/a |
| | | E-MAP-SPATIAL+ (R=2) | 0.07 ± 0.03 | **0.03 ± 0.03** | n/a |
| | | E-MAP-SPATIAL (R=1) | **0.01 ± 0.00** | **0.03 ± 0.03** | n/a |
| | | E-MAP-SPATIAL (R=2) | 0.03 ± 0.00 | **0.03 ± 0.03** | n/a |

## E.4. Performance under coupled confounding and interference

We also evaluate a confounder-modulated spillover setting by adding $\alpha \cdot U_s \cdot \bar{A}_{\mathcal{N}_s}$ to the outcome, so that spillover strength explicitly depends on the hidden confounder. We choose $\alpha$ so that this term has standard deviation approximately $0.5 \cdot \mathrm{std}(Y_s)$. This creates a direct coupling between confounding and interference and violates the separability condition in Theorem 1. As expected, this coupling degrades the performance across the board: all methods show substantially increased bias compared to standard benchmarks, confirming this is a fundamentally harder problem rather than a weakness specific approach to our problem. Despite this, the $p$-values remain near 0.5, indicating the C-VAE still reconstructs the latent field well. Overall, our method achieves similar performance to exposure mapping with GCNN achieving the best direct effect estimation.

*Table 10.* Performance under *local confounding* with **confounder-modulated spillover**. Results averaged over 10 runs with 95% confidence intervals. $r_d$: neighborhood radius in data generation; R: neighborhood radius used by the deconfounder. Lower values for ATE and SPILL indicate less bias. $p$ indicates the predictive $p$-value, with values near 0.5 indicating good model fit to 0.5. Percentage in environment denotes the fraction of observations receiving treatment.

| Environment | Confounder | Method | DIR | SPILL | $p$ |
|---|---|---|---|---|---|
| $PM\_2.5 \rightarrow m\ (r=1)$ | $q_{summer}$ | C-VAE-SPATIAL+ (R=0) | 0.26 ± 0.09 | 0.85 ± 0.00 | 0.52 ± 0.03 |
| | | C-VAE-SPATIAL+ (R=1) | 0.37 ± 0.08 | 0.85 ± 0.00 | 0.51 ± 0.01 |
| | | C-VAE-SPATIAL+ (R=2) | 0.49 ± 0.02 | 0.85 ± 0.00 | 0.51 ± 0.04 |
| | | DAPSM | 0.66 ± 0.03 | n/a | n/a |
| | | GCNN | **0.24 ± 0.05** | n/a | n/a |
| | | GMERROR | 0.57 ± 0.00 | n/a | n/a |
| | | DURBIN | 0.57 ± 0.00 | 0.77 ± 0.00 | n/a |
| | | S2SLS-LAG1 | 0.57 ± 0.00 | 0.90 ± 0.00 | n/a |
| | | SPATIAL+ | 0.30 ± 0.08 | n/a | n/a |
| | | SPATIAL | 0.53 ± 0.00 | n/a | n/a |
| | | E-MAP-SPATIAL+ (R=1) | 0.32 ± 0.07 | 0.69 ± 0.21 | n/a |
| | | E-MAP-SPATIAL+ (R=2) | 0.26 ± 0.08 | 0.77 ± 0.19 | n/a |
| | | E-MAP-SPATIAL (R=1) | 0.53 ± 0.00 | **0.68 ± 0.21** | n/a |
| | | E-MAP-SPATIAL (R=2) | 0.53 ± 0.00 | 0.75 ± 0.19 | n/a |

## E.5. Case Study on Real-World Arctic Data

Here, we present a real-world case study comparing C-VAE-UNET with a regular UNET on real-world Pan-Arctic data. The treatment is downward longwave radiation (LWDN) in the Laptev and East Siberian seas, the outcome is sea ice concentration (SIC), and heat flux (HFX) is used as an observed confounder. This setting features both spatial interference, since radiation changes in one subregion can affect sea ice in neighboring Arctic regions, and plausible unobserved spatial

confounding from large-scale atmospheric circulation, ocean heat transport, and cloud cover. We compare qualitatively against controlled climate-model perturbation results: Kapsch et al. (2016) find that a 5% change in LWDN changes SIC by roughly 3–4% annually and 5–7% in summer, with increased LWDN reducing sea ice. Under a 5% reduction in LWDN, UNET predicts only a 0.3% annual increase and a 0.7% summer decrease in SIC, failing to recover the expected seasonal response. In contrast, C-VAE-UNET with $\{R, d_Z\} = \{1, 32\}$ predicts a 1.5% annual increase and a 5% summer increase, recovering the expected direction and a summer magnitude close to the climate-model simulations. This suggests that latent-variable adjustment yields more physically plausible behavior than UNET. For context, Ali et al. (2024) report larger effects with STCINET (4% for annual and 44% for summer), but we view the controlled perturbation results of Kapsch et al. (2016) as the more relevant qualitative benchmark.

### E.6. Sensitivity to hyperparameters and spillover radius

**Hyperparameters:** To assess the robustness of our spatial deconfounder across different hyperparameter sets, we conduct a sensitivity analysis. Below, we provide figures that display how the hyperparameters of C-VAE-SPATIAL+ and C-VAE-UNET affect the estimation performance. Specifically, we assess the hyperparameters $\beta$ (KL term), the latent dimension $d_Z$, the learning rate, and weight decay. We observe the change in one parameter at a time, while optimizing the other hyperparameters conditional on the assessed parameter.

For C-VAE-SPATIAL+, we do not observe a consistent pattern in the error for the direct effect DIR. The estimation performance remains robust when changing a single hyperparameter while optimizing all others. For the spillover effect SPILL estimation, we generally observe that the error increases with $\beta$ but decreases as $d_Z$ grows. In our models, the optimal $\beta$ and $d_Z$ are determined through hyperparameter tuning on the MSE Loss. Datasets with large $r_d$ typically need low $\beta$ because the smoothness is lower. On the other hand, datasets with small $r_d$ need a higher $\beta$ to enforce smoothness constraints. Furthermore, the optimal value of $\beta$ depends on the nature of the unobserved confounder. For instance, models with a smooth confounder such as humidity $q_{\text{summer}}$ favor a larger $\beta$, whereas models with an anisotropic confounder like the population density $\rho_{\text{pop}}$ require a relatively smaller $\beta$. For C-VAE-UNET, both direct and spillover error remain relatively stable across the hyperparameter ranges, suggesting that the deconfounding stage is robust when paired with a flexible spatial outcome model.

**Neighborhood radius:** Furthermore, we assess the robustness of our spatial deconfounder with respect to different interference radii in Figures 5 to 6 for C-VAE-SPATIAL+ and Figures 7 to 8 for C-VAE-UNET. We observe that our spatial deconfounder is generally robust to misspecification of the interference radius. Note that we do not include $r = 0$ models in SPILL plots, as these models cannot include neighboring treatments, i.e., spillover effect, by design.

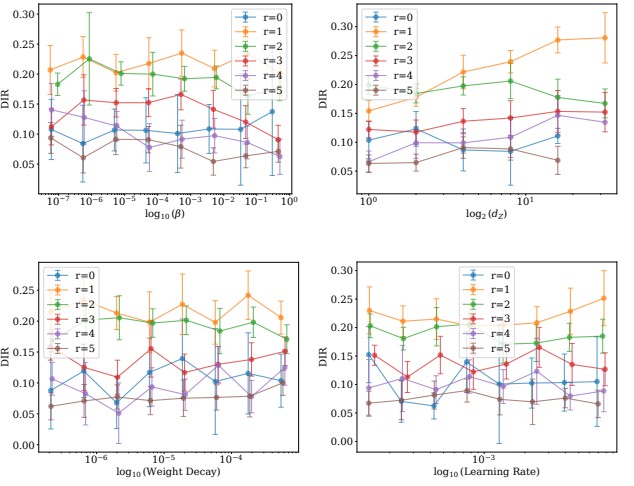

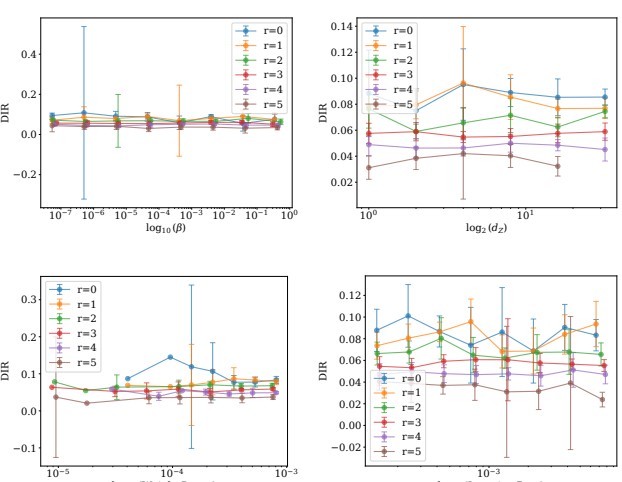

*Figure 5.* Sensitivity analysis for C-VAE-SPATIAL+ models trained on local confounding environment $SO_4 \rightarrow PM_{2.5}$ $(r_d = 1)$ with unobserved confounder $OC$. Each subplot shows DIR as a function of a hyperparameter across different neighborhood radii $r$. The error bounds represent the 95% confidence interval. The y-axis represents the error on the direct effect.

*Figure 7.* Sensitivity analysis for C-VAE-UNET models trained on spatial confounding environment $SO_4 \rightarrow PM_{2.5}$ $(r_d = 1)$ with unobserved confounder $OC$. Each subplot shows DIR as a function of a hyperparameter across different neighborhood radii $r$. The error bounds represent the 95% confidence interval.

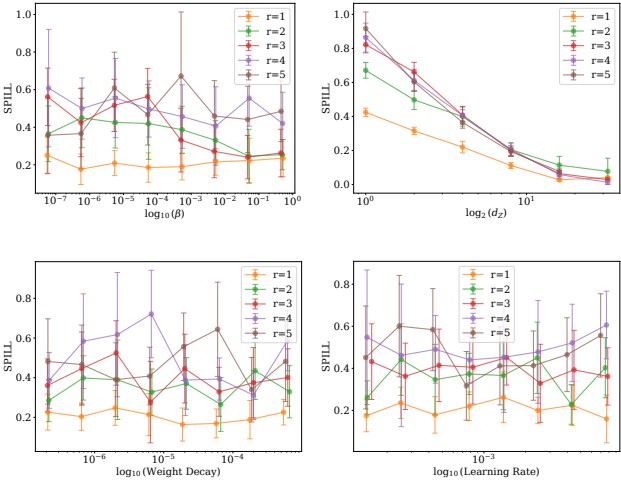

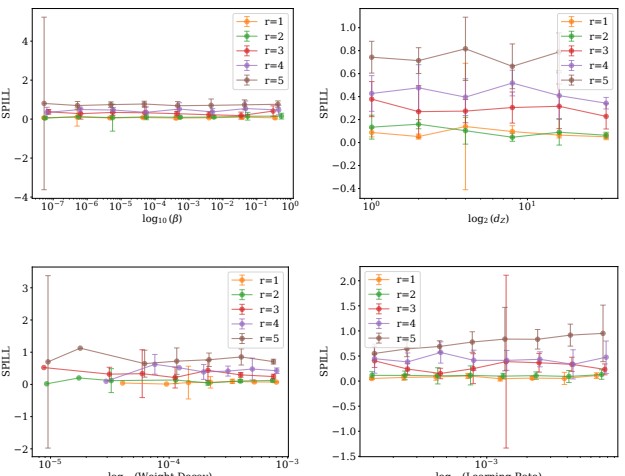

*Figure 6.* Sensitivity analysis for C-VAE-SPATIAL+ models trained on local confounding environment $SO_4 \rightarrow PM_{2.5}$ $(r_d = 1)$ with unobserved confounder $OC$. Each subplot shows SPILL as a function of a hyperparameter across different neighborhood radii $r$. The error bounds represent the 95% confidence interval. The y-axis represents the error on the spillover effect.

*Figure 8.* Sensitivity analysis for C-VAE-UNET models trained on spatial confounding environment $SO_4 \rightarrow PM_{2.5}$ $(r_d = 1)$ with unobserved confounder $OC$. Each subplot shows SPILL as a function of a hyperparameter across different neighborhood radii $r$. The error bounds represent the 95% confidence interval.

