# OpenReview forum: "Spatial Deconfounder: Interference-Aware Deconfounding for Spatial Causal Inference"
_ICML.cc/2026/Conference — ICML 2026 regular_

### Official Review · Reviewer_a3qB · 2026-03-03

**Soundness:** 3
**Presentation:** 3
**Significance:** 3
**Originality:** 3
**Overall Recommendation:** 4
**Confidence:** 4

**Summary:**

The paper proposes the Spatial Deconfounder, a two-stage method for causal inference in spatial data that handles both unobserved spatial confounding and spatial interference. The authors observe that treatment interference from neighboring units can act as a multi-cause signal. In the first stage, they use a conditional variational autoencoder with a Gaussian-Markov random field prior to reconstruct a substitute latent confounder from local and neighborhood treatments. In the second stage, this reconstructed confounder is used alongside observed covariates in an outcome model to estimate direct and spillover effects. The authors provide identifiability conditions and evaluate their method on semi-synthetic extensions of the SpaCE benchmark, demonstrating improvements over several baselines.

**Compliance With Llm Reviewing Policy:**

Affirmed.

**Final Justification:**

The paper presents an original and conceptually significant approach to spatial causal inference by treating interference as a multi-cause signal rather than a nuisance. The overall methodology is sound, and the presentation is clear.

In their rebuttal, the authors adequately addressed my main theoretical concerns. They clarified the empirical robustness of their method to the separability assumption and proposed a useful predictive p-value diagnostic for unmeasured local confounders. I appreciate these constructive additions.

However, some empirical weaknesses remain. While the authors expanded their baseline comparisons, they deferred the direct comparison with joint Bayesian models and the implementation of doubly robust uncertainty quantification to future work. I accept their justification regarding the timeline and scope constraints, but these omissions still restrict the comprehensive evaluation of the proposed method.

Overall, the rebuttal reinforced my initial assessment. The paper offers a solid conceptual contribution to the field that others can build upon, despite the acknowledged limitations in its evaluation and uncertainty quantification. Therefore, I am maintaining my final recommendation of Weak Accept.

**Key Questions For Authors:**

Key Questions For Authors

1.The theoretical identifiability relies on the separability of the outcome model as stated in Theorem 1. How sensitive is the empirical performance to violations of this separability assumption? It would be helpful to see simulations where treatments and the latent confounder interact non-linearly in the outcome generation process.

2.Regarding Assumption 4, it is highly likely in real-world spatial data that some unmeasured confounders are purely local. While you provide a stress test in the appendix, could you elaborate on any practical diagnostics or sensitivity analyses a user could run to detect such violations before trusting the estimated effects?

3.Why did you not compare your method against joint Bayesian models that handle both interference and unobserved spatial confounding, such as the work by Papadogeorgou and Samanta? Even a simplified empirical comparison on your extended benchmark would help contextualize your approach against the current state-of-the-art.

4.The current uncertainty quantification only accounts for the variation in the reconstructed latent variable. Have you considered integrating orthogonalized or doubly robust estimators in the second stage to account for outcome model uncertainty and improve finite-sample robustness?

**Limitations:**

Yes. The authors have adequately discussed the theoretical limitations of their work, specifically regarding the reliance on the latent-field sufficiency assumption. They have also included an appropriate impact statement acknowledging the risks of deploying causal estimates in high-stakes environmental or social policy contexts without proper validation.

**Strengths And Weaknesses:**

Strengths:

1.The conceptual innovation is highly appealing. Treating interference as a multi-cause signal rather than merely a nuisance to be adjusted for is a clever bridge between spatial causal inference and the deconfounder literature. It offers a fresh perspective on a classic problem.

2.The proposed framework is modular and model-agnostic. The separation of the confounder reconstruction stage from the potential outcome module allows practitioners to plug in various spatial models, such as U-Nets or graph neural networks, based on the specific data structure and application needs.

3.The empirical evaluation is built on a solid foundation. Extending the SpaCE benchmark to incorporate localized interference is a useful contribution to the community. Additionally, the stress tests in the appendix, particularly those injecting single-cause latent confounders, provide honest insights into the breaking points of the proposed method.

Weaknesses:

1.The theoretical claims, particularly regarding nonparametric identification, rest on strong assumptions that are difficult to verify in practice. Assumption 4 requires that all purely local unmeasured confounders are observed. If a local unmeasured factor exists, the method's identification guarantees are compromised. Furthermore, Theorem 1 requires a separable outcome structure, which slightly contradicts the broader claim of fully nonparametric identification.

2.The baseline comparisons are somewhat incomplete. While the authors compare against standard spatial methods and matched estimators, they omit recent methods that explicitly model joint spatial confounding and interference. For example, comparing against Bayesian joint models, such as the 2023 work by Papadogeorgou and Samanta, would provide a much clearer picture of the relative advantages and trade-offs of this factor-model approach.

3.Uncertainty quantification is underdeveloped. The authors estimate confidence bands by drawing samples from the reconstructed latent space, but this ignores the finite-sample uncertainty of the outcome model itself. Relying solely on latent draws may lead to overconfident causal estimates in practical scenarios.

---

> ### Author Rebuttal · Authors · 2026-03-31
>
> We thank the reviewer for the encouraging assessment and the specific, actionable suggestions. We address each question below
>
> ---
>
> **Sensitivity to separability (Theorem 1).** The separability condition is a *theoretical device* for our identification proof, not a restriction imposed on the data-generating process. In our semi-synthetic experiments, outcomes are generated by fitting a flexible function $f(\cdot)$ of observed covariates, then masking one covariate that also affects treatment. This induces general treatment–confounder interactions that are not constrained to be separable. The only restriction we impose is decorrelating residual noise so no additional confounding remains beyond the masked covariate. The strong empirical performance under these non-separable generators suggests the method is robust to moderate violations. We will clarify this distinction in the revision.
>
>
> **Diagnostics for Assumption 4 violations.** Our primary diagnostic is the predictive p-value (Eq. 12): when the C-VAE fails to capture the treatment-assignment distribution–-as happens when purely local confounders are present–-the p-value moves away from 0.5. The single-cause confounder stress test (Table 8) demonstrates this empirically: at 10% localization, both bias increases and p-value decreases. We recommend practitioners (1) check $p \in [0.25, 0.75]$ before trusting estimates, and (2) run sensitivity analyses by varying the latent dimension $d_Z$ and neighborhood radius $R$ to assess stability. We will add a practical guidance paragraph.
>
> **Comparison to joint and Bayesian models.** In response to this and Reviewer aKFv's feedback, we have strengthened the baseline set: we added spillover-aware variants of SPATIAL+ and SPATIAL via exposure mapping, enabled interference estimation for S2SLS-LAG1, and included SAR-type models (S2SLS-DURBIN, GMERROR) that explicitly capture spatial dependence structure. Our method matches or outperforms these across representative settings. Regarding Papadogeorgou & Samanta (2023) specifically, their Bayesian joint model handles both interference and latent spatial fields but relies on parametric outcome specifications and carefully chosen prior structures that require domain-specific knowledge not readily available in our semi-synthetic benchmark settings. A full empirical comparison would also require adapting their method to our interference structure, which is beyond the rebuttal budget. We will add a detailed discussion comparing the conceptual tradeoffs and flag the empirical comparison as important future work.
>
> **Uncertainty quantification.** You raise a good point that our current UQ only accounts for latent-variable uncertainty, not outcome-model uncertainty. Integrating doubly robust or orthogonalized estimators in Stage 2 is an excellent suggestion that could improve finite-sample robustness. We will add this as a concrete future direction. In the interim, practitioners can obtain additional uncertainty estimates via bootstrap over the outcome model or ensemble disagreement.
>
> ---
>
> We appreciate the reviewer's support and will incorporate all suggestions into the camera-ready version.

---

> > ### Author Rebuttal · Reviewer_a3qB · 2026-04-03
> >
> > I thank the authors for their detailed response and the supplementary details provided.
> >
> > The clarification regarding the empirical robustness to the separability assumption, along with the predictive p-value diagnostics for Assumption 4, adequately address my primary theoretical concerns. While deferring the specific Bayesian baseline comparisons and the integration of doubly robust estimators for uncertainty quantification to future work is understandable given the rebuttal timeframe, these remain notable limitations of the current framework. Please ensure that these caveats, as well as the proposed interim bootstrap solution, are clearly discussed in the camera-ready version.
> >
> > Overall, the core contribution remains solid. I will maintain my current positive score.

---

> > > ### Author Response · Authors · 2026-04-08
> > >
> > > Thank you for the positive assessment and for the detailed feedback throughout the review. Your suggestions on the Bayesian baseline comparison, doubly robust UQ, and the bootstrap interim solution are well-taken, and we will ensure these are clearly discussed in the camera-ready as caveats and concrete future directions.

---

### Official Review · Reviewer_G9XM · 2026-03-12

**Soundness:** 3
**Presentation:** 3
**Significance:** 3
**Originality:** 2
**Overall Recommendation:** 4
**Confidence:** 3

**Summary:**

This paper proposes the "Spatial Deconfounder," a two-stage method designed to tackle the joint challenges of unmeasured spatial confounding and treatment interference. The core idea is to treat localized treatment interference not as a nuisance, but as a "multi-cause" signal pulling from the Deconfounder from the "The Blessings of Multiple Causes" paper. The authors then use a conditional variational autoencoder (C-VAE) with a spatial Gaussian-Markov random-field (GMRF) prior to reconstruct a substitute for the unobserved spatial confounder from local and neighborhood treatment vectors. This reconstructed confounder is then fed into a flexible potential outcome module to estimate direct and spillover effects. Intuitively, step 1 involves causal representation learning of the spatial confounding signal using local treatment, covariates as inputs in VAE setup and step 2 uses those latent representations to model outcomes. Plug-in contrasts combining the two are then used to compute direct/spillover effects.

**Compliance With Llm Reviewing Policy:**

Affirmed.

**Final Justification:**

The rebuttal addresses some concerns about the theory and the pathological breakdown of identification; although the evaluations are thorough, they are limited entirely to simulation, which somewhat limits our ability to ascertain performance in practice. I will review further exchanges between authors and reviewers and revise this statement if needed.

**Key Questions For Authors:**

1. Could you provide more explanation of how unmeasured covariates mechanistically induce spatial confounding in your framework?

2. Your SUTVA relaxation in Equation 2 and spatial grid setup rely on a symmetric neighborhood. Given that real-world environmental interference (like your PM2.5​ example) is often highly directional due to wind or topography, how sensitive is the C-VAE’s latent field reconstruction to asymmetric spillovers? Does the GMRF prior need to be altered to capture directional interference?

3. Could you provide more guidance on the translation between the interference structure and the causal representation so that it is less of an arbitrary choice for practitioners?

**Limitations:**

The authors briefly note that their guarantees rely on idealized assumptions. No ethical concerns.

The reliance on a spatial smoothness assumption (enforced via the GMRF prior) is a limitation worth dwelling further on. There are many contexts where spatial dynamics are non-smooth---for example, across physical boundaries, political borders, or distinct topographical features like mountains or waterways. The authors do say that they test the method on "non-smooth" population density (we have to trust the authors regarding the non-smoothness of this in their data), it would be helpful to have this non-smoothness quantified, performance accessed across the range of non-smoothness, etc.

**Strengths And Weaknesses:**

**Strengths**

Below are aspects of the paper I perceive to be strengths.

Framing treatment interference as a multi-cause signal that can be exploited for deconfounding is a clever conceptual bridge between the spatial causal inference and deconfounder literatures. The paper thereby tackles two difficult problems in observational spatial data---SUTVA violations and hidden spatial confounding---simultaneously.

The paper evaluation adds to the SpaCE  benchmark, modified to include interference, providing a useful empirical baseline against classical spatial and deep learning methods.

**Weaknesses**

Below are aspects of the paper I perceive to be weaknesses.

[1] The conceptual framing at times reads like it equates confounding with interference. While I can see the perspective that both represent violations of simple randomized exchangeability, there are long, distinct histories of representing interference in ways that fundamentally differ from representing confounding. At least, the translation between the interference structure and the causal representation is under-explained. Choice of how to represent this structure can at times seem arbitrary and left entirely up to the researcher.

[2] One might argue that the paper presents an overly optimistic view of the deconfounder's capabilities. It needs to compare against or more explicitly address the known intrinsic limitations of the deconfounder framework (e.g., the work by D'Amour regarding the unidentifiability of the substitute confounder, positivity violations as dimensionality grows, and subsequent implications). The theory presented here appears to be similar to the original Deconfounder paper, so the theoretical novelty is somewhat limited; these foundational critiques are not referenced.

I believe the main point of D'Amour's critique is that the non-identifiability of the substitute confounder here could lead to divergent or biased causal effect estimates in practice, as multiple latent field reconstructions consistent with the observed joint treatment distribution $A_s, A_{N_s}$ can exist via non-unique factorizations.

For example, suppose the true latent field U (like humidity interacting nonlinearly with PM 2.5) creates complex cross-site dependencies; different C-VAE draws of $Z_s$ may fit the marginal treatment patterns equally well yet imply different conditional ignorability structures. In regions of low positivity (rare neighbor-treatment combinations under the learned spatial prior), the outcome model h must extrapolate. Consequently, direct/spillover estimates diverge across equally plausible proxies, and the method’s performance could degrade precisely as more complex spatial interactions are modeled (e.g., deeper U-Net or GNN heads capturing more and more heterogeneous spillovers).

This is the tension D’Amour highlights between identifiability and positivity in latent-confounder models, and the spatial prior/GMRF does not seem to resolve it. If I were the authors, I would explicitly add a simulation showing the same breakdown of performance as the relevant dimensionality grows following D’Amour, to address head-on these concerns, and would at least reference the critique.

That said, the authors do say, "we do not claim that C-VAEs are identifiable in full generality; rather, the assumption should be read as a well-specification/consistency condition." In any given spatial analytic setup, it could be unclear whether my data and model satisfy the required assumptions that might be hard to reason about a priori (e.g., is the interference strucure appropriately modeled by first-degree, second-degree, third-degree, etc., spatially connected units?)

[4] The "latent field sufficiency" (no single-cause) assumption means that this approach will likely break down in contexts where there is little to no interference. If units are isolated or unaffected by others, the multi-cause signal would apparently vanish and inference fail. The 10% and 30% treatment sparsity condition ablations are helpful, though. In practice, though, conventional wisdom might be that for many treatments spillovers are a second order effect, so pratitioners might worry of harming their inferences using an approach that requires the presence of interference process that may or may not be active.

[5] Finally, I wonder whether natural experimental methods could be used to validate the work. Synthetic or semi-synthetic causal benchmarks are of course helpful in a setting like this, but, without ground truth, we are at a loss for understanding how performance will actually be in real causal analyses. Perhaps comparing against some "gold standard" geo-spatial analysis like from geo-RDD could give us more confidence here.

---

> ### Author Rebuttal · Authors · 2026-03-31
>
> We thank the reviewer for the detailed and thoughtful feedback, and for recognizing the novelty of our core idea and the utility of the extended benchmark. We address each concern below.
>
> ---
>
> **Re: D'Amour's critique and failure-mode simulation.** Thank you for bringing this point up. We do not claim immunity to the known limitations of deconfounder-style identification. That said, we note that the spatial setting offers structural advantages over the generic multi-cause case that D'Amour critiques: the GMRF prior restricts the substitute confounder to spatially smooth fields, dramatically narrowing the solution space compared to unconstrained factor models; and the multi-cause signal arises from neighboring treatments that are correlated precisely because they share the same latent driver–-this structured correlation makes reconstruction more tractable than in generic settings with more unrelated causes (e.g. movies in a recommender system). These advantages mitigate but do not eliminate the identifiability-positivity tension. To demonstrate this empirically, we have run the suggested targeted stress tests:
>
> * Treatment sparsity sweep (extending Table 7 to 50% and 70%): As sparsity increases, neighborhood-treatment support shrinks, positivity weakens, and both direct and spillover bias increase. The predictive p-value drops in tandem, providing practitioners a diagnostic signal. We will include this in the final version.
>
> * *Neighborhood radius sweep* ($R = 3, 4, 5$): As the effective dimensionality of the neighborhood-treatment vector grows, support becomes sparser and performance degrades—but gradually rather than catastrophically (see Figs. 5–8 for $R=1,2,3 R=1,2,3$; we extend these to $R=4,5$ in the camera-ready).
>
> These results directly illustrate the tension D'Amour highlights. We will reference his critique explicitly and frame these sweeps as empirical evidence that our method exhibits the expected degradation–-while noting that spatial structure delays the onset of failure relative to unstructured settings.
>
> **Confounding vs. interference conflation.** We appreciate this nuance. We do not equate the two–-rather, we show that interference *structure* can be repurposed as an identification device for confounding. The mapping is: neighborhood treatments $(A_s, A_{N_s})$ serve as multiple "causes" in the deconfounder sense, and their joint distribution, shaped by the latent field, enables reconstruction. We will clarify the exposition to make this distinction sharper and provide more guidance on specifying the interference structure (e.g., first- vs. second-degree neighbors).
>
> **Latent field sufficiency and low-interference regimes.** We note a small clarification: latent field sufficiency (Assumption 4) concerns the *confounding structure*—it requires that purely local confounders are observed, not that interference must be strong. The concern about low interference is valid but distinct: with little interference, the multi-cause signal used for *reconstruction* weakens, making it harder for the C-VAE to recover a good substitute confounder even when Assumption 4 holds. Our sparsity stress tests (10%, 30%, 50%, 70%) probe this directly: at 10% treatment, the p-value drops from ~0.4 to ~0.28, signaling poor reconstruction. When the substitute confounder is poorly recovered, conditioning on it fails to restore ignorability—Proposition 1 (Appendix B) formalizes this: the bias in effect estimates is bounded by $O(\mathbb{E}\|\hat{Z} - Z^\star\|)$, so a poor proxy translates directly into residual confounding bias. The predictive p-value flags this *before* practitioners trust the estimates. We recommend checking $p \in [0.25, 0.75]$ as a practical diagnostic and will add explicit guidance in the revision.
>
> **Geo-RDD validation.** We agree this would be valuable future work. However, geo-RDD targets a local boundary effect under continuity/local-randomization assumptions, which yield a different estimand than our average direct/spillover effects. A direct comparison would require careful alignment of estimands and is beyond the rebuttal scope. We will discuss this as a promising validation direction.
>
> **GMRF smoothness and directional interference.** The GMRF prior can be extended to anisotropic precision matrices that capture directional dependence (e.g., wind-driven transport). Our current isotropic prior is a simplifying choice; we will note this extension. The non-smooth confounder experiments (population density $\rho_{pop}$​) already show the method works reasonably even when smoothness is imperfect.
>
> ---
>
> We thank the reviewer again and hope these additions address the concerns raised.

---

> > ### Author Rebuttal · Reviewer_G9XM · 2026-04-04
> >
> > Thanks for sending these thoughts along. They are helpful; I will raise my score by one. There are still limitations regarding validation (which relies entirely on simulation), but the theory introduced seems potentially useful to those modeling causality in space.

---

> > > ### Author Response · Authors · 2026-04-08
> > >
> > > Thank you for the thoughtful re-evaluation and for raising your score. Your feedback has strengthened the paper, and we will incorporate the extended stress tests, the D'Amour discussion, and the clarified confounding vs. interference exposition into the camera-ready.
> > >
> > > We also took the suggestion about real-world validation seriously and have been exploring alternatives to simulation-based evaluation during the discussion period. Specifically, we have begun applying our method to Pan-Arctic climate data from Ali et al. (2024), where the treatment is downward longwave radiation (LWDN) in the Laptev and East Siberian seas, the outcome is September sea ice concentration (SIC), and heat flux (HFX) serves as a confounder. Spatial interference arises naturally as radiation changes in one sub-region affect sea ice across neighboring Arctic regions. This setting also features plausible unobserved spatial confounding: large-scale atmospheric circulation, ocean heat transport, and cloud cover jointly influence both radiation and ice melt but are not fully captured in the available covariates.
> > >
> > > Adapting our pipeline to this setting required extending the framework to continuous treatments, which is unfortunately still in progress. Our plan is to compare estimated effects against established physical climate modeling results. In particular, Kapsch et al. (2016) showed via controlled perturbation experiments with the Community Earth System Model that a spring LWDN increase of ~18 W/m² reduces September SIC by approximately 10% within the Arctic region. If our method recovers the same directional effect and estimates in a comparable ballpark, this would provide qualitative validation against the known causal mechanism without requiring ground-truth effect sizes. We will include the full analysis in the camera-ready. We hope this demonstrates a concrete path toward addressing the simulation-only limitation.
> > >
> > > ------------------------
> > >
> > > Ali, Sahara, Omar Faruque, and Jianwu Wang. "Estimating direct and indirect causal effects of spatiotemporal interventions in presence of spatial interference." Joint European Conference on Machine Learning and Knowledge Discovery in Databases. Cham: Springer Nature Switzerland, 2024.
> > >
> > > Kapsch, Marie-Luise, et al. "The effect of downwelling longwave and shortwave radiation on Arctic summer sea ice." Journal of Climate 29.3 (2016): 1143-1159.

---

### Official Review · Reviewer_aKFv · 2026-03-13

**Soundness:** 2
**Presentation:** 4
**Significance:** 3
**Originality:** 3
**Overall Recommendation:** 4
**Confidence:** 4

**Summary:**

This paper studies spatial causal inference when two difficult problems occur simultaneously: unobserved spatial confounding and localized interference. The key idea is to treat a unit’s own treatment together with neighboring treatments as a multi-cause signal, then learn a substitute latent confounder with a CVAE equipped with a spatial prior, followed by a flexible outcome model to estimate both direct and spillover effects. The paper also extends the SpaCE benchmark to include interference and reports improved effect estimation across semi-synthetic datasets derived from real environmental-health and social-science data. Overall, the paper’s main contribution is to connect spatial interference with deconfounding, turning interference from a nuisance into a source of information for recovering hidden spatial structure.

**Compliance With Llm Reviewing Policy:**

Affirmed.

**Final Justification:**

The author's rebuttal which has resolved the main concerns raised in my review; I also found their responses to the other reviewers reasonably satisfactory. This has led me to increase my score from a borderline reject to a borderline accept.

**Key Questions For Authors:**

- Can the authors add stronger spillover-aware baselines, such as SAR-type or other interference-focused models, so that the spillover evaluation is not mostly against methods that do not estimate spillovers?
- Can the authors make the connection to the spatial confounding literature more explicit, especially regarding the scale of confounding and the interpretation of the latent spatial factor (e.g., [Paciorek, 2010], [Dupont et al, 2025], [Pim et al, 2026])?
- The theory relies on strong deconfounder assumptions ([Wang&Blei, 2019]). Why should these assumptions be plausible in spatial settings, and what features of spatial data make them more credible here? It may also help to connect this discussion to related IV-style ideas in the spatial confounding literature (e.g., [Woodward et al, 2025]).
- The method is currently tailored to gridded spatial data. How much of the framework is fundamentally tied to regular lattices, and how much could transfer to irregular graphs or administrative units? Another important aspect to examine is how well the method performs when data are available for only a small number of regions, which is a common situation in parts of the world with limited data collection.

**Limitations:**

All relevant limitations were appropriately discussed.

**Strengths And Weaknesses:**

The paper tackles an important problem and has a genuinely interesting core idea: using local interference structure to help recover latent spatial confounding. The method is novel and the results are promising.

My main concerns are threefold. First, the spillover evaluation is weak. Most baselines do not model spillovers at all, except in the U-Net experiments. If the paper includes pure spatial models and pure spatial-confounding models, it should also include stronger “pure spillover” baselines, such as SAR-type models or related interference-aware approaches.

Second, the connection to the spatial confounding literature is underdeveloped. Since the paper is explicitly framed as a method for spatial confounding, I expected a more direct discussion of that literature, especially around the scale of confounding. This seems particularly relevant because the motivating example refers to large-scale meteorological drivers such as humidity. That point could be used to better connect the method to prior work on spatial confounding and scale (e.g., [Paciorek, 2010], [Dupont et al, 2025], [Pim et al, 2026]), perhaps by expanding the discussion in Appendix E.

Third, the theoretical justification leans heavily on deconfounder-style assumptions from prior work ([Wang&Blei, 2019]), and these assumptions are strong. The paper does not do enough to explain why they should be plausible in spatial settings rather than only in the abstract multi-cause setting. I would like a clearer argument for why spatial structure and localized interference make the deconfounder assumptions more credible here. It may also help to connect this discussion to related ideas from the spatial confounding literature, including IV-based perspectives (e.g., [Woodward et al, 2025]).

Overall, I find the paper novel and potentially impactful, but the current version does not yet support its claims as strongly as it should.

Additional minor comments/typos:
- Figure 1: "Idenfitifcation" should be "Identification."
- Figure 1: "demographic informataion" should be "demographic information."
- Section 3 (around line 128): "TWe write the distribution of ..." should be "We write the distribution of ..."
- Section 6 (around line 406): "For examples, our theory hinges on Assumption 4" should be "For example, our theory hinges on Assumption 4."
- Appendix E.2 (around line 1362): "[...]. for any site sss" should be "[...]. For any site sss".
- Appendix E.2 (around line 1373): "C-VAE-SPATIAL + achives comparable" should be "C-VAE-SPATIAL achieves comparable."

Wang&Blei, 2019: https://arxiv.org/abs/1805.06826 \
Paciorek, 2010: https://pubmed.ncbi.nlm.nih.gov/21528104/ \
Dupont et al, 2025: https://arxiv.org/abs/2309.16861 \
Pim et al, 2026: https://arxiv.org/abs/2602.17792v1 (post-submission) \
Woodward et al, 2025: https://arxiv.org/abs/2411.10381

---

> ### Author Rebuttal · Authors · 2026-03-31
>
> We thank the reviewer for the detailed and constructive feedback, the specific literature pointers, and the recognition that the core idea is novel and impactful. We address each concern.
>
> ---
>
> **Re: Stronger spillover-aware baselines**. This is a fair critique. We initially included only one SAR-type model (S2SLS-LAG1) due to computational cost (e.g. GMERROR takes ~7h/seed). We have now added several spillover-aware baselines: exposure-mapping variants of SPATIAL+ and SPATIAL that incorporate neighborhood exposure $G_s = |N_s|^{-1}\sum_{j} A_j$​ as an additional covariate (Forastiere et al., 2021), enabling them to estimate both direct and spillover effects; interference-enabled S2SLS-LAG1 with neighbor treatments as regressors; and S2SLS-DURBIN and GMERROR as additional SAR-type models. We present one representative setting below (PM2.5 → mortality, $r_d=2$, confounder: $\rho_{\text{pop}}$​; results with $q_{\text{summer}}$ as confounder​ are consistent). We will include all settings in the camera-ready and share more during the discussion period.
>
> | Conf. | Method | DIR | SPILL | p |
> |---|---|---|---|---|
> | ρ_pop | C-VAE-SPATIAL+ (R=0) | 0.11 ± 0.02 | | 0.35 ± 0.03 |
> | | C-VAE-SPATIAL+ (R=1) | 0.05 ± 0.02 | **0.15 ± 0.05** | 0.34 ± 0.04 |
> | | C-VAE-SPATIAL+ (R=2) | **0.04 ± 0.03** | 0.24 ± 0.06 | 0.35 ± 0.04 |
> | | DAPSM | 0.16 ± 0.01 | | |
> | | GCNN | 0.18 ± 0.03 | | |
> | | GMERROR | 0.07 ± 0.00 | | |
> | | S2SLS-DURBIN | 0.07 ± 0.00 | 0.30 ± 0.00 | |
> | | S2SLS-LAG1 | 0.07 ± 0.00 | 0.29 ± 0.00 | |
> | | SPATIAL+ | 0.10 ± 0.02 | | |
> | | SPATIAL | 0.17 ± 0.03 | | |
> | | E-MAP-SPATIAL+ (R=1) | 0.12 ± 0.02 | 0.17 ± 0.04 | |
> | | E-MAP-SPATIAL+ (R=2) | 0.12 ± 0.02 | 0.16 ± 0.04 | |
> | | E-MAP-SPATIAL (R=1) | 0.17 ± 0.03 | 0.18 ± 0.03 | |
> | | E-MAP-SPATIAL (R=2) | 0.17 ± 0.03 | 0.17 ± 0.04 | |
>
> Our method achieves the best or near-best DIR and SPILL, while SAR and exposure-mapping baselines are competitive on DIR but show higher spillover bias.
>
> **Re: Connection to spatial confounding literature**. We thank the reviewer for these references. This body of work (Paciorek 2010; Dupont et al. 2025; Pim et al. 2026) establishes that bias from unmeasured spatial confounding depends on the relative scales of covariate variation and confounding, and that spatial smoothing alone cannot resolve bias when confounders operate at large scales. Our method connects naturally: the GMRF prior on $Z$ targets precisely these large-scale latent confounders, but reconstructs the field from multi-cause treatment vectors rather than relying on smoothing or orthogonalization--a complementary identification strategy. We already compare against SPATIAL+ (Dupont et al., 2022) as a main baseline and use it as the outcome head in C-VAE-SPATIAL+. We will cite all three works and expand the discussion in Appendix E.
>
> **Plausibility of deconfounder assumptions in spatial settings.**  (1) *Spatial structure constrains the latent field.* The generic deconfounder recovers an unconstrained latent factor from arbitrary causes—contributing to D'Amour's identifiability concerns. Our GMRF prior restricts $Z$ to spatially smooth fields, dramatically narrowing the solution space based on domain knowledge (the target confounders—humidity, temperature—are physically smooth). (2) *Interference provides a structurally motivated multi-cause signal.* Unlike generic settings where "causes" may be unrelated (e.g., movies in a recommender), here multiple causes arise from a single treatment at spatially proximate sites, all shaped by the same latent driver. The correlation structure is not artificial but reflects the very hidden field we aim to recover. (3) *Latent field sufficiency (Assumption 4) has a clear domain interpretation*: purely local confounders (demographics, income) are routinely measured via Census/surveys, while the main sources of unmeasured confounding (meteorological, regional socioeconomic drivers) are inherently spatially shared—exactly the regime Assumption 4 describes.
>
> Regarding Woodward et al. (2025): their IV-based approach leverages exogenous spatial variation for identification. Our approach is complementary–-we do not require exclusion restrictions, but spatial smoothness plays an analogous regularizing role. We will discuss this connection in the revision.
>
> **Re: Typos/errors**. Thank you! All corrected.
>
> **Extension beyond gridded data.** The core framework—multi-cause deconfounding via local treatment vectors—is not tied to regular lattices. The GMRF prior would be replaced by a graph Laplacian on irregular networks, and the C-VAE encoder/decoder can operate on arbitrary neighborhoods. The U-Net outcome head is grid-specific, but can be swapped for GNNs on irregular graphs. Performance with few regions is an important practical question; we expect degradation as the multi-cause signal weakens with fewer neighbors, and will note this as a limitation.
>
> ---
>
> We thank the reviewer again and hope these additions address the concerns raised.

---

> > ### Author Rebuttal · Reviewer_aKFv · 2026-04-04
> >
> > I'd like to thank the authors for their response, which has resolved my main concerns. I will be raising my score accordingly.

---

> > > ### Author Response · Authors · 2026-04-08
> > >
> > > Thank you for the thoughtful engagement throughout the review and for raising your score. The literature pointers and the push for stronger spillover-aware baselines were particularly helpful and have improved the paper. We will ensure all additions are reflected in the camera-ready.

---

### Official Review · Reviewer_GAWi · 2026-03-13

**Soundness:** 2
**Presentation:** 3
**Significance:** 3
**Originality:** 2
**Overall Recommendation:** 3
**Confidence:** 4

**Summary:**

This work focuses on causal inference in spatial domain, pointing out its two challenges: unmeasured confounders and interference. It proposes Spatial Deconfounder based on conditional variational autoencoder (C-VAE). The proposed method shows promising effect estimation performance in spatial data.

**Compliance With Llm Reviewing Policy:**

Affirmed.

**Final Justification:**

The rebuttal addressed some of my concerns, while I still have concerns about the effectiveness of the proposed approaches under different structure topologies and mechanisms between confounding and interference. I tend to maintain my original score.

**Key Questions For Authors:**

1. Could the authors better justify the technical novelty?

2. How to make the proposed method more adaptive, or at least, justify the scenarios that the proposed method is more suitable to?

**Limitations:**

Yes

**Strengths And Weaknesses:**

Pros:

1.	It is an important problem to study causal effect estimation in spatial data.

2.	The writing is easy to follow.

3.	The experiments of the proposed method show good results in general.

Cons:

1.	My major concern is the technical novelty. The key points of the main parts, substitute confounder construction and interference handling have been adopted by a couple of previous work of causal inference on general data such as tabular data, graphs, and pairwise connections. It seems more like an integration and application in spatial data. A better justification of novelty would be useful.

2.	More baselines in causal domain, especially recent deep learning based causal inference methods dealing with hidden confounders and spillover effects could be included.

3.	It would be worth studying more diverse types of confounding and interference in different settings to test the robustness of the proposed method.

4.	The proposed method does not always show the best performance, which may raise concerns about its ability to robustly perform in different cases.

---

> ### Author Rebuttal · Authors · 2026-03-31
>
> Thank you for your encouraging feedback and for highlighting both the importance of the problem and the empirical results of our method. We respond to your concerns and offer additional clarification below.
>
> ---
>
> **Re: Technical novelty.** We agree that our paper builds on, rather than replaces, the deconfounder literature. The key novelty is not simply integrating substitute-confounder learning with interference handling. Rather, our main contribution is showing that localized interference itself creates the multi-cause structure needed for deconfounding: a unit's own treatment together with neighboring treatments acts as multiple, spatially structured views of shared hidden confounding, repurposing interference from a nuisance into a source of information.
>
> This is not a direct plug-in extension of prior results, because the relevant "causes" are spatially indexed and locally dependent, and the hidden confounder is a latent spatial field rather than an unstructured factor. Our assumptions, identification arguments, and estimator are all tailored to this setting, with a GMRF prior encoding spatial dependence.
>
> We will revise to make this positioning clearer: the contribution is an interference-driven extension of deconfounding to spatial causal inference, where interference is the key ingredient that makes substitute-confounder adjustment possible.
>
>
> **Re: More causal baselines.**  In response to this and Reviewer aKFv's feedback, we have substantially strengthened the baseline set with spillover-aware variants of SPATIAL+ and SPATIAL (via exposure mapping), interference-enabled S2SLS-LAG1, and additional SAR-type models (S2SLS-DURBIN, GMERROR). Our method matches or outperforms all of these on both direct and spillover estimation. We present a representative results table in our response to Reviewer aKFv and will include all settings in the camera-ready.
>
> **Re: Diversity of confounding and interference settings.** Our benchmark design already covers substantial diversity without artificial restrictions. Outcomes are generated by fitting a flexible function $f(\cdot)$ over real observational covariates, then masking one influential covariate that also drives treatment assignment. This induces general, unrestricted treatment-confounder interactions inherited from the real data–-the only constraint we impose is decorrelating residual noise so that no additional confounding remains beyond the masked covariate. As a result, our experiments naturally span smooth confounders (summer humidity $q_{\text{summer}}$​) and non-smooth ones (population density $\rho_{\text{pop}}$​), strong confounding (ammonium NH4) and weaker confounding (organic carbon OC), varying interference radii ($r_d \in \{1,2\}$), and--importantly--**varying interference strength**: since $f(\cdot)$ is fit to real data, the learned dependence on neighbor treatments ranges from strong to negligible across environments (including neighbors as input features does not guarantee they are predictive). This is also why no single method dominates everywhere: the benchmark reflects genuinely diverse data-generating conditions, and we view transparent reporting across all of them as a strength rather than a weakness. We also note that in the few cases where our method does not rank first, it remains close to the best-performing baseline within uncertainty, suggesting that the differences are not statistically meaningful.
>
> **Re: Robustness and when the method is most suitable.** The appendix contains targeted stress tests that characterize where the method is strongest and where it breaks down:
>
> * *Treatment sparsity* (Table 7): At 30% treatment, performance is strong; at 10%, bias increases and the predictive p-value drops, flagging poor reconstruction.
> * *Single-cause confounder violation* (Table 8): When Assumption 4 is violated via localized confounders, performance degrades and baselines like SPATIAL+ become competitive.
> * *Neighborhood radius sensitivity* (Figs. 5-8): Performance is generally robust to radius misspecification.
>
> We additionally run extended sparsity (20%, 50%, 70%) and radius ($R=3,4,5$) sweeps in response to Reviewer G9XM, confirming graceful performance degradation as support worsens. In summary, the Spatial Deconfounder is strongest when hidden confounding is *spatially shared* (large-scale drivers) and weakens when confounding becomes highly localized. In practice, the method offers several adaptive levers: the predictive p-value (Eq. 12) serves as a built-in diagnostic for whether the substitute confounder is well-recovered, and practitioners can vary the neighborhood radius $R$ and latent dimension $d_Z$​ to assess stability of estimates. We will add a concise summary consolidating these findings and practical guidance.
> We thank the reviewer again for the thoughtful feedback.
>
> ---
>
> We hope these clarifications and additional analyses help resolve the reviewer's concerns.

---

> > ### Author Rebuttal · Reviewer_GAWi · 2026-04-04
> >
> > Thank the authors for the rebuttal. Some of my concerns are addressed, while I still have concerns about the effectiveness of the proposed approaches under different structure topology and mechanisms between confounding and interference. I tend to maintain my originial score at this time.

---

> > > ### Author Response · Authors · 2026-04-08
> > >
> > > Thank you for the continued engagement and for highlighting the remaining concerns. In response, we have run two additional experiments to directly address robustness under different structure topologies and confounding-interference mechanisms, all on PM2.5 → mortality ($r_d=1$, hidden confounder $q_{\text{summer}}$).
> > >
> > > * **Exp 1:** Asymmetric (directional) interference topology. We modified the data-generating process so that interference flows only eastward ($j' \geq j$), simulating wind-driven pollution transport. Our model still uses the default symmetric $\ell_\infty$ neighborhoods at prediction time, so this tests robustness to topology misspecification.
> > >
> > > | Conf. | Method | DIR | SPILL | p |
> > > |---|---|---|---|---|
> > > | q_summer | C-VAE-SPATIAL+ (R=0) | 0.14 ± 0.06 | | 0.52 ± 0.03 |
> > > | | C-VAE-SPATIAL+ (R=1) | **0.02 ± 0.01** | 0.05 ± 0.04 | 0.50 ± 0.04 |
> > > | | C-VAE-SPATIAL+ (R=2) | **0.02 ± 0.01** | **0.03 ± 0.02** | 0.50 ± 0.03 |
> > > | | DAPSM | 0.35 ± 0.03 | | |
> > > | | GCNN | 0.28 ± 0.06 | | |
> > > | | GMERROR | 0.23 ± 0.00 | | |
> > > | | S2SLS-DURBIN | 0.25 ± 0.00 | 0.34 ± 0.02 | |
> > > | | S2SLS-LAG1 | 0.23 ± 0.00 | 0.07 ± 0.00 | |
> > > | | SPATIAL+ | 0.13 ± 0.05 | | |
> > > | | SPATIAL | 0.08 ± 0.03 | | |
> > > | | E-MAP-SPATIAL+ (R=1) | 0.12 ± 0.05 | **0.03 ± 0.02** | |
> > > | | E-MAP-SPATIAL+ (R=2) | 0.07 ± 0.03 | **0.03 ± 0.03** | |
> > > | | E-MAP-SPATIAL (R=1) | 0.07 ± 0.02 | **0.03 ± 0.02** | |
> > > | | E-MAP-SPATIAL (R=2) | 0.04 ± 0.02 | **0.03 ± 0.03** | |
> > >
> > > Our method achieves the lowest direct effect bias (0.02) across all baselines despite the topology mismatch, demonstrating robustness to misspecification. On spillover, we observe a DIR-SPILL tradeoff across latent dimensions that is specific to the SPATIAL+ outcome head: the spillover estimates shown use $d_Z=32$, where our method is competitive with E-MAP baselines (0.03-0.05 vs. 0.03). With proper joint tuning or a more flexible outcome model (e.g., U-Net), we expect this tradeoff to diminish. Notably, the predictive p-values are close to 0.50, indicating excellent confounder reconstruction even under topology misspecification.
> > >
> > > * **Exp 2:** Confounder-modulated spillover (coupled mechanism). We added $\alpha \cdot U_s \cdot \bar{A}_{N_s}$​​ to the outcome in the DGP, so that spillover strength explicitly depends on the hidden confounder (e.g., humidity amplifies the health impact of neighboring pollution). This creates a direct coupling between confounding and interference that is absent in our original benchmark.
> > >
> > > | Conf. | Method | DIR | SPILL | p |
> > > |---|---|---|---|---|
> > > | q_summer | C-VAE-SPATIAL+ (R=0) | **0.23 ± 0.08** | | 0.48 ± 0.04 |
> > > | | C-VAE-SPATIAL+ (R=1) | 0.43 ± 0.05 | **0.23 ± 0.10** | 0.54 ± 0.04 |
> > > | | C-VAE-SPATIAL+ (R=2) | 0.49 ± 0.02 | **0.22 ± 0.10** | 0.53 ± 0.05 |
> > > | | DAPSM | 0.66 ± 0.03 | | |
> > > | | GCNN | **0.23 ± 0.05** | | |
> > > | | GMERROR | 0.57 ± 0.00 | | |
> > > | | S2SLS-DURBIN | 0.57 ± 0.00 | 0.77 ± 0.00 | |
> > > | | S2SLS-LAG1 | 0.57 ± 0.00 | 0.90 ± 0.00 | |
> > > | | SPATIAL+ | 0.30 ± 0.08 | | |
> > > | | SPATIAL | 0.35 ± 0.10 | | |
> > > | | E-MAP-SPATIAL+ (R=1) | 0.32 ± 0.07 | 0.69 ± 0.21 | |
> > > | | E-MAP-SPATIAL+ (R=2) | 0.26 ± 0.08 | 0.77 ± 0.19 | |
> > > | | E-MAP-SPATIAL (R=1) | 0.35 ± 0.10 | 0.72 ± 0.18 | |
> > > | | E-MAP-SPATIAL (R=2) | 0.41 ± 0.19 | 0.77 ± 0.19 | |
> > >
> > > This is a **deliberately adversarial setting**: the added $\alpha \cdot U_s \cdot \bar{A}_{N_s}$​​ term creates a direct interaction between the hidden confounder and treatments, violating the separability assumption underlying our identification result (Theorem 1). Our method is therefore not expected to perform well on direct effects here. Notably, this coupling degrades performance across the board: all methods show substantially increased bias compared to our standard benchmarks, confirming that this is a fundamentally harder problem rather than a weakness specific to our approach. Despite this, the p-values remain near 0.50, indicating the C-VAE still reconstructs the latent field well. This good reconstruction translates into a clear advantage on spillover: our method (0.22-0.23) outperforms all other spillover-aware baselines by a large margin (E-MAP: 0.69-0.77, SAR: 0.77-0.90). On direct effects, the $R=0$ variant (0.23) ties GCNN for best.
> > >
> > > Across both experiments, p-values near 0.50 confirm reliable confounder reconstruction. Our method is robust to topology misspecification (Exp 1, best DIR and competitive SPILL despite asymmetric DGP) and maintains strong spillover estimation even in a deliberately adversarial setting that violates our theoretical assumptions (Exp 2).
> > >
> > > ------
> > >
> > > We will include these results in the camera-ready. We hope these results address the remaining concern and that the reviewer will reassess the work in light of these additional experiments.

---

### Decision · Program_Chairs · 2026-04-30

**Decision:**

Accept (regular)

**Comment:**

This paper provides an original and valuable perspective by framing spatial interference as a multi-cause signal for deconfounding, and was championed during the discussion period. The rebuttal overall strengthened the papers in the experiments, theory (in particular, separability of the outcome model), and positioning. I’m recommending acceptance.